

# The Pacific Ocean heat engine: global climate's regulator

Roger N Jones* and James H Ricketts

Institute of Sustainable Industries and Liveable Cities, Victoria University, Melbourne, Victoria 8001, Australia

*Correspondence to*: Roger N. Jones (roger.jones@vu.edu.au)

**Abstract**

Climate change is routinely represented as a smoothly changing signal surrounded by statistical noise. However, on decadal timescales, warming proceeds as a sequence of steady-state regimes punctuated by abrupt shifts. Here we present evidence that this process is regulated by a heat engine spanning the tropical Pacific Ocean. The eastern-central Pacific maintains steady-state conditions, collecting heat and delivering it to the Western Pacific warm pool. This acts as distributor,

transporting heat upwards and to the poles. The heat engine is networked within the climate system, linking different oscillations and circulations as heat energy is dissipated. The process is self-regulating. Steady-state regimes will persist until they become unstable and need more or less power depending on the direction of forcing. Under greenhouse gas forcing, shifts initiated within the heat engine propagate broadly across the shallow ocean, followed by warming over land and at higher latitudes. The heat engine was in free mode during the early 20[th] century, dominated by decadal variability. From the

1960s, it switched into forced mode, initiating a stepladder-like pattern of warming in regional and global climate. The most recent shift commenced in the warm pool in December 2012, ending the so-called hiatus (1997–2013). During 2014–15, surface temperatures warmed abruptly by ~0.25 °C globally and >0.5 °C over northern hemisphere land and high latitudes. With increasing forcing, the heat engine will shift more frequently. Rapid decreases in greenhouse gas emissions will slow the process and potentially, could stabilise it. Managing unavoidable change requires developing the capacity to predict

shifts in advance. Planning for rapid changes in extreme events is an urgent priority.

## 1    Introduction

Understanding how the climate responds to increasing greenhouse gases on decadal timescales is essential for managing changing climate-related risk. In its Fifth Assessment Report, the IPCC endorsed linear warming, stating "On decadal to interdecadal timescales and under continually increasing effective radiative forcing (ERF), the forced component of the

GMST (global mean surface temperature) trend responds to the ERF trend relatively rapidly and almost linearly (medium confidence; Stocker et al., 2013)." While over interdecadal timescales long-term warming may be considered linear if averaged decadally, on decadal timescales, it forms a staircase-like pattern (Belolipetsky et al., 2015;Jones and Ricketts, 2017). Findings of step-like change over decadal timescales have been reported for the atmosphere (Belolipetsky et al., 2015;Bartsev et al., 2016;Bartsev et al., 2017;Saltykov et al.;Varotsos et al., 2019;Chikamoto et al., 2012;Jones,





2012;Belolipetsky, 2014), oceans (Reid, 2016;Reid and Beaugrand, 2012;Beaugrand, 2004;Thompson et al., 2017) and in widespread ecosystem and environmental responses (Reid et al., 2016;Beaugrand et al., 2019;Conversi et al., 2010;Overland et al., 2008).

Standard practice treats natural climate variability and anthropogenic forcing as separate processes (Kirtman et al., 2013). This is encapsulated in the signal-to-noise model of trend-like change upon which variability is superimposed. This model

dominates statistical analysis, shapes how future projections are developed and preferences the underlying theory that supports this model, even though viable alternatives exist (Jones et al., 2013;Franzke, 2014;Branstator and Selten, 2009). Radiative-convective theory that leads to direct warming of the atmosphere underpins this model (Ramanswamy et al., 1991;Hansen et al., 1997;Manabe, 2019), although additional heat from the ocean is needed to close the energy balance relationship (Raper et al., 2001;Meehl et al., 2005;Trenberth et al., 2016;Hansen et al., 2005).

The alternative position is that change and variability interact, producing a nonlinear response on decadal timescales. How this could manifest has been the subject of much speculation, given the many possible outcomes. Using severe testing that assessed shifts and trends as rival processes we showed that warming is not trend-like as in the signal-to-noise model but proceeds through a progression of shifts that sometimes coincide with decadal regime changes (Jones and Ricketts, 2017).

Severe testing uses statistical tests with probative criteria: a hypothesis needs to pass tests that its rivals fail with a high

probability (Mayo, 2005;Mayo and Spanos, 2010). Six tests were developed to determine whether radiatively-forced warming over decadal timescales was dominated by shifts (fast warming) interspersed with periods of relative stability, or by gradual warming modified by variability. Step changes in mean temperature were detected using the multi-step Maronna-Yohai (1978) bivariate test, which calculates a stable set of the minimum number of steps in a times series meeting predetermined detection criteria, usually p<0.01 (Ricketts and Jones, 2016;Ricketts, 2015;Ricketts, 2019). In these tests,

observed and modelled surface temperatures were analysed as monotonic changes, pure steps with no trends, and timeseries divided into fast and slow components: internal trends were measured between change points and the gap between the previous and next internal trend across a change point were measured as shifts. Shift and trend ratios – the ratio of fast and slow warming as a proportion of total warming were also estimated. The probative criteria focused on whether the steplike or trend-like component better explained forcing.

These tests showed (Jones and Ricketts, 2017):

1.  Historical warming was dominated by shifts coinciding with decadal regime changes; regional shift/total warming ratios in some regions were up to 100% and the ratio for global mean surface temperature (GMST) from five records was >50%;

2.  The ensemble pattern of shifts of GMST in 105 CMIP5 GCMs driven by historical forcing reproduced the observed
pattern of regional to global shifts (p<0.01);

3.  Shifts have almost three times (2.9) the explanatory power of trends in describing the relationship between RCP4.5-forced warming 2006–2095 and equilibrium climate sensitivity (ECS, n=93);



4.  Temperature in three regions in Australia, US and UK is mostly stationary until the second half of the 20$^{th}$ century – a relationship also reproduced by GCMs for the Australian region;

5.  A range of other variables, including temperature extremes, fire weather, rainfall, and local sea level rise show compatible shifts and timing; and

6.  Gradual change failed heteroscedasticity tests that step-like change passed in both observations and model output.

If there was an underlying gradual component in the data, these tests would have detected it but didn't. We concluded that climate change and variability combine in a single process that can be described as enhanced climate variability (Jones and Ricketts, 2017). This is consistent with Corti et al. (1999), who proposed that if climate were to produce a nonlinear response to forcing "it would imply that anthropogenically forced changes in climate would project primarily onto the principal patterns of natural variability, even though such natural variability may occur predominantly on timescales much shorter than that of the imposed forcing." Finding the underlying process driving this became our main focus. At the time we described warming as a storage and release process where heat was stored in the shallow ocean and released into the atmosphere (see also Reid, 2016; Reid et al., 2016). We had nominated the Indo-Pacific warm pool as a potential source but had no specific trigger mechanism in mind.

When subsequently, step changes in sea surface temperatures (SST) in the western Pacific warm pool and the central-eastern Pacific cold-tongue were analysed, they preceded those elsewhere. Here those shifts are linked to the wider system of climate oscillations, and shifts in regional and global surface temperatures, to show how the Pacific Ocean heat engine acts as the principal regulator and heat engine for global climate. The difference between this and the standard model in terms of energy flow is tiny, but the consequences are profound. Instead of roughly 1% of the top of the atmosphere energy deficit of 0.71 W m$^{-2}$ remaining in the atmosphere, all available heat trapped by anthropogenic greenhouse gas forcing not taken up by the ground or permanently melting snowpack and ice, is absorbed by the ocean. This additional heat is absorbed into the same processes of dissipation associated with climate variability. How these processes respond is the subject of this paper.

The paper is presented in two parts. The first part introduces the heat engine and describe its behaviour. A tracking model identifying shifts in sea surface temperatures in tropical Pacific is linked to shifts in the broader climate system. We then present a series of analyses describing interactions with climate oscillations, feedbacks and the basic representation of the heat engine in a climate model ensemble.

The second part presents a conceptual model of the heat engine and how it interacts with broader climate. This is used to interpret the results presented in the first part. It summarises the difference between the linear stochastic and deterministic nonlinear dynamic theories, contending that the latter can explain what the former cannot. We argue that the current radiative-convective and linear stochastic justifications for gradual warming are physically implausible and offer an alternative model building on Lorenz' attractors, thermodynamic limits and complex system networks. The result is a self-regulating process of nonlinear change on decadal timescales that leaves the long-term magnitude of projected warming unchanged but follows quite a different process.



## 2   Pacific Ocean heat engine structure

As the largest single source of heat from the tropical regions the western Pacific or Indo-Pacific Warm Pool has been called global climate's heat engine (De Deckker, 2016;Cravatte et al., 2009;Kjellsson et al., 2014;Gagan et al., 2004;Ma and Yu, 2014). It covers the area of the western Pacific and eastern Indian Ocean with sea surface temperatures (SSTs) above 28.0 °C

or 28.5 °C. A region of enormous evaporation and convection, the warm pool transports latent heat into the atmosphere, fuelling the East Asian monsoon and linking to other monsoonal systems (Pierrehumbert, 2000). The resulting rainfall feeds a body of fresher, warmer and less dense seawater that sits on a thermocline up to 200 m deep (Figure 1).

The cold tongue comprised of upwelling cooler waters in the eastern and central Pacific, acts as the collection area for the heat engine. Winds and currents move heat being absorbed into the shallow ocean across over 15,000 km of the tropical

Pacific east to west. This is slightly 'uphill' in terms of temperature and sea level, with a 6–8 °C difference from the far east to the west and 400–500 mm in sea level (Wyrtki, 1975;Bigg, 1990;Harries et al., 1983). This process is bolstered by strong internal feedbacks (Dijkstra and Neelin, 1995) keeping SST in the central Pacific relatively constant. The thermocline, a dense boundary separating shallow, lighter warm water from deeper, denser cool water forms a wedge, shallow in the east and deeper in the west. In normal, steady-state conditions the westward movement of wind, water and warmth keeps the

whole system stable. The surface winds that blow east to west return east at altitude and subside, forming the Walker Circulation (Figure 1)(Gill, 1980;Julian and Chervin, 1978;Bjerknes, 1969).

During an El Niño event, the winds switch east and the thermocline tilts the same way, collapsing the warm pool, heating the eastern Pacific and the atmosphere. During a La Niña event, upwelling cool waters are blown west as the thermocline shallows in the east and conditions are generally cooler. This heat engine is the reverse of the classical Carnot heat engine

where heat moves from the hot to cold reservoir (Kleidon, 2016). A corollary of the second law of thermodynamics states that energy can only flow from cold to hot components of a system unless external work is being done (Kleidon, 2016). This distinguishes the heat engines in the tropical Pacific and Atlantic from most others. Here heat flows from the cold to the hot reservoir, being channelled and made available for dissipation. The external work is provided by incoming radiation being converted into kinetic energy, directed by trade winds produced by the Coriolis effect and by internal positive feedback

effects (Neelin and Dijkstra, 1995;Sverdrup, 1947). Interannual to decadal and possibly longer oscillations are part of the Pacific heat engine, influencing its behaviour. These states reflect the basic structure of ENSO, which responds to forcing on timescales from interannual to millennia (Cane and Clement, 1999;Clement and Cane, 1999;Pierrehumbert, 2000).

Even if global climate is in equilibrium with incoming radiation from the sun (i.e., incoming and outgoing radiation is in balance at the top of the atmosphere), internally, it never is (Kleidon, 2009;Trenberth and Hurrell, 1994;Leith, 1975;Kleidon,

2012). The tropics are a region of gain and the high latitudes a region of loss, so heat moves from the equator to the poles on a continual basis. Heat engines within the climate system are involved in dissipating this energy. They can be stable, periodic or non-periodic. Two major stable heat engines on each side of the central Pacific are the north and south Hadley Cells.





**Figure 1. Cross section of heat engine showing the Walker circulation, warm pool and cold tongue (upper) along with average sea surface temperatures from Dec 2012 with TWP, TEP and Niño 3.4 areas delineated (lower).**

These lift moist, tropical air that rains out and descends in the mid latitudes as dry air (Pierrehumbert, 2000). Subsurface heat is also transported away from the tropics in stable ocean eddies.



Internally generated climate variability is largely a product of oscillations that couple the shallow ocean and atmosphere,
such as the El Niño-Southern Oscillation (ENSO) and Pacific Decadal Oscillation (PDO; Mantua and Hare, 2002),
producing climate regimes. These regimes switch between different modes, maximising the potential to transport heat from
warm to cold regions in the most efficient manner. Efficiency here is a loaded concept because energy in climate can be
transported through radiation, conduction and convection, each requiring different levels of 'work' (Kleidon 2016; even
radiative exchange that does no kinetic work (Delgado-Bonal, 2017)). Distinguishing free from forced behaviour is therefore
very difficult (Kleidon, 2016;Lorenz, 1979).

## 3 Pacific Ocean heat engine behaviour

The main roles of the tropical Pacific Ocean in a changing climate are understood as (Pierrehumbert, 2000;De Deckker,
2016;Gagan et al., 2004;Clement and Cane, 1999):

(i) its role as a source of large-scale convection, distributing latent heat and providing uplift feeding into
meridional heat distribution (i.e., away from the equator),

(ii) its role in ENSO, distributing heat in El Niño and cooling in La Niña events,

(iii) exhibiting decadal variability through the PDO and linked mechanisms including its modulation of the ENSO
cycle, and

(iv) warming and expanding under global warming (Cravatte et al., 2009;Weller et al., 2016), while also being
modulated by decadal variability.

In discussing changes on millennial and orbital timescales, Cane (1998) suggests that the physical links in the chain of
effects extending from the Pacific Ocean to global climate are not restricted to ENSO periodicity and should hold over
longer timescales (Vecchi et al., 2008). However, despite the acknowledgement of its importance in the global energy cycle,
there is no consistent picture of how the tropical Pacific heat engine influences current and future climate. Assessments have
mainly concentrated on the east-west gradient as a diagnostic of El Niño and La Niña-like tendencies (Vecchi et al.,
2008;Kjellsson et al., 2014;Kim and An, 2011), or the warm pool as a source of heat (Chen et al., 2004;Rasmusson and Hall,
1983).

The cold tongue and warm pool form the cold and hot reservoirs of the heat engine. These are delineated following Peyser et
al. (2016) according to the first Empirical Orthogonal Function (EOF) of AVISO satellite sea level data 2003–15: is the
average temperature of the western Pacific (TWP, 120° W–180° W and 20° S–20° N) and the temperature of the eastern
(central) Pacific (TEP, 100° E–160° E and 20° S–20° N). TEP is almost fully superimposed over the Niño 3.4 area (Fig. 1).
Temperatures from NCDCv4 (incorporating ERSSTv4) and ERSSTv5 were the main data sets investigated (see methods and
data in SI). Step changes in time series using the bivariate test were analysed for TEP, TWP and twelve global, hemispheric
and zonal (30°) average temperatures for land-ocean, land and ocean. TWP and TEP were measured from ERSTTv5 because
of its higher resolution, an advantage in the western Pacific with its many islands. Also assessed were shifts in indices for the



Pacific Decadal Oscillation (PDO), the Atlantic Meridional Oscillation (AMO) and Atlantic Meridional Overturning Circulation (AMOC) Index. All measured shifts are p<0.01 unless otherwise stated. Details of data sets, test and results are listed in the SI, in addition to extensive post-processing, testing for false positives.

GMST shows step changes in 1937 of 0.22 °C, 1977 0.19 °C, 1987 0.14 °C, 1997 0.23 °C and 2014 0.27 °C (Fig 2a). TWP contains step changes in 1902 of -0.34 °C, 1921 0.17 °C, 1942 0.21 °C, 1978 0.17 °C, 1995 0.28°C and 2013 0.28 °C (Fig. 2b). Step changes in TWP broadly correspond with step changes in GMST since about 1940 and the others, the decrease in 1902 TWP is reflected in the oceans, 1921 NH land and 1987 NH (Table S8). TEP contains one confirmed step change in 1977 of 0.46 °C (Fig. 2d), and potentially a second in 2014. This shows TEP as largely being stable and TWP as responsive. The step change in TWP in 1942 is probably related to the 1937 shift as recorded in NH and global mean surface temperature. Adjustments made to pre-WWII and WWII data in ERSSTv4 and 5 have moved this date later than in earlier versions but 1937 remains stable in the NH and global record. WWII SST data is also biased warm in some regions (Liu et al., 2015), affecting the southern tropics and southern hemisphere and warm pool.

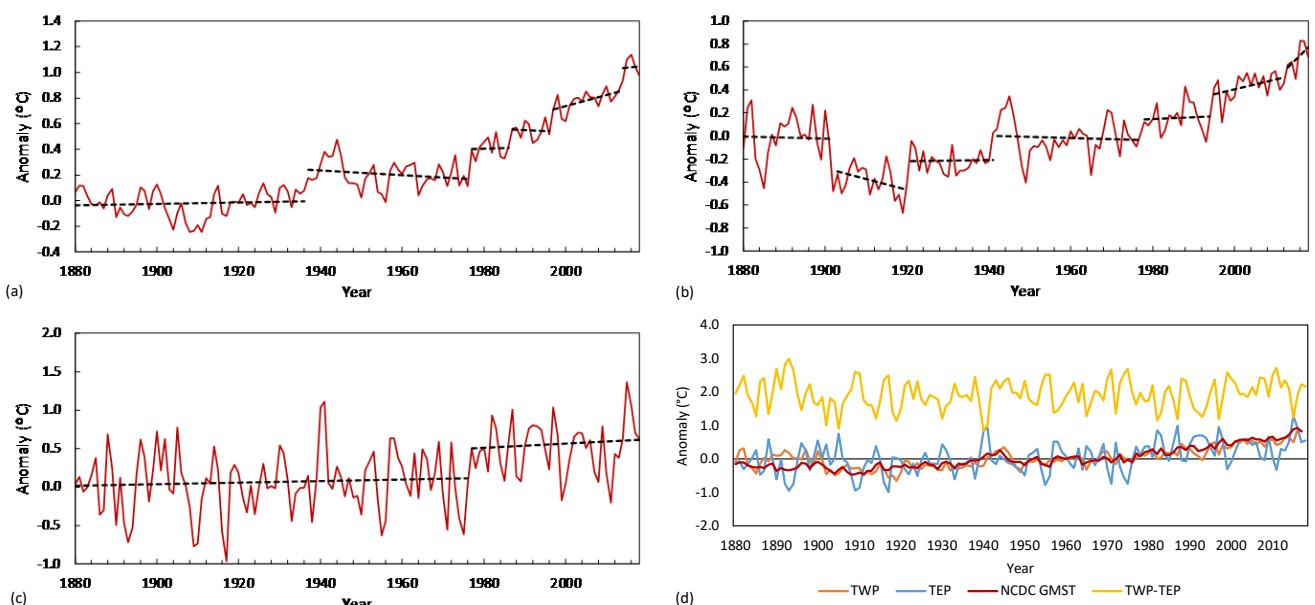

**Figure 2(a) GMST (NCDC) 1880–2018 analysed using the bivariate test for shifts in mean (p<0.01); (b) TWP (ERSSTv5) as for (a); (c) TEP (ERSSTv5) as for (a); (d) TWP, TEP and GMST (NCDC) shown with the TWP-TEP temperature gradient over the same period.**

The gradient between TWP and TEP over the period 1880–2018 was 1.94 °C, TWP averaging 28.20±0.31 °C and TEP 26.26±0.44 °C (Fig. 2d). This gradient of almost 2 °C has remained relatively constant over the historical period, but is mediated by the PDO (Sect. 4.1). The gradient between the hot and cold reservoirs is a key diagnostic of the heat engine and its constancy, a sign of its self-regulating behaviour, as discussed later. The overall east-west gradient across the Pacific is a





critical influence on ENSO behaviour (Kim and An, 2011). Satellite measured sea level 1993–2018 between the Indonesian region and ENSO3.4 region also remains relatively constant over time.

Warming over the historical period from a baseline of 1880–1899 was 1.05 °C for GMST, 0.77 °C for TWP and 0.62 °C for
TEP. The lower warming from TWP and TEP is due to the cooling in 1902–03, abrupt in TWP. The TWP closely tracks GMST from the post-1902 low until 1998, when the 1997–98 shift in GMST accelerated past the TWP. The correspondence between Pacific temperature and GSMT has been noted (Cane, 1998;Pierrehumbert, 2000), Pierrehumbert (2000) recognising the inherent tippiness of the tropical Pacific.

### 3.1 Tracking shifts

The timing of shifts in TWP and TEP preceding those elsewhere suggested an origin in the tropical Pacific. Deseasonalised monthly data was analysed to pinpoint the timing for those regions where steps had previously been detected at $p<0.01$ or $p<0.05$ (a less stringent criterion has been used for detecting steps in the most recent decade because of the limited time elapsed). Testing located a few false negatives in steps detected using annual data, where a single step masked two either side, or steps where annual data was displaced by a few years due to interannual variability. These are discussed in the SI.

A tracking model testing for sustained, warm anomalies in TWP was developed. TEP did not show the same recharge-discharge behaviour. Anomalies were calculated from the cumulative mean of the current regime since the last known shift. The model began in Jan 1947 to avoid the poorer data quality of the WWII period (See Folland et al., 2018). Each month the cumulative mean was updated and a six-month running mean of the difference from that cumulative mean calculated. Sustained six-month anomalies $>0.2$ °C from the running mean were associated with regime shifts within TWP and/or more
widely.

Fig. 3a shows monthly TWP and TEP with the cumulative mean of successive regimes 1960–2018. A regime shift takes some time to register statistically at $p<0.05$ or $p<0.01$ – from two to fifteen years depending on the size of the shift and the background variance. In Fig. 3, a pre-detection period of 48 months is shown as a dotted line. For TWP, four shifts are nominated, including 1968 at $p<0.05$. Two shifts for TEP are shown, 1976 and 2014. The latter had not reached the $p=0.05$
threshold by 2018 (currently $p<0.2$) but its size at 0.46 °C is the same as 1977.

Fig. 3b shows the six-month TWP anomalies of 0.2 °C. Two added events not associated with regime changes shown in Fig. 3a are labelled: the 1987–88 shift in the NH and 2010 shift in the SH. The only outlier not directly associated with widespread regime change was in 2001, this date shows up in a few annual shifts, in the tropics and high northern latitudes, most of which do not carry through to monthly data (Table S8).

Shifts after 1947 were divided into six periods: 1968–69 (SH, Fig. 3c), 1976–79 (global, Fig. 3d), 1986–88 (NH), 1995–98 (global), late 2000s (distributed), 2013–15 (global). All are shown in Figs 3c–g except the late 2000s event.





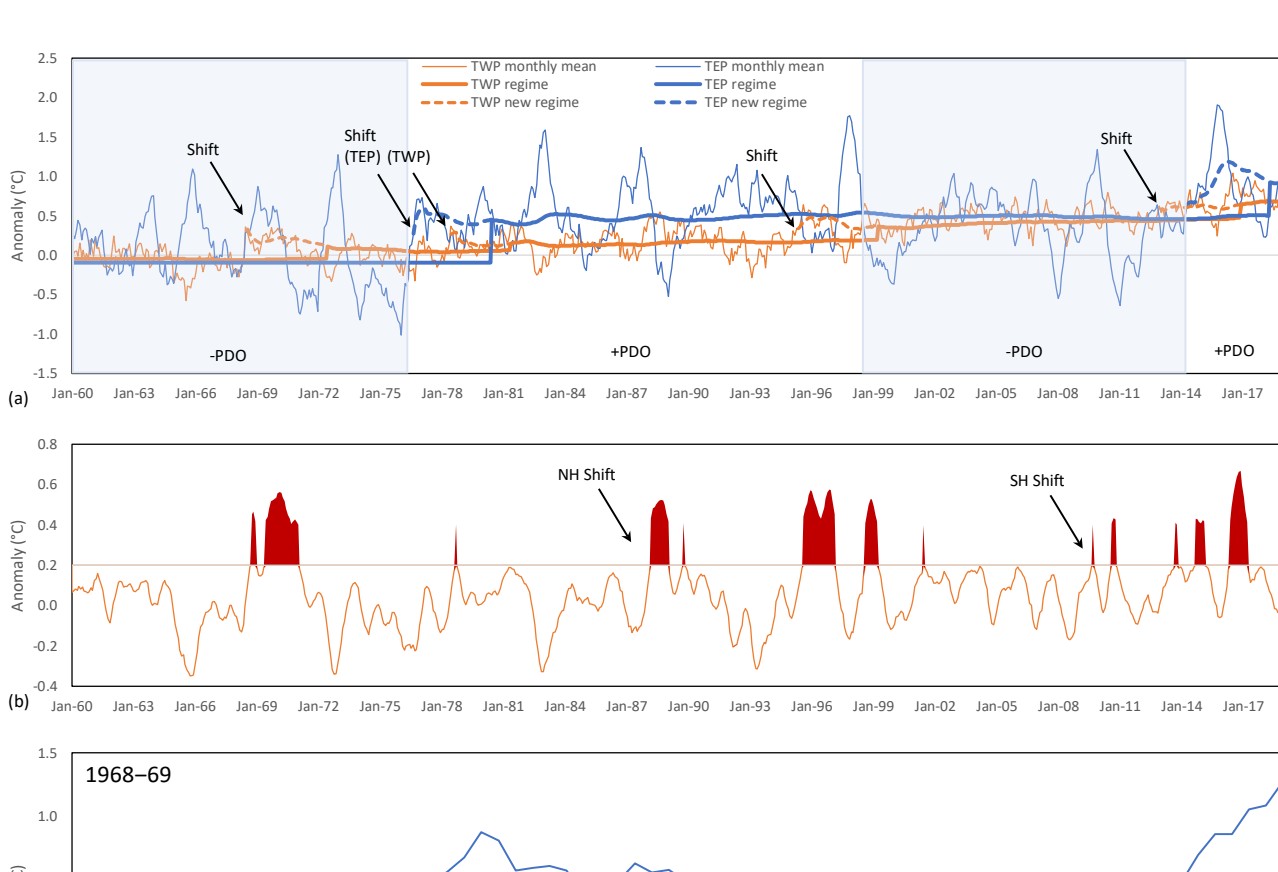

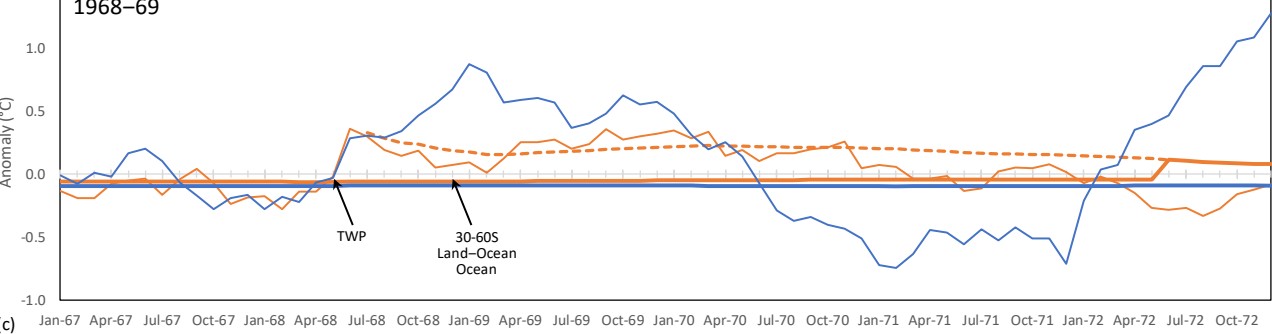


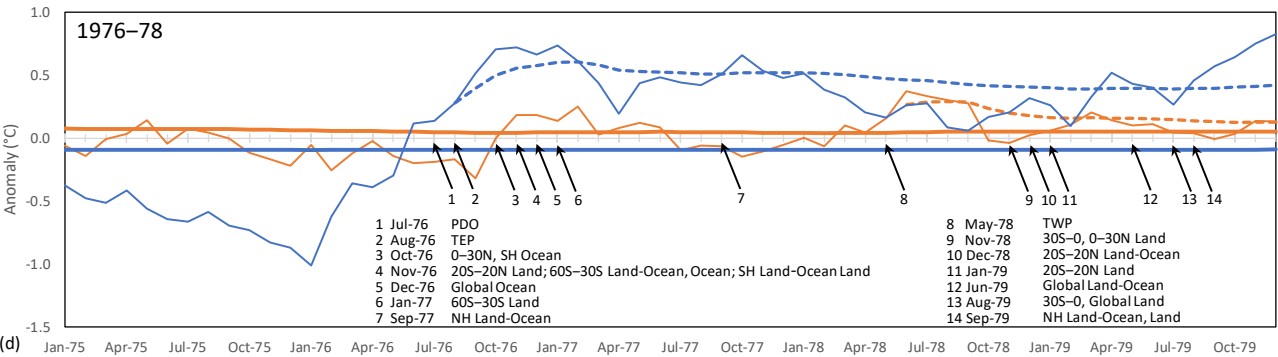



**Figure 3. Tracking model for the heat engine showing (a) monthly anomalies for TWP and TEP from ERSSTv5 and cumulative means for current regime (pre-detection dashed) noting the timing of the shifts detected for four regime changes in since 1947 along with PDO phases (light blue shading), (b) six-month retrospective mean showing 0.2 °C exceedances from the cumulative mean (red shading) and the two shifts that not associated with shifts in TWP; sequences of shifts 1968–69 (c), 1976–78 (d), 1986–88 (e), 1995–98 (f) and 2012–15 (g), legend as in (a).**

Fig. 3c shows the shift in TWP occurred May 1968 followed by 60° S–30° S ocean and land-ocean in Dec 68, with a sustained anomaly in TWP to Sep 1971. These shifts affected Australian SST and land temperatures in southern Australia, and were coincident with a downward shift in winter rainfall in SW western Australia (Jones and Ricketts, 2019a). This heat





transferred into the TEP in an El Niño event in 1972–73 leading to an increase in northern Australian rainfall of 19% and
warming along the east coast of Australia (Jones and Ricketts, 2019a).

The 1976–78 event was initiated with a shift in the PDO in Jul 1976 and TEP in Aug 1976 (Fig. 3d). The following sequence
shows ocean then land-based warming in the SH in 1976–77, followed by NH warming in 1978–79. The SH shifts were
prompted by the TEP. The TWP responded in May 1978, followed by more general warming in the NH tropics, finally
registering as a global phenomenon in mid 1979.

The origin of the 1986–88 event was difficult to locate, showing up first in the tropical ocean 0–30° N in Oct 1986 (p<0.10
annually) and in the global oceans in Dec 1986 before propagating through the NH during 1987–88 (Fig. 3e). This was
possibly due to a large-scale change in circulation. After the initial ocean shift, the regime shift appears to be largely
atmospheric and over land, its main impacts felt in western Europe and parts of north America, coinciding with the 1988
drought. Its impacts were comprehensively documented by Reid et al. (2016).

The 1995–98 event began with a shift in the AMO in Mar 1995 and TWP in Apr (Fig. 2f). Based on timing, this suggests a
teleconnection between the AMO and TWP. Warming progressed to the SH mid latitude ocean by Feb 1996 and land by
Aug 1996. In early 1997, NH land shifted warmer under the influence of an El Niño as shown by warmer conditions in TEP.
This was followed by warming further north during 1988. The PDO shifted to its negative phase in Jun 1998 with the
inception of La Niña conditions.

A brief 0.2 °C anomaly in the warm pool in Oct 2009 and a longer one during the next year coincided with land-ocean and
ocean 60° S–30 °S in Dec 2009. This is associated with shifts in Australian climate (Jones and Ricketts, 2019a). It may be
similar to 1968, where a shift into the SH mid latitudes affecting Australia preceded the large 1976/77 shift.

The most recent event began with a shift in TWP in Dec 2012. The 0.2 °C threshold was subsequently exceeded Sep–Oct
2013, Sep 2014–Feb 15 and May 2016–Mar 17. Shifts occurred in mid to high NH latitudes in mid to late 2013, at the
hemispheric and global scale for ocean and land-ocean in the first half of 2014 and land in the second half. The PDO shifted
to positive in Mar 2014 and precedes a potential shift in TEP by a month. In mid-2014, the PDO shifted with a large El Niño
cascading through the tropics and southern hemisphere, with shifts in NH and global land registering in Dec 2014.

The increase in temperature 2014–2018 over 1997–2013 in the NCDC record was 0.26 °C globally, 0.17 °C in the SH and
0.34°C in the NH. Allowing for spatial differences, just over half the warming occurred in SST and two-thirds in the NH. In
NH latitudinal zones, 00°–30° N and 30° S–60° N rose by 0.31 °C and 60° S–90° N by 0.66 °C. Land temperatures in
Europe increased by 0.59 °C.

### 3.2  Early 20th century shifts

The timing of pre-WWII shifts is less reliable because of issues with data quality and coverage. The SH ocean and land-
ocean shifted cooler in mid-1901. The AMO index shifted negative in April 1902 and downward shift in TWP occurred in
May 1902, the same month the Mt Pelée and Soufrière eruptions affected the troposphere. In Oct 1902 the Santa Maria
eruption reached the stratosphere (Sato et al., 1993). The 1902 shift of -0.34 °C in TWP took until 1937 to recover fully.


Positive shifts in late 1911 in SH ocean and in late 1914 in the NH ocean may have been a partial recovery from this cool period. The high northern latitudes shifted positive in mid-1919 and TWP responded in Nov 1920, global land in Dec 1920 and NH land in Jan 1921. This only positive shift in TWP not associated with an El Niño, occurring just before a strong La

Niña event in 1921 (Wolter and Timlin, 2011). A series of shifts from 1924–25 do not show any particular spatial co-ordination and are mainly tropical and NH, culminating in regime shifts in the PDO in Sep 1925 and AMO in Dec 1925. The bivariate test run on 5° x 5° gridded data shows shifts forming the horseshoe pattern of the AMO signature in the northern Atlantic (Ricketts, 2019).

The NH shifts 1920–25 may have been partially driven by increasing solar radiation, which set an early peak around 1920

(Hegerl et al., 2018;Egorova et al., 2018;Wang et al., 2018). Analysing the different forcings of the early 20th century, Meehl et al. (2003) suggested that much of the warming (from the 1920s to 1940s) was due to solar forcing, that it was greatest in the areas where that radiation increases were experienced and that the land-ocean warming ratio was 2.6–2.7 compared to 1.3–1.5 later in the century. This supports the idea that the Pacific Ocean warming was responding to warming largely concentrated in the NH.

The 1937 shift was recorded in tropical and SH oceans but was also large enough to register in global SST. Australian mean annual SST shifted upwards (1938) but land temperatures were unaffected. The Indian subcontinent and low-latitude South America may also have shifted upwards, based on the assessment of 45° longitude, 30° latitude zonal segments (not shown). Problematic SSTs in the tropics and SH just before and during WWII make it difficult to pinpoint exact times and locations.

## 4   Decadal variability

Most phase changes in both the PDO and AMO have coincided with shifts in temperature, some leading and others lagging. This led to a further exploration of the timing of decadal regime shifts and their relationship with the heat engine and climate shifts more broadly.

### 4.1  Pacific Decadal Oscillation

Although PDO is mainly associated with the northern Pacific (Mantua and Hare, 2002;Newman et al., 2003), Newman et al.

(2016) separated the PDO into tropical and north Pacific components. Here we deal with the PDO index via its interaction with the tropical heat engine, and its influence on how heat is collected and dissipated (details on data used in the SI). Our results suggest that oscillatory impacts in the tropical Pacific have flow-on effects in the N and S Pacific related to the northern PDO and Interdecadal Pacific Oscillation (IPO)(Power et al., 1999;Henley et al., 2015), which combines all three.

Three PDO records beginning in 1880, 1890 and 1911 were analysed for shifts between known phases. The period 1896–

1925 is interpreted as neutral (averaging -0.06) but behaved mostly positively (Henley et al., 2015). The phase changes between periods in Table 1 register as shifts with p values deceasing over time (Table S5). This is consistent with tighter coupling of the PDO within the heat engine under forcing. Phase lengths show two sets of 29 and 22 years, followed by a





negative phase of 15 years 1999–2013 (Table 1), coinciding with the so-called hiatus. Shorter phase lengths and a stronger signal may also be a response to forcing, as analysed for observations and the 21st century (Bonfils and Santer, 2011;Dong et al., 2014).


The TWP-TEP temperature gradient of 1.94 °C is ≥2 °C for negative phases (La Niña-dominated) and <1.9 °C for positive phases (El Niño-dominated) and is shown in Fig. 4a with the PDO Index; the two are inversely correlated (-0.62, p=4.8 x 10⁻¹⁶). These differences in gradient are mainly due to variations in TEP, which cooled slightly each for each phase shift from positive to negative, and warmed when PDO shifted negative to positive. For the last two negative-to-positive phase


shifts, TEP stepped up by ~0.46 °C, compared to the earlier changes in 1896 and 1925 of 0.13 °C and 0.12 °C, respectively. The most straightforward explanation is that the temperature responses to earlier changes reflect internal variability and the latter two are also responding to forcing. Meehl et al. (2009) analysed the contribution of natural and anthropogenic forcing to the 1976 shift, comparing it in climate model output produced by baseline and 20th century forcing. They addressed it through a 15-year filter of the IPO, so looked at shifts that had been partially smoothed. In comparing a shift in 1960s with


one in the 1970s, they estimated that most of the 1970s shift was forced. A subsequent shift to a negative IPO in the same model (late 1990s), they interpreted as unforced. This is consistent with the behaviour of the heat engine outlined above. The 1976 shift was associated with a disruption of the thermocline through a subsurface process initiated in the south-east Pacific a number of years earlier (O'Kane et al., 2014). This implies that large shifts in the relatively stable cold tongue could be predicted some time in advance. Whether the speed of such changes could be influenced by forcing is an interesting


question.

**Table 1: PDO phase, temperature anomalies in TEP and TWP, period, difference between TEP and TWP, change in difference and length of phase.**

| PDO phase | TEP anom (°C) | TWP anom (°C) | Period | TEP-TWP (°C) | δTEP-TWP (°C) | Length (y) |
|---|---|---|---|---|---|---|
| Negative | -0.08 | 0.00 | 1880–1895 | 2.19 | | 16+ |
| Neutral | 0.05 | -0.26 | 1896–1924 | 1.81 | 0.37 | 29 |
| Positive | 0.17 | -0.13 | 1925–1946 | 1.84 | -0.02 | 22 |
| Negative | 0.08 | -0.05 | 1947–1975 | 1.99 | -0.15 | 29 |
| Positive | 0.57 | 0.17 | 1976–1998 | 1.77 | 0.22 | 22 |
| Negative | 0.39 | 0.47 | 1999–2013 | 2.23 | -0.46 | 15 |
| Positive | 0.92 | 0.70 | 2014–2018 | 1.93 | 0.30 | 5+ |

For TWP, apart from the temperature decrease in the early 20th century, which was not associated with the PDO, all


successive phases have been warmer (Table 1). All phase changes in the PDO since the 1970s have been associated with increases in TWP of 0.2 °C to 0.3 °C (in 1978, 1997 and 2014).  Shifts in the NH in 1925–26 and globally in 1976–77 and 2013–14 coincided with shifts in PDO from negative to positive phase. Most warming since 1880 has taken place during positive phases, consistent with the standard explanation based on trend analysis (Folland et al., 2018). Since 1880, PDO was





positive 30% of the time for 76% of total warming, negative for 43% of the time for 28% of total warming and the neutral
phase saw a 4% decrease. Most of this warming has occurred as shifts (93%) compared to trends.

During its negative phase, the cool tongue as represented by TEP is cooler, the intake area wider, advection is enhanced and
the gradient ≥2 °C, when positive, TEP is warmer, the intake area narrower, convection enhanced and gradient <2 °C. The
reduced TWP-TEP gradient during positive phases is because TEP is slightly warmer, the area channelling heat into the
warm pool is narrower, meridional transport is reduced and the warm pool receives more heat. In negative PDOs TEP is
cooler, the TWP-TEP gradient is larger, the distributive area wider, and more heat is moved outwards and downwards and
less into the warm pool (Zhang et al., 2009;Allen and Amaya, 2018).

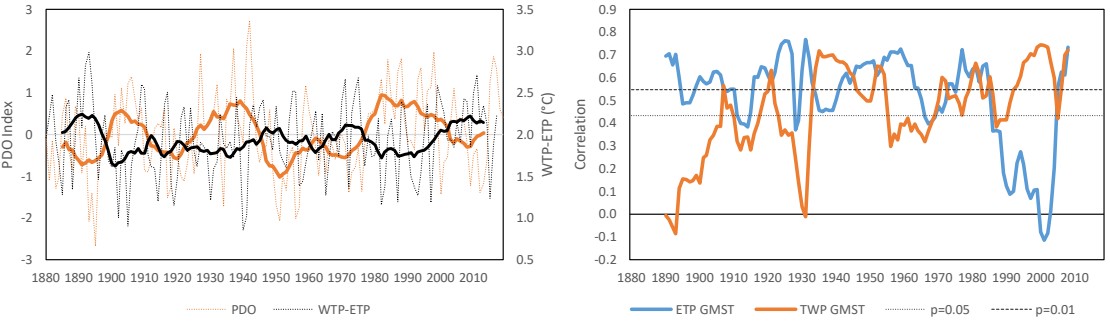

**Figure 4(a) Superposition of PDO Index and TWP-TEP difference shown with 11-year running means and (b) the 21-year running
correlation between TEP GMST and TWP GMST with p=0.05 and p=0.01 thresholds.**

## 4.2 Atlantic Meridional Oscillation

Most shifts in AMO phases have followed shifts in temperature except for 1963, which is not linked to any other shifts and
1995, which precedes the shift in TWP by one month, starting the 1995–98 sequence. All shifts in the AMO are p<0.01
(Table S6) indicating that the strength of phase changes in the AMO is not being influenced by increasing forcing.

The AMO shifted negative in Apr 1902 followed by TWP in May but was preceded by the SH and mid latitude NH ocean.
The shift to negative phases in the AMO have often been associated with volcanic eruptions (Birkel et al., 2018;Swingedouw
et al., 2017) but this earlier cooling preceded the 1902 eruptions. The shift in Dec 1925 was preceded by shifts in SST that
may have originated in the high northern latitudes, but were certainly oceanic. The negative shift in Aug 1963 followed the
Agung eruption in March 1963, supporting the volcanic hypothesis.

The 1995 shift in the AMO has been documented as a regime shift (Gray et al., 2003;Enfield and Cid-Serrano, 2006;Alheit
et al., 2014;McCabe et al., 2004). It preceded the shift in the warm pool and heads up a sequence of events that affect the
mid latitude SH in 1996 and more widely in 1997. Investigating AMO–PDO linkages, Zhang and Delworth (2007) suggested
the northern Pacific may shift a decade after 1995, but the PDO shifted in June 1998. The AMO is currently in a positive
phase lasting 23 years. McCarthy et al. (2015) suggest that ocean circulation and sea level is an early precursor and that a
negative transition may be underway.



Some analysts have suggested networked behaviour between various oscillations, including the AMO and PDO, where they couple during regime shifts (Swanson and Tsonis, 2009;Tsonis and Swanson, 2012;Wang et al., 2009), although this is interpreted as a change in the rate of trend (Tsonis, 2018). Teleconnections can also be seen in alternating lead and lag behaviour between the AMO and PDO (Wu et al., 2011;Zhang and Delworth, 2007).

### 4.3 Atlantic Meridional Overturning Circulation

The Atlantic meridional overturning circulation (AMOC) Index 1871–2016 constructed from HadISST by Caesar et al. (2018) was also analysed, shifting down in 1969 by -0.60, shifting up by 0.38 in 1997 and down by -0.74 in 2014, the latter mirroring phase shifts in the PDO (all p<0.01). The recent slowing of the AMOC has been attributed by some to forcing (Rahmstorf et al., 2015;Caesar et al., 2018). Dima and Lohmann (2010) identified two modes in Atlantic deep-water formation, a slow mode decreasing gradually since the late 1930s and a fast mode shifting negative and positive around 1970
and in the late 1990s. The 1969 shift was coincident with a shift in TWP and was followed by shifts coinciding with warming events, reversing in phase with the PDO. There are two camps as to whether AMOC responds to internal and external forcing in a regime-like way, or smoothly (Wunsch and Heimbach, 2013).

Two similar SST indices have been used to trace AMOC back to just before 1000 AD, showing increases from 1500 to 1800, a slight decrease in the 19$^{th}$ century, before those noted above (Caesar et al., 2018;Rahmstorf et al., 2015). Direct estimates
of AMOC flow volumes are only available from 2004, so whether they reflect the index precisely or follow a more general pattern remains to be determined. If it is regime-like, the timing suggests teleconnections to the heat engine and the PDO.

Examples of regime-like behaviour include Thompson et al. (2010), who report on sudden drop in NH temperatures centred on the area of deep water formation in the north Atlantic around 1970 that appears to mirror warming in the SH. Testing abrupt shifts in correlation between SST 30° N–60° N and GMST shows a reversal from positive to negative 1969–1988,
when the upward shift in the SH occurred. The previous downward shift in that region in 1901 coincided with cooling in the tropics and SH, whereas in the later one they are opposed.

The timing may indicate a shift in dissipation rates from a slowly circulating system in ocean overturning to a rapid one in the form of the Pacific heat engine. Interactions between slow ocean-driven mechanisms and faster atmospheric ones have been noted in a range of studies (Farneti, 2017;Zhang and Delworth, 2007;Wu et al., 2011). This raises the possibility that if
there is a network of oscillating and circulating systems linked spatially, that they are also linked temporally, able to adjust the rate of fast and slow dissipation based on the amount of forcing.

### 5 Broader climate influences

### 5.1 Shifts in autocorrelation

Beaulieu and Killick (2018) identified a change in autocorrelation within historical GMST. When they applied segmented
trend to five GMST records, with a breakpoint in 1962 or 1972 depending on the data set, autocorrelation was high before





the change point and low (whitened) afterwards. We explored this change in autocorrelation using Rodionov's (2015) test that detects abrupt shifts in correlation to see whether it produced a regime-like response.

After removing the steps (segment means) between the shifts shown in Fig. 2c from NCDC GMST, we tested the lag-1 autocorrelation in the residuals for shifts. Autocorrelation shifted abruptly from 0.59 in 1881–1968 to 0.02 in 1969–2018

(p=0.0002). Land temperature autocorrelation shifted in 1972 from 0.35 to white (-0.08, p=0.02). SST shifted from positive to neutral in 1914 and back to positive in 1938 (0.74 to 0.09 to 0.74, p=0.002, 0.01), shifting neutral again in 1957 (0.74 to 0.16, p=0.006), showing the surface ocean whitening during warming periods (Fig. S3). Under sustained forcing, SST shifted white in 1957, land-ocean in 1969 and land in 1972, reflecting an abrupt tightening of the heat engine under forcing, beginning with SST. Autocorrelation is present when the climate is not warming, the ocean being the most sensitive to these

changes but under tighter coupling, interannual variability retains little information of the previous year's climate. This is shown in Fig. 5a, where the cumulative sum of year to year differences within the residuals shows random walks before 1950 and limited memory afterwards (shown by the straight line around zero deviation from the mean).

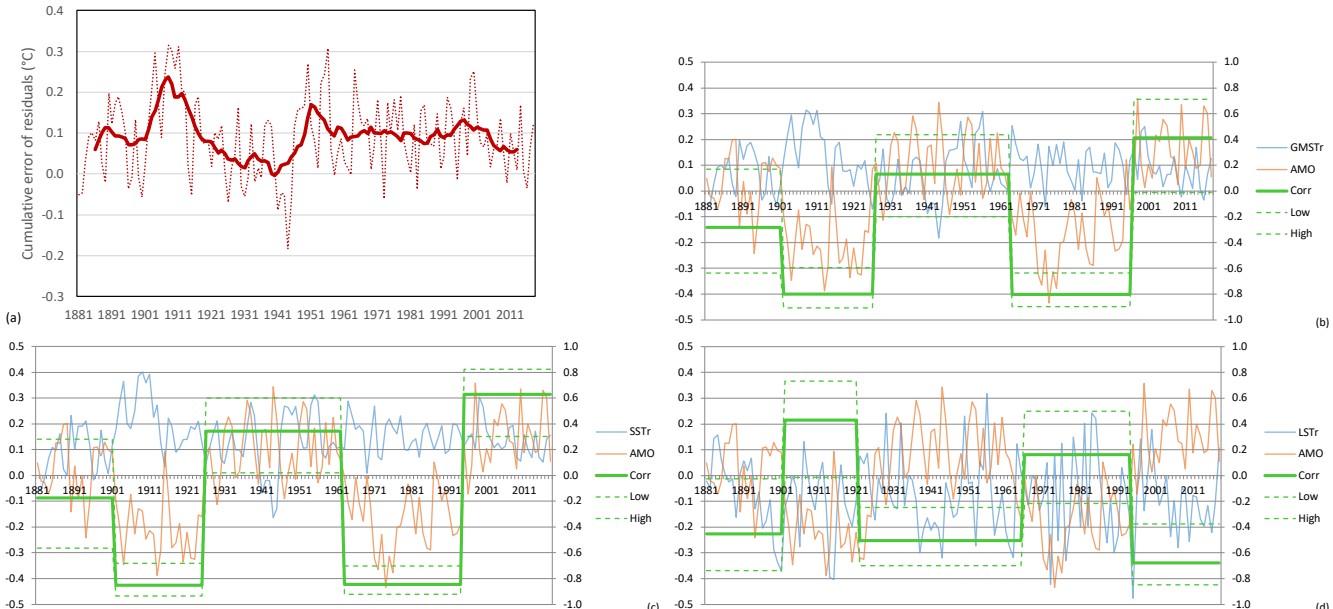

**Figure 5(a) Cumulative lag-1 difference from NCDC GMST 1881–2018 with step changes removed (GMSTr), showing annual data and 11-year means; (b) shifts in correlation between data in (a) and the AMO index (Rodionov (2015) with target p<0.05, cutoff length = 20, tuning constant = 3σ); (c) as for (b) but ocean only (SSTr); (d) as for (b) but land only (LSTr). The left vertical axis is the index/anomaly in °C and the right axis, correlation coefficient.**

The cumulative sum of the residual differences in Figure 5a and the AMO index was tested for shifts in correlation using

Rodionov's test with a p<0.05 threshold. Land-ocean and ocean switched sign almost in unison with AMO phase changes, being anti-correlated during negative phases, but during positive phases moved from no correlation to strong correlation as forcing increased. Land produced a mirror image, inversely correlated with the ocean and land-ocean becoming more anti-correlated during positive phases over time (Figs 5b-d).



Therefore, until the 1950s, a period of limited or no warming, global average temperatures were able to undertake random
walks from steady-state under the influence of the AMO. Even though the influence of the AMO on interannual variability
has increased, its decadal influence, other than switching sign, has ceased. Correlations with the PDO were also tested but its
influence was intermittent. The tightening of the system coincided with the heat engine moving from free to forced mode.

## 5.2  East-west influence of the Pacific on global temperature

TWP represents the warming side of the heat engine and TEP, the cooling side. Fig. 4b shows the 21-year running
correlation between TEP and GMST, and TWP and GMST with p=0.05 and p=0.01 thresholds. When TEP was more highly
correlated with GMST, climate was either cooling slightly or maintaining steady-state conditions. TWP was briefly
associated with warming events early in the record, peaking in 1907, 1921, 1935, 1952 then coincident with a strong El Niño
in 1972. The only sustained period of influence before the 1970s is centred on WWII, which may be an artefact (Sect.
S2.1.2). TEP peaked in 1977, marking the upward shift in temperature and both TEP and TWP were high during the positive
PDO phase 1977–1998. During the negative phase 1999–2013 the influence of TEP on GMST plummeted, becoming
slightly negative before recovering after 2013 as both move into phase with the shift to positive PDO and simultaneous large
El Niño events.

GMST was associated with TEP in the first half of the 20th century and TWP in the second half. TEP may have been
influenced by recent negative PDO phase when briefly becoming negatively correlated with GMST. These changes are
consistent with analyses of early 20th century tropical SST variations being more La Niña-like and later being more El Niño-
like (Liu et al., 2005). The same correlations were carried out for 29 GCMs models forced by RCP2.5 1961–2100 (Fig. S2),
showing they follow similar patterns, with TEP being dominant early and the influence of TWP lower in the early 20th
century, decreasing towards 2100 as forcing levels off.

## 5.3  Representation of the Pacific Ocean heat engine in climate models

TWP and TEP were calculated from an ensemble of 30 RCP4.5 GCMs from CMIP5 (Table S7), with 29 having available
GMST records, allowing two issues to be addressed: (1) whether GCMs contain a Pacific Ocean heat engine and (2) whether
the behaviour of those heat engines can be linked to shifts in GMST.

The most basic diagnostic for the heat engine is the temperature gradient between TEP and TWP. The average difference
between TEP and TWP 1880–2018 in the models was 1.30±0.06 °C, compared to the observed difference of 1.94 °C,
situated at the 90th percentile of the model distribution. The gradient 1861–2100 in the models ranges between 0.53 °C and
2.44 °C and remains relatively constant under forcing, deviating slightly in some models as the rate of forcing declines late
in the 21st century.

The total number of shifts in TEP (n=122) is roughly half that in TWP (n=242). Warming is dominated by shifts rather than
trends, with shift step/ratios averaging 0.70±0.24 in TEP and 0.67±0.14 in TWP. The average step size in TEP is 0.55±0.17
°C and in TWP 0.28±0.14 °C, about 20% higher than observations but showing a similar ratio east to west. Fig. 4b was





reproduced for each of the 29 models with available GMST and shows a variety of behaviours between TWP and TEP with GMST, with some models showing what appears to be pronounced decadal variability (Fig. S2).

Of the 25 GCM records with available estimates of equilibrium climate sensitivity (ECS), the correlation between average TEP step size in each of the models with ECS is 0.70 (p=0.0001) and TWP step size and ECS is 0.39 (p=0.05). The average
TEP step size ECS correlation to 2018 in the models is 0.60 (p=0.015). This correlation is greater than any of the diagnostics comparing historical warming in models to ECS recorded in JR2017, and is due to spatial patterns of atmospheric feedback (see next section). Correlation between average step size in the models and total warming to 2100 for TEP is 0.41 (p=0.04, n=29) and TWP 0.19 (p=0.37, n=29).

An estimate of ECS from observations was calculated from the linear regression between TEP and ECS in the climate model
ensemble. Based on shifts in 1976 and 2014 averaging 0.46 °C, observed ECS is an estimated 3.2±0.6 °C ($r^2$=0.46). A much more sophisticated analysis using spatial representations of sensitivity produced the same estimate, but with a larger range of uncertainty (1.5 °C to 8.1 °C, 5% to 95% confidence) (Andrews et al., 2018).

If the heat engine is influencing GMST, then the timing of shifts in TWP and TWP should show a relationship with those in GMST. For 29 GCMs, shifts in GMST within ±1 year of those in TWP and/or TEP (167 from a total of 352) matched 133
out of 279 global shifts. The probability each matching shift in the heat engine and GMST being random was calculated for each model. This allowed for a time gap of seven years between shifts in each of TEP and TWP (the sampling interval in the multistep bivariate test), and allowed for coinciding shifts TEP and TWP matching a shift in GMST. The average probability of there being no relationship within the models was 4.67 x $10^{-4}$. After being adjusted for the likelihood of there being no matches, the resulting average p was 0.009 with median p being 1.57 x $10^{-8}$ (Table S9). Further work would be needed to
understand the nature of these relationships in more detail.

## 5.4 Atmospheric feedbacks

Shifts in TWP associated with a heat build-up exceeding a temperature anomaly of 0.2 °C in the tracking model implies a self-regulating role in the heat engine (see also Dong et al., 2019). TEP represents a negative feedback at the ocean surface absorbing heat whereas TWP represent a positive feedback, losing heat. These are counterbalanced by a positive atmospheric
feedback in the east and negative feedback in the west. The latter only have a net effect if mean temperature in either region changes.

Spatial variations in atmospheric feedback are strongly positive in areas of subsidence and weak to negative in areas of uplift. Andrews and Webb (2018) changed SST patterns in a set of atmospheric GCM experiments, applying a 4 °C warming to patches in the SE (30° S–0°, 80°–100° W) and W (15° S–15° N, 150° E–170° E). Warming of the SE Pacific patch had a
local effect but high cloud feedback. Warming the W Pacific patch, centred over TWP, heat was transported to the mid-latitudes but cloud feedback was negative (Andrews and Webb, 2018). The correlation between cloud feedback and the west Pacific patch was -0.49 and the SE Pacific patch was 0.93. A SE–W index was more influential over overall feedback than either region alone, but the SE region was more important than the W.



Artificial warming of small patches in the tropics and extra-tropics shows those in areas of subsidence produce a more
positive cloud feedback and local warming (Zhou et al., 2017). Over areas of uplift such as the warm pool, they have a
neutral to negative cloud feedback and widespread warming. For the tropical Pacific, the precise locations of the regions
forced does not matter greatly as long the E–W difference is maintained (Andrews and Webb, 2018;Zhou et al., 2017).

Building on these studies, Dong et al. (2019) forced an AGCM with historical SST, identifying feedback responses
consistent with those above. They also conducted patch experiments, reporting on four ellipses 40° in length and 15° width
near the tropics and 25° high latitudes, over TWP, SE of TEP (S tropic far E Pacific), NW high latitude Pacific and high
latitude Southern Ocean below New Zealand. They measured GMST, top of the atmosphere radiation deficit, estimated
inversion strength and low cloud markers.

The warm pool patch transported heat to the upper troposphere, which spread to the whole troposphere, increasing outgoing
longwave radiation (OLR). Local inversion strength and low cloud decreased above the patch, but globally it increased,
raising albedo. This is the source of the negative cloud feedback mentioned above. The east Pacific and high latitude patches
produced local warming and limited outside change, but mainly in different signs to the warm pool patch. They also assessed
the response from each gridpoint of SST across the global domain, confirming that over the historical period that warming in
the warm pool had contributed to negative feedbacks, mainly due to OLR.

Abrupt 4 x $CO_2$ simulations also show nonlinearity in the first few decades. Andrews et al. (2015) analysed the CMIP4 runs,
where 23 of 27 models showed nonlinearity where feedback increased over time, mainly due to tropical influences and low
cloud albedo. When uniform SSTs were imposed, not allowing the ocean to respond, the outcome was linear. This follows
the historical pattern, where heat is absorbed into the ocean for some decades before warming starts in earnest. The sequence
from free to forced warming with a delay occurs in a coupled model or AGCM with an SST pattern representing a coupled
process, but is absent in AGCMs forced by a uniform SST. This points to the heat engine being an emergent property of the
coupled ocean-atmosphere system.

Dong et al. (2019) combined a 4 x $CO_2$ simulation with their gridpoint technique to show that the warm pool dominated
early and remained important, but as the eastern Pacific and high latitude regions warm, sensitivity increased. They isolated
the warm pool region (about twice the size of TWP) and calculated its effect on the proportion of global radiative feedback
as a function of change in temperature, showing that the warm pool continued to dominate over time. As warming spreads
into regions beyond the warm pool, outgoing radiation changes relatively little but sensitivity rises (Dong et al., 2019).

The relationship between shifts in surface temperature, cloud and outward longwave radiation (OLR) 1979–2016 was
examined by Saltykov et al. (2017) for the zones 30° S–30° N, 70°–30° S and 30°–70° N. The application of Hari's multiple
sequential t-test method (Reid et al., 2016) identified upward shifts in OLR in 1988 and 1997–98. Correspondence between
surface temperature and OLR was assessed. Tropical and NH midlatitudes shifted in unison, whereas SH OLR shifted in
tandem with NH and is more highly correlated with NH temperature (0.87) than with SH temperature (0.55) (Saltykov et al.,
2017). OLR responds in step with shifts in warming in a manner that maintains radiative balance between the hemispheres



(Saltykov et al., 2017). This reinforces the link between the warm pool and OLR, showing that shifts at the surface are being reflected at the top of the atmosphere.

## 6    Conceptual model

The heat engine model leaves the core greenhouse theory intact – increasing greenhouses gases trap additional heat near the planetary surface creating an energy imbalance at the top of the atmosphere. Warming will proceed until radiative equilibrium is achieved. It departs from the standard model on how that is achieved.

The following discussion builds on the points made in the introduction:

1.    We outline the physical constraints that merge the heat trapped by natural and anthropogenic forcing into a
single process. The additional heat not taken up by effective sinks (ground, melting snow and ice) is initially stored in the shallow ocean, becoming available for dissipation, including circulation into the deep ocean. A large amount of the surface heat absorbed is entrained into the Pacific Ocean heat engine.

2.    The heat engine maintains a series of steady-state regimes that oscillate on decadal timescales, influenced by external forcing and internal variability. In additional to ongoing circulation mechanisms, the heat engine
governs a storage and release system that includes ENSO and the PDO. More broadly, the shallow ocean-atmosphere supports climate regimes, which on decadal timescales are dominated by the PDO/IPO and AMO.

3.    To explain warming as series of regime changes, we apply the concept of Lorenz attractors acting on decadal timescales. These have three or more dimensions represented by simplified nonlinear equations and can be used to explain nonlinear behaviour in complex systems, such as convection. A robust aspect of the heat engine is
the constant TEP TWP temperature gradient that represents its ability to do work. Forcing produces a regime shift to more (less) work, which can be initiated either in the warm pool or cold tongue depending on the balance between supply and demand.

4.    The heat engine is networked to other oscillations and circulations in the climate system via teleconnections on temporal and spatial scales, relaying information on the capacity of the climate system to dissipate energy in
the most effective manner possible. These teleconnections are two-way. Under radiative forcing, shifts are first telegraphed via the ocean, resulting in widespread regime changes in SST, followed by atmospheric warming leading to shifts in land surface temperature.

These last two propositions are speculative and we have drawn from the literature in developing them, but they have supporting theory and evidence, and are testable.



## 6.1 Physical constraints


The main proposition of the conceptual model is that climate change is a single process combining various forcings and internal variability, and that the separation of forced gradual change from internally-generated nonlinear variability cannot be physically sustained for the following reasons:

(i)      Ocean heat conductivity and capacity dominates that of the atmosphere. Water has roughly 3,200 times the

capacity by volume and 24 times the conductivity of air. When air passes over a water surface, sensible and latent heat are exchanged until the two reach equilibrium, reached within a 300 m distance (Morton, 1986). The heat capacity of the atmosphere is equivalent to the top 2.5 m of ocean (Deser et al., 2003) and the annual average mixing depth of the atmosphere into the ocean is 90 m (Hartmann, 1994). Therefore, the shallow ocean has ample capacity to absorb available atmospheric heat. The ocean constitutes 70.8% of the global surface

area and approximately 75% of the tropical surface area, so dominates surface heat exchange.

(ii)      Total anthropogenic forcing is currently ~2 W m$^{-2}$ but net take-up is roughly equivalent to Earth's current energy imbalance of 0.71 W m$^{-2}$ (Loeb et al., 2018). Accumulation in the atmosphere 1993–2008 calculated from surface temperature was 0.007 W m$^{-2}$ per annum, likely to be over-estimated (Hansen et al., 2011). Scaling the estimates of Hansen et al. (2011) from the GMST record they used provides a conversion of 33 to

estimate the changing atmospheric heat flux.

(iii)      The damping effect of the ocean surface returns anomalies of >1 W m$^{-2}$ in the atmosphere to the mean steady state within months – an effect especially strong over the eastern equatorial Pacific (Park et al., 2005;Yoshimori et al., 2016). Circulation of the atmosphere over the ocean provides a strong negative feedback as heat is absorbed. Therefore, there is no physical barrier for the ocean to the take-up of the additional 1% or

so that is supposed to stay resident in the atmosphere.

(iv)      The shift/step ratio for the ocean globally is 0.88, and is between 0.73–1.05 for most zones except for the northern hemisphere extra tropics (0.58 30° N–60° N, 0.48 60° N–90° N; Table S3), so most of the ocean is close to steady state between regime shifts. Heat content in the upper ocean shows strongly regime-like behaviour (Ricketts, 2019).

(v)      The outcome is that the climate system cannot distinguish between the heat dissipated by the natural greenhouse effect (~155 W m$^{-2}$ (Schmidt et al., 2010)) and that trapped by the anthropogenic greenhouse effect. The standard model treats these as separate processes but if all heat is absorbed into the system that governs natural variability then the enhanced greenhouse effect will present as enhanced natural variability.

Therefore, under longwave radiative forcing, the atmosphere does not warm independently of the ocean. Ozawa et al. (2003)

describe climate in two parts: the radiative-transfer component, which is additive and responds linearly to forcing (Meehl et al., 2004;Marvel et al., 2015), and the dissipative system, which is multiplicative and responds nonlinearly.





The absorption of heat into shallow ocean storage makes it available for dissipation via nonlinear processes. Gradual warming cannot take place in the atmosphere or shallow ocean because negative surface feedbacks retain a stable mean state. High shift/step ratios that show warming in global SST is dominated by shifts (88%) rather than trends (12%) (JR2017 and
Table S3) shows that it does not.

## 6.2 Theoretical approaches

Two main theoretical approaches addressing the relationship between forced change and internal variability are the linear stochastic approach (Hasselmann, 1976;Leith, 1975) and deterministic nonlinear dynamics often based on Lorenz attractors (1963, 1968, 1975). The linear stochastic approach utilises the fluctuation dissipation theorem, which assumes a system will
produce a response linearly proportional to a small, incremental forcing as it moves towards equilibrium (Franzke et al., 2015;Ghil, 2014, 2019;Lucarini et al., 2017;Christensen and Berner, 2019). The stochastic part is produced by fast interactions in weather and climate inducing nonlinear responses in decadal and longer-term variability. These are superimposed on each other to form the standard signal-to-noise statistical model.

This theory has two problems with nonlinear change involving forced climate shifts. Firstly, Hasselmann (1976) in
combining the stochastic and linear dissipative elements, based it on the premise that the climate does not 'flip-flop'. It does. Secondly, neither the atmosphere or the shallow ocean produces the gradual response required, being dominated by regime-like behaviour (Jones and Ricketts, 2017;Bartsev et al., 2017).

The alternative, where changes in climate project onto the principal patterns of natural variability (Corti et al., 1999) has foundered, largely because of the many possibilities as to the end result (e.g., Bartsev et al., 2017). With a lack of empirical
evidence, there is little shared understanding as to what the result would look like. Most efforts to model how attractors might respond to forcing have also been on shorter timescales (e.g., Palmer, 1993;Palmer, 1999;Hannachi et al., 2017). These involve phenomena such as synoptic and seasonal regimes, and interannual oscillations, mapped onto attractors and seeing how they change over time (Corti et al., 1999;Hannachi et al., 2017;Bartsev et al., 2017).

Decadal regimes present a problem because of the relatively brief period of observations in establishing statistical
relationships and the capacity of climate models to represent them. Most authors who recognise decadal regime shifts see them as combining with an underlying trend (Tsonis, 2018;Overland et al., 2008;Branstator and Selten, 2009). Some regard the two approaches as complementary (e.g., Ghil, 2014, 2019;Chekroun et al., 2011;Hasselmann, 1999). However, if shifts in climate can be attributed to forcing at the expense of trends as in JR2017, this compromise retains the same problem as the linear response model (Bartsev et al., 2017).

Lorenz (2006) in referring to regime change considered the difference between those that occur within oscillations and those where there was little chance of return. The latter he did not deem regimes, referring to them as climate change. We see similar types of shift in mean climates on decadal timescales. One aligns with regime changes in oscillations such as the PDO or AMO where there is no net change in overall energy balance, but a distribution of heat in changing circulation patterns and/or sensible and latent heat balance (JR2017). The other is in response to an external forcing and is one way.



Over land, these manifest as a shift in sensible heat, showing abrupt shifts in maximum temperature-precipitation relationships partitioning sensible and latent heat in proportion to available soil moisture (Jones, 2012). Over the ocean they manifest as shifts in mean temperature without the partitioning provided by land surface feedbacks, so consequently, are more difficult to attribute.

## 6.3 The heat engine

The description of the heat engine and following two sections is speculative and attempts to link observations to theory, whereas nonlinear dynamics are usually modelled using simplified examples. How heat engines work in non-equilibrium systems with many dimensions remains a topic of much disagreement between specialists in the field (Lucarini et al., 2014;Lucarini et al., 2010;Kleidon, 2016;Kleidon and Lorenz, 2004;Ozawa et al., 2003;Franzke and O'Kane, 2017). How the heat engine manifests nonequilibrium steady states is the subject of this section, with power the variable of most interest.

A schematic of the heat engine is shown in Fig. 6. Kinetic energy from the east, trade winds and the Walker circulation moves currents formed from upwelling cold water in the southeast and return waters in the northeast via the subtropical gyres. Although these fluctuate on seasonal and interannual timescales and are influenced by the PDO, long term, the supply of kinetic energy into the heat engine will be roughly constant. The cold reservoir is the more stable of the two reservoirs, and although the cold tongue is large, its fulcrum lies in the Niño 3.4 area. The warm pool lies in the west and is less stable.

The heat engine discharges heat convectively from the warm pool and Hadley Cells, and advectively to the Indian Ocean, the Pacific Tropical countercurrent and subtropical gyres. The Pacific tropical countercurrent acts as a counterbalance and is the route travelled by El Niño events.

The 'uphill' gradient in the heat engine is sustained by incoming kinetic energy. If this remains relatively constant, the amount of power coming through the engine will be moderated by both supply and demand for the dissipation of energy.

Energy supply is dominated by the negative feedback of the ocean surface across the central-eastern Pacific, responding to incoming radiation and greenhouse gases reflecting heat downwards. The warm pool is the more sensitive part of the engine. Shifts in TEP are roughly twice as large as those in TWP and half as frequent.

During the historical period, the heat engine occupied two states: a relaxed (free) state to the 1950s–1960s dominated by internal variability and a tightly coupled (forced) state thereafter dominated by external forcing.





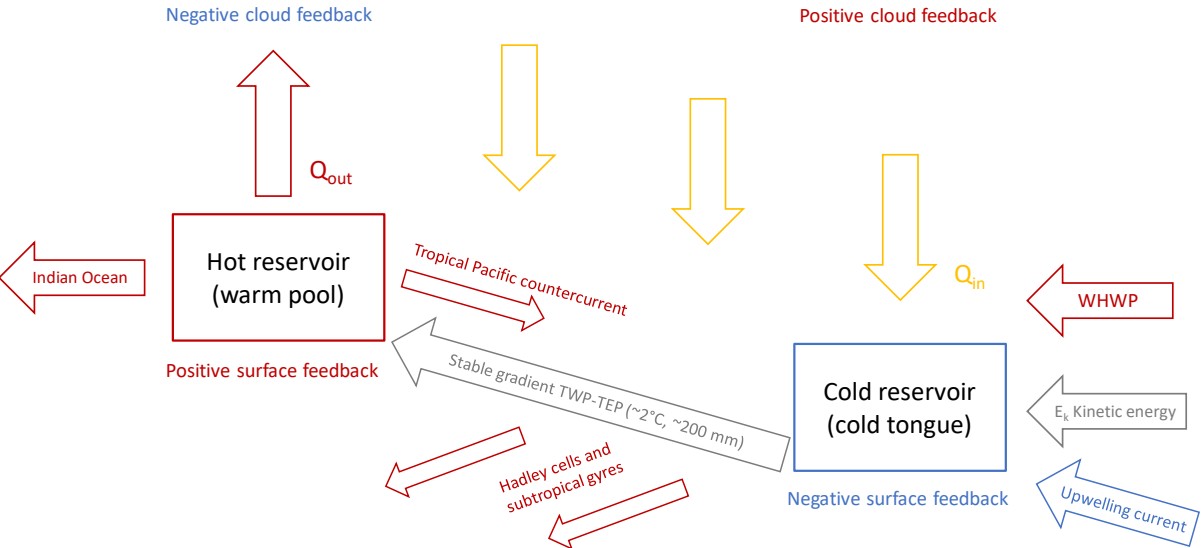


**Figure 6: Schematic of the Pacific Ocean heat engine showing kinetic energy (Ek) and power in and out (Q). Red denotes circulating heat, yellow incoming radiation, blue circulating cold and grey, incoming kinetic energy. WHWP is the western hemisphere warm pool.**

When the climate is close to or in equilibrium, the heat engine is in free mode and dominated by the cold tongue. Globally,
interannual climate variability is able to depart on short-lived random walks influenced by the AMO. The heat engine shows two-way coupling with higher latitudes. If forcing elsewhere in the climate system is negative, or power input reduces, TWP will shift downwards, because of the reduced demand to dissipate heat energy. If forcing is positive elsewhere (e.g., shortwave forcing in the extratropics), the heat engine will warm, which appears to have contributed to the 1921 shift in TWP. Such adjustments may be to maintain the symmetry in dissipation rates between the hemispheres – in GCMS the
transfer of energy to rebalance asymmetric warming occurs quickly (Haywood et al., 2016;Hawcroft et al., 2018).

TWP returned to its late 19$^{th}$ century mean value in the late 1930s, which was followed by an extended period of limited change dubbed the 'long pause' (Broecker, 2005). During this period, heat was absorbed by the ocean without warming the atmosphere. First the ocean, ocean-land and land temperatures progressively whitened as the heat engine changed from free to forced mode.

From 1968, all warming events were led by shifts in tropical SST, leading to new regimes being established. These were followed by El Niño events that warmed the atmosphere, extending over land. The warm pool was the early mover in 1968, 1995 and late 2012. The cold tongue – Niño 3.4 area was the early mover in 1976 and possibly the NH tropical ocean in 1986. Some shifts coincided with regime changes in the AMO or PDO, or just preceded them. The warm pool has been associated with a temperature anomaly of >0.2 °C for at least a 7-month period in every shift. Every shift has been followed
by one or more El Niño events that continue to warm the atmosphere (Fig 3b). This one-two action of warming in the shallow ocean preceding warming over land is the reverse of the conventional view, where warming over land leads warming of the ocean.





## 6.4 Oscillators, attractors and change

Palmer (1993) showed that if forcing was introduced into the Lorenz equations, the residence time can change in predictable
ways. The strength of forcing maintains the number and spatial patterns of regimes, but changes their frequency (Slingo and
Palmer, 2011). A role for external forcing in modifying the behaviour of decadal oscillations is increasingly being
recognised (Bellomo et al., 2018;Murphy et al., 2017;Di Lorenzo and Mantua, 2016). Bartsev et al. (2017) propose four
possibilities: 1) the climate system possessing multiple stable stationary states with stochastic-like variability being of
external origin; 2) there is a set of more simple bistable systems with only one system switching, also due to external origins;
3) variability is an inherent property of the system, chaotic or near chaotic and shifts are transitions between attractors; 4)
there is a system of interconnected (coupled) oscillators and shifts are transitions between attractors of this super system. The
latter invokes a network and Bartsev et al. (2017) also considered combinations of the above.

Two oscillators within the heat engine are ENSO and the tropical expression of the PDO. For ENSO, four oscillators that
provide negative feedbacks have been proposed: A delayed oscillator of reflected waves from the western ocean boundary
moving east, a recharge-discharge oscillator between the east and west, a western Pacific oscillator that adds a specific warm
pool component to the delayed oscillator, and an advective-reflective oscillator where currents move the warm pool east,
which are then reflected back (Wang, 2018;Wang et al., 2017). A unified oscillator theory unites all four, suggesting they
combine in different ways when various events occur. These oscillations may be self-sustained and inherently unstable, or a
stable oscillator that is modified by noise to produce instability (Wang, 2018;Wang et al., 2017). The recharge-discharge
oscillator is present, consistent with the sea level seesaw introduced by Wyrtki (1975) and revisited by Peyser et al. (2016).

The second is the PDO. Ramesh and Cane (2019) using standard deviation in the Nino 3 area mapped it as an attractor in
three dimensions showing it occupied an inner and outer orbit in observations, the intermediate complexity Zebiak-Cane
model and baseline runs in two GCMs. The ZC model was the most successful producing both positive and negative
regimes. Observations bore the closest resemblance to that model, which was not conclusive due to the length of record, and
one of the two GCMs reproduced a similar pattern. In the one that didn't variability was wind-driven, lacking an oceanic
component. They conclude that mapping this or a closely-related attractor has the potential to forecast Pacific decadal
variability (Ramesh and Cane, 2019).

Whether mean shifts in temperature within the heat engine also map onto an attractor or attractors (east and west) has not
been ruled out, but nor can it be ruled in at present. The heat engine appears to self-regulate according to a thermodynamic
limit, such as maximum entropy production (Kleidon and Lorenz, 2004;Kleidon, 2009;Ozawa et al., 2003) and also becomes
unstable if energy supply and/or demand falls below a given amount. Maximum entropy production is "where complex
systems with sufficient degrees of freedom maintain a steady state at which entropy production is maximized" (Kleidon,
2012). This concept does not have a physical basis and is largely descriptive (Westhoff et al., 2019) as is the case for
attractors. Yet influenced by internal and external forcing, the climate system is either shifting between oscillating steady
states or to a new state, so there has to be a source of instability that leads to such shifts.





The presence of free and forced modes in the heat engine complicates the picture somewhat. Pauluis (2011) discusses a mix of sensible and latent heat in an idealised steam cycle and Carnot cycle, pointing out that the steam cycle produces less work and that Carnot efficiency is not reached until saturation. When the heat engine is in free mode, TEP has the greater relationship with GMST possibly due to the predominance of descending dry air in the east. When the climate is warming,

TWP is prominent and linked to GMST, and SST whitens. Under that model, TWP and the ocean more generally will drive convection. Perhaps the free mode phase dominated by the TEP is running below Carnot efficiency, allowing temperature to go on random walks influenced by climate variability increasing autocorrelation (Sect. 5.1, Fig. 5), whereas during forced change, the heat engine runs hot and interannual variability whitens.

Adding incremental forcing to the classical Lorenz convective system produces a linear response (Christensen and Berner,

2019). However, the same authors also find that for a climate prediction to be accurate, a model has to be able to capture the slowest growing modes (Christensen and Berner, 2019) as was the case for the Pacific decadal variability example of Ramesh and Cane (2019), suggesting that experiments without those modes may not yield appropriate results.

Lorenz (2002) describes an experiment by Malkus (1954) who heated a fluid layer between two reservoirs and observed heat transport increase in discrete steps as new convective modes were established. This may be related to self-organised

criticality where the highest throughput is obtained producing an exponential frequency/magnitude relationship (Bak, 1996;Lorenz, 2002). For example, 420,000 years of Vostok ice core temperatures (Petit et al., 1999) under natural forcing follow such a progression (author's own calculation).

Under these conditions, a shift in temperature may the result of a system that is no longer stable within thermodynamic limits and needs to establish a new convective regime. The shallow ocean becomes unstable as heat builds and this instability is

telegraphed from the initial trigger point to other regions. Shifts in atmospheric temperatures over land follow, once those higher temperatures can be sustained by warmer SSTs.

If we consider the warm pool and cold tongue as potential wells that maintain steady state, the cold tongue is relatively insensitive to changes in incoming energy. Based on measurements of temperature as a proxy for power, the well in the cold tongue is roughly 0.46 °C deep. For TWP, it is roughly 0.15–0.25 °C deep. Shifts will not occur until the limits of the stable

state are approached, measured as an anomaly >0.2 °C in TWP persisting for more than about nine months. Opposingly, a reduction in forcing would lead to the engine dropping from forced to free mode. Further reductions would then lead to shifts to cooler conditions as less energy becomes available for dissipation. Following Lorenz (2006) preference, a redistribution of energy as in a decadal oscillation would be classified as a regime shift, whereas a change in temperature and associated variables would constitute a powershift, akin to a change in gear in an automatic gearbox.

**6.5 Network**

Once a shift is initiated, SST increases in regions teleconnected to the warm pool. All of the observed shifts from 1968 have seen the first response in the ocean and all but one (1986) was initiated in the tropical Pacific. These are followed by shifts in SST in the extra-tropics as regime change is initiated and established. Network analyses of climate have become increasingly



common, but most are more relevant to timescales of less than a decade. Networks can be analysed spatially, by seeing how

closely different locations are to each other, or as networks of networks, using indices of large-scale phenomena. For example, de Viron et al. (2013) analysed lags from 25 indices, showing that the two main orthogonal modes explaining total variability in SST corresponded with the AMO and PDO. The AMO lagged 12 months after the indices and the others, including the PDO, 12 months before. The short timescale (1948–2011) did not allow longer than decadal periodicity to be clearly defined, but all modes contained a ~20-year periodicity.

At the spatial scale, Tantet and Dijkstra (2014) constructed an interaction network from deseasonalised, detrended gridded SST from the HadISST data set. This was based on correlation with no lag and measured the number of connections for each gridpoint average across the globe. Key nodes were then identified and grouped into ten main nodal fields (communities). The largest covered the E central, N and SW Pacific and W Indian 68.7% of the total, the eastern N Atlantic 6.2% and W Pacific warm pool 6.1%. At this scale, ENSO clearly dominated. They then filtered out the 2 to 7-year band to assess decadal

variability. The resulting major communities were the E Pacific with small links elsewhere at 31.2%, the Pacific warm pool – N Indian 26.3%, N Atlantic linked to central N Pacific 15.3% and Southern Ocean 11.2%. At decadal scale, the E Pacific and warm pool become more evenly balanced in extent.

Correlations with global mean temperatures were then calculated. For the Pacific warm pool–Indian ocean community correlation with land-ocean was 0.84, land 0.77 and ocean 0.78; the E Pacific community was 0.66, 0.44, 0.71 respectively

and north Atlantic community 0.65, 0.49 and 0.67. When fitted to GMST, the warm pool community multiplied by a factor of 0.72 produced a correlation of 0.71. The warm pool multiplied by a factor of 0.60 and N Atlantic by 0.31 produced a correlation of 0.87. When fitted to land surface temperatures, the warm pool multiplied by factor of 0.86 and N Atlantic by 0.29 produced a correlation of 0.66 (Tantet and Dijkstra, 2014). Therefore, using detrended data on a decadal timescale, the Pacific warm pool – N Indian ocean region is the principal network of influence, the N Atlantic with some N Pacific is

second and the Eastern Pacific cold tongue is third (Tantet and Dijkstra, 2014). By a completely independent method, the warm pool is identified as having the primary network influence on global temperature.

Three ensemble members of the MPI-ESM-LR climate were similarly analysed. They showed a simpler networked structure that ranked the cold tongue first and warm pool second when correlated with GMST. This is consistent with our conclusion that the heat engine is present in GCMs but is simpler in structure. On a decadal basis, heat engine interactions within this

model are more dominated by the eastern Pacific than the warm pool and N Atlantic (Tantet and Dijkstra, 2014).

The following authors have linked shifts in GMST to widespread spatial coupling between selected indices (de Viron et al., 2013;Tsonis et al., 2007;Wang et al., 2009;Tsonis and Swanson, 2011;Swanson and Tsonis, 2009;Tsonis and Swanson, 2012). Four modes of variability, ENSO, PDO, and the North Atlantic Oscillation (NAO) and North Pacific Index were combined into a network and tested for synchronisation (cross correlation) and coupling strength, measuring whether they

are moving in phase with each other or not. Five periods of synchronisation were identified, three ending with a climate shift and two without. Periods identified with shifts were around 1910, 1940 and the mid-1970s and without were the 1920s and



1950s (Tsonis et al., 2007). As we have noted, the 1920s exhibit shifts that were larger and more widespread than those a decade earlier. Later 2001–02 was added as another period ending in a climate shift (Swanson and Tsonis, 2009).

The timing in these studies has coupling following the shifts documented here rather than preceding them (see Swanson and Tsonis, 2009). This group have not included the AMO or related index of longer-term variability in the Atlantic and nominate coupling between the NAO and other modes as the common element in all recorded shifts (Swanson and Tsonis, 2009;Wang et al., 2009;Tsonis, 2018). However, the NAO contains little or no decadal structure and we suggest their method is responding to the reorganisation of regional climates as new climate states are being established. Largely atmospheric oscillations such as the NAO, may therefore be responders rather than initiators.

We also see some evidence of switching between different modes based on the rate of dissipation as forcing requires a change from slow to fast processes. For example, if the AMOC cannot move heat poleward fast enough, more rapid processes will take over. The three main sets of dissipative processes based on the structure of the climate system are fast, dominated by the atmospheric processes (circulation and feedback); moderate, dominated by the shallow-ocean atmosphere system and slow, dominated by deep ocean overturning. A fourth, mainly geochemical, is much slower (Kleidon, 2012) and not so important on these timescales.

Fast atmospheric feedbacks involving changes in rainfall and cloud respond quickly to changes in forcing (Zelinka et al., 2013). As forcing increases, dissipation will switch from slow-moderate processes to moderate-fast processes. This also delineates the different ocean basins. The N Atlantic is the principal slow-moderate Basin and Southern Ocean secondary. The Pacific Ocean is the principal moderate-fast Basin and the Indian Ocean secondary. During the first stage of positive forcing, if the heat engine is in free (resting) mode, the shallow ocean will take up heat until it becomes tightly coupled east and west in the tropical Pacific. The warm pool then takes over as the main dissipator of additional heat, coupled to other modes in climate system. This continues until the supply exceeds the capacity of the eastern Pacific, which then begins to warm, simultaneously shifting global warming from low atmospheric feedback to high. AMOC slows down during this process because it cannot dissipate sufficient heat. Consequently, dissipation switches to faster pathways and effective climate sensitivity increases from low to high due to the role of feedbacks as part of that process.

This raises the question – under continued forcing at what point does the current configuration of the climate network become inherently unstable and switch to new modes?

## 6.6 Terminology and characterisation

The existing language describing climate change is framed around linear change. Terms like trend, rate of change in terms of °C per decade and the statistics surrounding these are very familiar. There is no consistent language for describing nonlinear change. Many of the terms like oscillations, regimes, shifts, steps, breakpoints, rapid, abrupt, strange attractors, have technical definitions but these differ between disciplines (Kopp et al., 2016;Imkeller and Von Storch, 2012). There is no accepted taxonomy for characterising changes between states, or for characterising specific states. Westhoff et al. (2019) are



endeavouring to do this for thermodynamics in climate. There is a need also to do this for the nonlinear processes described
in this paper and more generally.

For example, Overland et al. (2008) list definitions of regime based on displacement or shifts, underlying mechanisms and the nature of forcing (external and internal). All are relevant for understanding impacts, but are inadequate for describing the interactions between the heat engine and decadal regimes under forcing. We have called the size of the dislocation between internal trends a 'shift' to distinguish it from a step – the statistical measure between two states assuming no trend, but both
are commonly used interchangeably. Here we suggest a regime shift for an oscillation following Lorenz (2006) and a powershift for a one-way change to a new state. Some events may involve both (e.g., 1976–78, 1995–98 and 2012–15).

The context for how language is used is also important. Terminology can be theory-based, methodological (statistical, conceptual and mathematical models) or practice-based, where it has a practical meaning for the end user. Ocean heat uptake efficiency is a term that started as a methodological concept but is now largely considered to affect the rate of warming,
whereas we argue that because there is no physical barrier for shallow ocean heat uptake it is not physically meaningful. The more important issue is how fast does heat come back out of the ocean? For this question, deep ocean heat uptake efficiency may well have physical meaning (Yoshimori et al., 2016).

Having a sensible scientific discussion about alternative models is difficult if one model has an established language and the other does not. The default is to gravitate towards the language of linear trends because it is the topic most well understood.
If climate change is nonlinear on the timescales that matter, this is a problem.

## 7    Conclusions

Here we describe a heat engine spanning the tropical Pacific Ocean, comprised of the cold tongue in the east-central Pacific that collects and concentrates heat and the warm pool in the west that disperses it. We propose that the heat engine plays a self-regulating role in the climate system by interacting with a network of oscillations and cycles that couple the ocean and
atmosphere. The result is a series of steady-state regimes manifesting on decadal timescales.

When SST representing the cold and warm reservoirs of the heat engine were analysed for shifts in mean temperature, most shifts preceded larger scale responses within the climate system. Tracking SST from 1947 revealed a sequence of shifts in mean temperature beginning in the tropical ocean and propagating more widely, followed by ENSO-related atmospheric warming in the atmosphere. Five of six event sequences since 1968 were initiated from the heat engine, sometimes
coinciding with a phase shift in the PDO or AMO.

The heat engine is an emergent property of the climate system and plays a self-governing role. It is also present in coupled AOGCMs, though in a less sophisticated form than in observations. The main point of difference between this and the standard model is that all available heat in the atmosphere is absorbed by the ocean and later released during a change in state, rather than gradually warming the atmospheric column directly.





The heat engine concept provides explanations for phenomena where clarification is currently being sought or that have been misdiagnosed:

- The so-called hiatus (1997–2013) was a hiatus, but this is the rule rather than the exception. Its extended length was due to the reasons limiting forcing at the surface summarised in Medhaug et al. (2017). The current hiatus is likely to be short-lived because of increasing forcing and a positive PDO.

- The recent warming (2014–15) was a shift, not the resumption of a steeper trend. It will be followed by more shifts in the near future.

- Historically, warming was initially delayed due to the absorption of heat by the shallow ocean. Once this capacity was reached, the heat engine switched from free to forced mode. Warming dominated by the warm pool will lead to lower atmospheric feedbacks. The use of this warming profile to estimate climate sensitivity
will produce an estimate at the lower end of the range of uncertainty.

- Once warming in the central eastern Pacific proceeds apace, realistic estimates of climate sensitivity can be obtained. Using step size in TEP and a CMIP5 model ensemble, our crude estimate of ECS is 3.2±0.6 °C.

- Autocorrelation within GMST is largely a product of the AMO and was present when the heat engine was in free mode. Once the heat engine switched from free to forced, interannual variability whitened substantially.

- Transient warming is not a product of how fast heat is absorbed by the ocean but the result of how fast it is released.

- Under greenhouse gas forcing, the ocean warms before the land – the land warming first is an artefact of signal-to-noise statistics – the ocean warms first but the land warms faster.

- The PDO Index provides a reliable signal for the behaviour of the decadal oscillation within the tropical
Pacific, modulating the performance of the heat engine. The full Pacific decadal variability has accompanying north and south components.

- Under sustained greenhouse gas forcing, additional heat moves from the tropics to the mid and high latitudes, not the other way around.

- Warming is a two-speed process – in most locations the vast majority of warming will occur in <5% of elapsed
time.

The tropical Pacific heat engine is the major pacemaker of global warming. The last shift major shift was initiated at the end of 2012 and took place during 2014–15. Every powershift carries the risk that other systems may be de-stabilised creating further feedbacks; e.g., ice melt-albedo, ice-sheet destabilisation-sea level rise, permafrost-methane or wildfire-$CO_2$ emissions. Given the current positive PDO and rate of forcing, the next shift can be expected to be initiated in the next few
years. Rapid decarbonization will slow the amount of heat moving through the heat engine, raising the possibility of stabilizing climate. Even if rapid decarbonization were to succeed, several more shifts over coming decades are likely. Stabilisation of the tropical Pacific heat engine will not stop the heat already moving towards high latitudes, especially in the

ocean, which could potentially disrupt the climate system further, but it will prevent worse outcomes. Understanding what these risks are and how the heat engine may behave in overshoot and rapid reduction scenarios is a priority.

The identification of a process of regime-like change over decadal timescales has important implications for how changing climate risk is characterised. Shifts in climate will have their largest impact on extreme events, which cascade through natural and human systems (Jones et al., 2013;Ebi and Hess, 2017). Developing the capacity to predict future shifts in the heat engine, and its potential links with the various oscillations in the tropics and extra-tropics is critical. Developing a predictive capacity would also require reorienting current research on decadal scale climate change from producing

ensembles of decadal trends to better understanding the mechanisms behind regime shifts, powershifts and how to predict them.

## 8    Code availability

The experimental version of the MSBV code used here can be found in Jones and Ricketts (2019b), in addition to a manual version in Microsoft Excel©. As this material is still in review, reviewer access is via:

https://datadryad.org/stash/share/7OmkMfNbdgFdPIxvD5qyqhmrzh69ug_tT5MNZ2JMPTQ

## 9    Data availability

The data generated, tracking model and data for the figures can also be found in Jones and Ricketts (2019b) at https://doi.org/10.5061/dryad.n02v6wwsp. As this material is still in review, reviewer access is via: https://datadryad.org/stash/share/7OmkMfNbdgFdPIxvD5qyqhmrzh69ug_tT5MNZ2JMPTQ

## 10    Team list

Roger N. Jones, Victoria University
James H. Ricketts, Victoria University

## 11    Author contributions

RJ conceived the study, JR coded and tested the multi-step model and developed the post-step verification detailed in the SI,
both conducted analyses, RJ led the paper with contributions from JR.

## 12    Competing interests

The authors declare that they have no conflict of interests.



## 13 Acknowledgements

JR is the holder of a Victoria University postgraduate research scholarship. Data sources include the National Oceanographic
Data Center and United States National Climatic Data Center, UK Met Office Hadley Centre, the KNMI climate explorer,
the IRI/LDEO Climate Data Library Colombia University. CMIP5 archives are made available by the modeling groups, the
Program for Climate Model Diagnosis and Intercomparison (PCMDI) and the WCRP's Working Group on Coupled
Modelling (WGCM). The U.S. Department of Energy's Program for Climate Model Diagnosis and Intercomparison provides
coordinating support and led development of software infrastructure in partnership with the Global Organization for Earth
System Science Portals.

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
