# Peer review of "The Pacific Ocean heat engine: global climate's regulator"

_Earth System Dynamics, 2019_

## Short Comment (SC1) · 14 Dec 2019

The authors provide analysis of consequence of temperature shifts, which were previously observed in a number of studies related to various regions. They as I know are the first who showed that temperature shifts at tropical west Pacific warm pool precede most of the globally important shifts in other regions. Then they described proposed mechanisms of the way how tropical Pacific heat engine maintains surface temperatures behaves as quasi-steady regimes punctuated by abrupt shifts. Both data analysis and description of proposed underlying energy flows provide new insights to the understanding of climate phenomena previously observed by many authors but still not fully understood. Nevertheless, I suggest minor revisions before publication for the clarification descriptions of tropical Pacific heat engine: 1) The authors numerous times use

the words sink and source (and also "absorbing heat", "losing heat") describing energy flows of tropical east Pacific (TEP) and tropical west Pacific (TWP). But usually they don't specify what energy exchange is considered in a particular part of text – radiative flows to/from space, exchange with atmosphere, with deep ocean, with nearby regions. I suggest making clear specifications to what energy exchange word sink or source is used in any particular part of text. 2) I suggest providing more detailed explanations for the consideration of the TEP and TWP as a separate reservoirs based on hydro-climatology. Because from the first view it seems that TEP and TWP are the same body of water commonly (excluding ENSO events) moving from west to east like in a big tube. 3) Sometimes authors use terms "negative feedback" and "absorbing heat" (correspondently, "positive feedback" and "losing heat") like synonyms. For example, in the beginning of "Atmospheric feedbacks" section they wrote: "TEP represents a negative feedback at the ocean surface absorbing heat whereas TWP represent a positive feedback, losing heat." By definition, negative climate feedback is any process where climate feedback decreases forced warming. So some initial forced warming (e.g. "anthropogenic warming") causes a secondary change that reduces forced warming (by absorbing additional "anthropogenic" heat in deep ocean, reflecting it back to space and etc.). In general negative feedback may be provided by region losing heat also. Thus in the cited sentence, for example, it is not clear if "negative feedback" is a consequence of "absorbing heat", or a coincidence of these two properties is a result of various other processes. Also in "Atmospheric feedbacks" section it is not clear how this negative/positive feedbacks are related to observed regimes and shifts of surface temperature anomalies. Thus I suggest to add in this section paragraph clarifying authors view on underlying processes.

---

## Referee Comment (RC1) · Anonymous Referee #1 · 21 Jan 2020

The manuscript describes an approach to the study of the role the tropical Pacific ocean has for the thermodynamics of the climate system, relying on the thermal engine concept. Starting from a previous work by the authors, a rigorous statistical testing is used in order to investigate step-like changes in the observed evolution of sea-surface temperatures (SST) in the Tropical Western Pacific (TWP) and Tropical Eastern Pacific (TEP), relating them with the evolution of global mean surface temperatures (GMST) and some indices describing inter-annual and decadal natural variability (in particular the Pacific Decadal Oscillation, PDO, the Atlantic Multi-decadal Oscillation, AMO, and ENSO). It is argued that the tropical Pacific can be seen as a thermal engine transporting heat up-gradient (i.e. from a cold to a warm region), and that this mechanism is characterized by two modes: until the second half of the 20th Century a free mode is

associated with the natural variability at the decadal timescales, while in more recent years a forced mode is established, in which the natural variability is enhanced by the anthropogenic greenhouse gas (GHG) forcing. The authors argue that these findings provide evidence that the classical view of the response of the system to the GHG forcing as a trend-like behavior with superimposed noise-like natural variability is faulty in describing the transient climate change on a decadal timescale.

Overall, I appreciated that the authors started from a climatological point of view (i.e. from observations) in addressing the challenging issue of how climate change projects on the natural modes of variability at inter-annual to decadal timescales. However, conclusions do not seem supported by sufficient evidence from results, at least the way these have been illustrated.

In the first part, authors argue that step-like changes in TEP and TWP, in part related to changes in the phase of the AMO and PDO, as well as ENSO events, shape the decadal evolution of GMST. The emergence of step-like changes is detected through the usage of established statistical methodologies. On the contrary, the propagation of the signal (cfr. Sect. 3.1) is discussed in terms of timing across different time series, and this qualitative arguments severely undermine the robustness of the results. The authors seem to claim that conclusions similar to those based on observational-based datasets can be drawn from investigation of CMIP model outputs. A visual inspection of the model results shown in the supplementary material, though, does not seem to support this conclusion.

In the second part (Sect. 6), arguments are provided to describe the Pacific ocean thermal engine, how its changes propagate to the global climate, and how its internal variability is affected by the forcing through nonlinear mechanisms. This section is deliberately qualitative, and in some parts speculative, and this is of course absolutely fine. The problem is, that the section has the form of a review of existing literature, particularly in sections 6.2, 6.4 and 6.5, with insufficient or completely absent relation to results described in the first part. These sections alone sum up to almost 40% of

the manuscript, making it dispersive and difficult to read. This is not appropriate, in my opinion, for an original research article, and shall be reconsidered.

Overall, even though the results from the first part are potentially interesting and supporting the conclusions given at the end of the manuscript, I think that a more rigorous scientific approach is needed in order to make the manuscript worth publishing. My suggestion is that the authors consider splitting it into two parts. In a first paper (approximately sect. 2-5), the statistical methodology is explicitly accounted (instead of referring to a previous paper), the shifts in TEP-TWP temperatures and their relations with GMST, AMO, PDO, ENSO are explicitly addressed through rigorous methods for causal inference detection. The network approach outlined in sect. 6.5 might also be helpful in this respect. In a second paper (sect. 6), more emphasis is given on the description of the conceptual model of Pacific thermal engine. In doing so, the heat sources and sinks, the working temperatures, the Carnot efficiency and actual efficiency have to be quantitatively addressed, possibly considering the compliance to the 1st and 2nd laws of thermodynamics via energy and entropy budget.

I pointed out in the following that the authors sometimes refer to a vocabulary which is not specific and potentially confusing for a scientific audience. I suggest that the authors make an appropriate usage of terms, avoiding expressions such as "flip-flop", "disrupt the climate system". Related to this, the manuscript is lengthy, and in some parts difficult to read, and it might be worth considering a significant reduction.

In the following, specific comments are provided and, where applicable, suggestions on how to address them.

Specific comments

l. 33: I do not understand why the authors oppose the approach of emerging trend-like evolution of the forced climate response, to approaches where the non-linear interactions between natural variability and forced response are explicitly taken into account. As found in the IPCC 5th AR, at sect. 11.3.1.1 (I suggest that, when possible, the au-

thors refer to the original publication, rather than the chapter of the assessment report where the information is taken from, e.g. Hawkins and Sutton 2009, in this case), "The evolution of the S/N ratio with lead time depends on whether the signal grows more rapidly than the noise, or vice versa". This does not rule out the possibility that the noise itself is affected by the forcing; it just says that, if the forced signal grows faster than any other change due to the internal variability of the system, the latter is treated as noise. The rationale behind the S/N approach is not to be found in the radiative-convective model, rather in the fact that the near-surface layers of the atmosphere, and particularly the land surface, are heated over timescales comparable to the inter-annual and decadal timescales of the natural variability. Furthermore, one may add that the forced response should be treated as a trend-like response specifically in the case of climate models, where the internal variability (which is related, but is not coincident, in the model world, to the concept of actual natural variability of the system) is (almost) insensitive to the forcing over a wide range of timescales. In addition to that, I believe that it is nowadays commonly accepted that the regional forced response is unavoidably affected by changes in the statistics of the natural variability due to the forcing, so that the trend-like approach alone is not a viable option.

ll. 41-42: showing that a signal responding to a monotonic trend has a not strictly monotonic behavior does not necessarily imply that there is some sort of interaction between the forcing and the internal variability of the system, contrary to what seems to be suggested here.

l. 61: it is not clear to a reader who is not aware of the 2017 paper, what "explanatory power" means, especially given that a measure for that seems to be implicitly adopted here;

ll. 63-64: is this result from the 2017 paper relevant in this context? Does it relate to the results shown in this manuscript?

ll. 81-82: in my opinion it is a bit misleading to look at the Earth's energy imbalance (EEI; I think this is what the authors are referring to here) as an atmospheric energy deficit. The role of the atmosphere is determining the rate of absorption/scattering/emission of solar/infrared radiation through its chemical composition. The atmosphere has a very small thermal inertia, limiting the storage of radiative heat in its interior to (less than) 1%, whereas all the remaining EEI goes into the ocean, warming the surface, melting sea-ice and glaciers (cfr. Von Schuckmann et al. 2016).

ll. 97-98: I do not see which literature reference introduces this definition and, more importantly, which evidence is behind it. Certainly, the paper by Kjellson et al. 2014 does not address this definition.

ll. 117-118: this up-gradient flow of heat might well be compensated by subsurface return flow, to my understanding. If this is the case (although I have no reference in mind for that), does it make sense to consider the upper part of the ocean as a closed system, given also its heat exchanges with the deep layers in the subsidence and uplift regions?

l. 124: I do not agree with the statement that internally the system is not in equilibrium with incoming radiation from the Sun (if I interpreted correctly what the authors mean here). Climate is not in thermodynamic equilibrium with its surroundings, rather it is in a statistically steady state, meaning that, in the absence of any forcing, the net energy input equals the net energy output over sufficiently long timescales. More appropriately, it can be said that the system is a non-equilibrium dissipative steady-state system. The atmosphere is on average in energetic balance with its surroundings, and same can be said for the oceans, even though the time needed to achieve such a balance greatly varies depending on the subsystem that is taken into consideration.

ll. 136-139: I think that the authors might discuss to what extent the efficiency in the meridional heat transport is maximised in the climate system. One can have meridional transports simply as a consequence of the differential in diabatic heating (cfr. Lucarini et al. 2011, JAS), but energy can be transported (and transformed) in many different

ways, and the moisture is a critical feature, in this respect (cfr. Yang et al. 2014, Clim. Dyn.).

ll. 170-171: are these values provided with an uncertainty range? If so, is it possible to have them shown?

ll. 174-175: given that the existence of step-like changes in the TEP and TWP and the mechanisms underlying them are at the core of the results, statements like this one shall be corroborated by more quantitative arguments. At l. 202 the authors claim that they start to track changes after 1947, because of the poor quality of data prior to WWII. Then why caring of shifts happening in 1937-1942?

Figure 2d: the authors compare here anomalies in GMST, TWP and TEP with TWP-TEP difference. The visual effect is that variability in the anomaly time series are damped by the temperature gradient offset about 2 K. I suggest that the gradient and the anomalies are shown in different panels.

l. 190: I do not understand the relevance of this statement. The GMST are roughly stationary from 1880 to 1920. If we consider the 1900-1920 instead of 1880-1899 as baseline period, the warming in TEP and TWP would probably be consistent with GMST warming.

Sect. 3.1: the title and the first sentence in this section suggest that at least qualitative arguments are provided to explain a causality mechanism connecting shifts in TEP/TWP with regime changes in natural variability (namely, PDO and AMO). The section is rather a collection of insight descriptions of step-like changes, in which the propagation of the signal is argued in terms of their coincidence with regime shifts and impacts over various aspects of climate variability. I also struggled with the definition given of "tracking model", given that the approach here shown seems to me rather an interpretative framework of the observed time series. It is interesting to infer causality links locally and remotely (the latter is partly accomplished in sect. 5). Nevertheless, there are several rigorous methods that might better serve the scope (e.g. Granger

causality and derivatives) and I think that they should be addressed here. Figure 3: I found this figure very difficult to read. This is probably because it spans two pages, and also because the captions are not very informative, especially referring to panels (c)-(g). Please consider expanding the caption and/or splitting it in several figures.

ll. 231-257: the description of selected shifts is linked to several events that happened in different regions of the Earth. The connection seems basically motivated by the timing, but there is no specific argument for these links to be descriptive of large-scale processes occurring into the system, so I wonder why it is relevant to consider them.

l. 274: do the authors refer to an increase in the incident solar radiation, or are they referring to net solar radiation? This shall be specified.

Sect. 4.2: as in my comment about ll. 231.257, the authors propose here a connection between changes in TEP, TWP and TEP-TWP gradients with the AMO. They provide a survey of results available in literature about AMO phase shifts and link them to the step-like changes found in sect. 3.1, but nowhere is suggested that the two changes are correlated, nor any process responsible for it is indicated. While I understand the motivation for the arguments about PDO (in sect. 4.1), I do not see the reason for this discussion about AMO.

ll. 372-376: the statements in this paragraph also seem to suggest that the arguments about the timing in AMOC changes, PDO and TEP/TWP/TEP-TWP changes are rather speculative and a rigorous analysis is missing, relating these evolutions. Given that the AMO and AMOC are barely mentioned elsewhere in the manuscript, the authors might consider simply withdrawing these paragraphs.

l. 388: I am a bit confused about what "tightening of the heat engine" mean in this context.

ll. 389-390: this could be related to the definition of forced and free fluctuations described in Lorenz, 1979 (the authors cite it elsewhere in the text).

[Figure]

Sect. 5.2: given that the authors refer many times to the co-variability of GMST, TWP and TEP, I do not understand why this section is only at this stage of the manuscript, and not before (same for Figure 4b). I would suggest that it is moved before sect. 3.1, after Figure 2 is described.

ll. 433-435: I wonder why reporting the number of detected shifts is important here, especially given that the authors claim at l. 437 that many models show a "pronounced decadal variability". Typically, the multi-model analysis is ill-posed, if one does not provide a weighting scheme or qualitative arguments discerning the models. This shall be addressed when providing multi-model averages.

Sect. 5.4: the aim of this section is supporting results shown in previous sections but, in order to do so, authors provide a survey of previous analyses (Andrews et al. 2015; Andrews and Webb, 2018; Dong et al. 2019). I think that it would be more appropriate if the authors would extend this discussion describing the correlation of their data with actual measures of the atmospheric feedback, such as OLR, the cloud radiative effect (CRE). These are available either as satellite data (e.g. CERES EBAF), Reanalyses (ERA5) or climate model outputs (CMIP5, CMIP6).

l. 462: what do the authors mean by "positive spatial variations in atmospheric feedbacks"?

ll. 533-558: the authors claim here that treating the forced response and the internally-generated nonlinear variability in a separate way has no physical foundations, and they give arguments to explain that. Provided that I tried to explain in my comment to l. 33 that this sharp separation is due to a partial misinterpretation of the theory and modelling of the climate response and forcing attribution, I will try here to problematize some of the arguments. The main underlying argument they provide is that the fact that the ocean uptake of the energy imbalance determined by the anthropogenic forcing implies that the response of the system is modulated by the regime changes in oceanic variability. The authors claim at ll. 505-507 that the warming ultimately determined by

the greenhouse gases is not affected by their conceptual model of the Pacific thermal engine. Rather, the model seeks to better characterize the paths through which the warming can be achieved. The S/N approach is indeed aimed at determining the overall response of the system to the anthropogenic forcing at timescales for which most of the modes of natural variability, including oceanic multi-decadal variability, is treated as noise. When looking at decadal variability or shorter timescales, the way variability is affected by the forcing is the main subject of investigation in many fields, ranging from North Atlantic weather regimes (e.g. Strommen and Palmer, 2018), ENSO regime changes (e.g. Kim et al. 2014, Cai et al., 2015, Kohyama et al. 2018), to the impact of resolution in climate models (e.g. the European project PRIMAVERA H2020). At ll. 549-550 they argue that the marginal uptake of heat by the atmosphere is soon to be uptaken by the ocean. I do not see why this should be relevant, given that the atmosphere exchanges heat mainly through latent heat and isotropic emission of LW radiation: the latter, mainly affects the exchanges of heat between the atmosphere and the continents, and is only partly emitted towards the ocean's surface. Given that, the atmospheric marginal heating is the net result of all these exchanges, thus it should not be accounted for as a heat source for the ocean.

l. 575: the authors might be more specific and explain what they mean in a scientific context when they say that the system "does not 'flip-flop'".

ll. 578-583: surprisingly, the authors seem to ignore that significant rigorous results have been achieved on characterizing the impact of the forced variability on a chaotic system. This includes approaches based on studying the parametric smoothness of minimal QG models (Lucarini et al. 2007, Physica D), the crisis of the attractor by means of Koopman operators (Tantet et al. 2018, Nonlinearity), stochastic perturbations of edge states (Lucarini and Bodai 2017, Nonlinearity). The authors refer to Bartsev et al. 2017 to motivate the statement that the climate forcing projection over the leading modes of natural variability is inadequate to explain the nonlinear response of the system, given the inevitable presence of multistable regimes. Bartsev et al. 2017,

though, use conceptual models to demonstrate that multiple stable state possibly exist in a complex system, such as the climate system. Rigorous arguments have to be provided (which is deliberately not the aim of Bartsev et al. 2017), to explain how this applies to the real system. In fact, a survey of available literature would probably convince the authors that, despite the lack of sufficient observations, precise constraints have been provided on to what extent the Hasselmann-type response dominates and when the chaotic nature of the system emerges, leading the system towards critical transitions (cfr. Ghil and Lucarini, 2019, arXiv:1910.00583 for a review).

l. 641: the authors might want to provide references on the land warming leading the oceans (if it's not just a matter of pace of the warming).

ll. 668-675: I do not understand how this mention of the maximum entropy production principle fits the remainder of the discussion in this paragraph.

l. 681-683: the authors seem to suggest that the Pacific heat engine can be treated as a Carnot cycle. Clearly, the Carnot approach only provides an ideal constraint to the efficiency of the heat engine, and this should be clarified here.

Sect. 6.5: this section contains a very long review of a few previous works using network analysis. This includes ll.715-735, where results from Tantet and Dijkstra 2014 are extensively discussed. No original analysis is provided, supporting the consistency between mentioned literature and the results here shown. Given that this is manuscript is submitted as an original research article, I would suggest that the authors either provide an application of the network analysis in this context or remove this part.

l. 769-771: again here, I think that the dichotomy "linear-nonlinear" is unreasonably emphasized, whereas the two frameworks are actually complementary. Therefore, I do not think that there is a lack of vocabulary for the nonlinear context, only because terms like "trends", "rate of change" etc. are extensively used.

l. 838: I am not really sure what the authors mean by "disrupting" the climate system.

Technical corrections

l. 390-392: the straight line the authors refer to in the text is not visible in Figure 5a.

l. 484: I believe that the authors refer here to 'CMIP5'.

l. 514: replace "additional" with "addition"

l. 544: remove "provides".

l. 693: add "be" after "may".

l. 831: remove "shift".

---

## Referee Comment (RC2) · Anonymous Referee #2 · 28 Jan 2020

"The Pacific Ocean heat engine: global climate's regulator" Roger N. Jones and James H. Ricketts

This work builds upon a previous study of the authors that showed that decadal scale warming as monitored by Global surface temperature is not "trend-like" but behaves like a progression of fast warmings ("shifts") and periods of stagnation. The present paper tries to relate this concept to processes related to ENSO and the "Pacific heat engine" by statistical analyses of warm-pool and cold-pool temperature time series, their relation to climate indices (PDO,AMO,AMOC...) in the context of shifts. A conceptual model of the heat engine is presented in a second part of the paper.

While I find the concept very interesting and I appreciate the effort of detailed analyses, I found this paper very difficult to read. It is very long, wordy and not very focused.

[Figure]

The many statistical analyses and detailed descriptions are difficult to follow. How the conclusions made in the text arise from the Figures is mostly difficult to see.

The following changes need to be made before it can go into a more rigorous review of the science:

1) The paper should be substantially shortened. I suggest to leave part 2 out and leave it for a separate paper.

2) Introductory material is mixed-in with the results. Please clearly distinguish between introduction/discussion of previous work and unique results from the present study.

3) I suggest a clear re-structure of the paper: Describe in the introduction in a focused way the starting point, also the state-of-the art with respect the the relation to climate indices etc (a lot of this now is mingled in with the results sections). It should be clearly stated at the end what hypotheses you are investigating or which questions you are addressing. After the introduction, there needs to be a methods and data section where the statistical methods are explained and data described. At present, this is all in the SI, and e.g. the "tracking model" and how it is used is unclear. The methods section needs to lay out how the hypotheses will be tested with the statistical methods. The results sections are overloaded with detailed information but the storyline is not clear. Methods description and results need to be delineated. The text needs to directly refer to the Figure panels so one can track the conclusions drawn.

4) Use present tense when refering to the results. It is very difficult to track whether described results are from the present study or refering to some previous results.

5) There are many unclear and unscientific phrases throughout the paper: e.g.: ... "the heat engine is networked within the climate system" (abstract)... Examples from the intro: ...the process is "regulated" by a heat engine spanning ... ...shifts are "linked to the wider system of climate oscillations"... ...the difference between "this" and the standard model "in terms of energy flow" .. ...how it interacts with "broader climate"...
section 2: ...heat "being channeled and made available for dissipation"... page 11: ..."because of issues with".... page 17: ...the "tightening of the system" coincided with the heat engine moving from free to forced mode... etc. There is a need to go through the paper again and make statements as clear and scientific as possible.

Structural comments:

Figure 1a is something you find in any climate/ocean dynamics textbook. Instead of this rather basic Figure it would be better to redraft something related to the heat engine hypothesis, recharge-discharge theory with focus on your hypothesis.

Section 1: Bullet points 1-6 read like the conclusions section from the Jones and Rickets 2017 paper and are not needed in this way here. Line 77: what is meant by subsequently and elsewhere? Line 84: here, hidden in a side sentence, it is stated for the first time what is the subject of the paper. It is not clear though what is meant by "processes" because I find this paper to be mainly about statistical analyses and not necessarily the underlying processes.

Section 2 and the beginning of section 3 are introduction material. What exactly do you mean by heat engine "behaviour" in the title section 3? Line 158 suddenly results start. Lines 169 ff: too many numbers, difficult to follow. Section 3.1: first sentence: where do you show this? explain tracking model in methods, it is otherwise very difficult to follow section 3.1 Section 4: First sentence: where do you show this? Lots of material and references to previous studies to be moved and discussed in introduction.

Section 5.3: is this needed? It seems like a proper analysis and description of climatological data is enough.

Conclusions: here I suggest to clearly distinguish what are new hypotheses and interpretations rather than conclusions drawn from the statistical analyses in this paper. A clear summary of the statistical results needs to be presented first.

---

## Author Comment (AC1) · 20 Nov 2020

Thank you for these comments, which we greatly appreciate. Regarding specific points:

[1] Identifying energy flows more clearly

This is a good suggestion and our vagueness reflected a general uncertainty as to which types of energy are involved. We could have been more specific about the nature of heat flows, where shifts in temperature are involved. However, responding to referees' comments has made some aspects much clearer. As the system is complex, we need to consider multiple forms of energy exchange that include dry and most atmospheric heat transfer, sea surface temperatures, and atmospheric feedbacks including cloud feedbacks. Some we can describe with relative confidence, others we

cannot, but more precise terminology is needed.

[2] More detailed explanations for TWP and TEP being separate reservoirs rather than locations along a flow tube (our rephrasing).

We are sorry if we have given this impression. Even though the paper describes an east-west heat flow, we were at pains to be quite clear that TWP and TEP are distinct entities. The first diagrams are simplified to provide an overall view, but the reality is more complex.

In the responses to the Referees this has been elaborated on and we now have a much better picture of TEP and TWP as distinct entities. The relationship between the two, moving from a cold to hot reservoir up gradient sets up a tension, because if not for the Coriolis effect, movement would be in the other direction. There is two-way exchange between the two on the surface and subsurface, event though the net heat flow is east-west.

We propose that during forced mode (post 1968) TWP acts a thermostat but carries little memory of the heat cycling through it. TEP acts as the more stable node of the two and Granger-type analyses shows that it is the major source of shifts in temperature. It is possible that the two have been close to a critical state since 1968, if not before.

Therefore, in heat engine terms they may be thought of as reservoirs but they also act as nodes within a broader network. If the global climatic network was considered to have a central node, as TEP is the most obvious candidate. Our Granger analysis suggests that TWP acts as the store and TEP the release.

[3] Positive and negative feedbacks acting by definition as to how these contribute to global warming

We don't agree with this definition and have used positive and negative feedback in the discussion paper as to whether processes are amplified or dampened given a specific forcing. The basic structure of both heat engines and attractors in a thermodynamic

system requires both positive and negative feedbacks.

Changes to the paper(s)

In any revisions, we will take all these points into account, including clarifying [3]. Point [1} will receive particular attention because we consider the other two have been covered under other responses.
* * *

---

## Author Comment (AC2) · 20 Nov 2020

Overall comment

The manuscript describes an approach to the study of the role the tropical Pacific ocean has for the thermodynamics of the climate system, relying on the thermal engine concept. Starting from a previous work by the authors, a rigorous statistical testing is used in order to investigate step-like changes in the observed evolution of sea-surface temperatures (SST) in the Tropical Western Pacific (TWP) and Tropical Eastern Pacific (TEP), relating them with the evolution of global mean surface temperatures (GMST) and some indices describing inter-annual and decadal natural variability (in particular the Pacific Decadal Oscillation, PDO, the Atlantic Multi-decadal Oscillation, AMO, and

[Figure]

ENSO). It is argued that the tropical Pacific can be seen as a thermal engine transporting heat up-gradient (i.e. from a cold to a warm region), and that this mechanism is characterized by two modes: until the second half of the 20th Century a free mode is associated with the natural variability at the decadal timescales, while in more recent years a forced mode is established, in which the natural variability is enhanced by the anthropogenic greenhouse gas (GHG) forcing. The authors argue that these findings provide evidence that the classical view of the response of the system to the GHG forcing as a trend-like behavior with superimposed noise-like natural variability is faulty in describing the transient climate change on a decadal timescale.

Overall, I appreciated that the authors started from a climatological point of view (i.e. from observations) in addressing the challenging issue of how climate change projects on the natural modes of variability at inter-annual to decadal timescales. However, conclusions do not seem supported by sufficient evidence from results, at least the way these have been illustrated.

In the first part, authors argue that step-like changes in TEP and TWP, in part related to changes in the phase of the AMO and PDO, as well as ENSO events, shape the decadal evolution of GMST. The emergence of step-like changes is detected through the usage of established statistical methodologies. On the contrary, the propagation of the signal (cfr. Sect. 3.1) is discussed in terms of timing across different time series, and this qualitative argument severely undermine the robustness of the results. The authors seem to claim that conclusions similar to those based on observational-based datasets can be drawn from investigation of CMIP model outputs. A visual inspection of the model results shown in the supplementary material, though, does not seem to support this conclusion.

In the second part (Sect. 6), arguments are provided to describe the Pacific ocean thermal engine, how its changes propagate to the global climate, and how its internal variability is affected by the forcing through nonlinear mechanisms. This section is deliberately qualitative, and in some parts speculative, and this is of course absolutely

fine. The problem is, that the section has the form of a review of existing literature, particularly in sections 6.2, 6.4 and 6.5, with insufficient or completely absent relation to results described in the first part. These sections alone sum up to almost 40% of the manuscript, making it dispersive and difficult to read. This is not appropriate, in my opinion, for an original research article, and shall be reconsidered.

Overall, even though the results from the first part are potentially interesting and supporting the conclusions given at the end of the manuscript, I think that a more rigorous scientific approach is needed in order to make the manuscript worth publishing. My suggestion is that the authors consider splitting it into two parts. In a first paper (approximately sect. 2-5), the statistical methodology is explicitly accounted (instead of referring to a previous paper), the shifts in TEP-TWP temperatures and their relations with GMST, AMO, PDO, ENSO are explicitly addressed through rigorous methods for causal inference detection. The network approach outlined in sect. 6.5 might also be helpful in this respect. In a second paper (sect. 6), more emphasis is given on the description of the conceptual model of Pacific thermal engine. In doing so, the heat sources and sinks, the working temperatures, the Carnot efficiency and actual efficiency have to be quantitatively addressed, possibly considering the compliance to the 1st and 2nd laws of thermodynamics via energy and entropy budget.

Authors' response.

We thank the reviewer for their careful consideration of the paper and detailed responses. Many we agree with and some we do not – and will carefully try to explain why. We realise it is a lengthy and difficult paper, particularly the second part. We felt, and still do that the behaviour of the heat engine illustrated in part one needs a conceptual explanation, especially as it departs from the standard explanation of gradual atmospheric warming.

A major goal of the paper was to strengthen the case for warming being a non-gradual process, established by severe testing in Jones and Ricketts (2017, JR2017). The dis-
cussion paper had two aims: (1) to present a process and mechanisms that identify where the shifts in temperature originate and how they propagate through the climate system, and (2) to describe a conceptual model or framework for how this works. Referee 1 was not persuaded that climate change cannot be gradual, but an episodic response to forcing is fundamental to the whole hypothesis. We aim to provide a more convincing interpretation of the evidence in the following.

We are grateful that Referee 1 appreciates the focus on observations. The overall approach has been informed by a natural science approach – analysing observations to better understand how the climate behaves as a complex dynamic system while articulating how and why the current understanding cannot explain the results of those analyses. We fell short in doing so. Prompted by the comments and queries of the referees, we aim here to provide a more complete explanation of the conceptual model in addition to tightening up the description of the heat engine and associated network.

Referee 1 suggests that more attention be paid to the quantification of the thermal aspects of the heat engine, considering the 1st and 2nd law of thermodynamics, quantifying Carnot efficiency and actual efficiency. We believe we are dealing with a complex thermodynamic system that is self-organising where dissipation occurs in a fluid-dominated system far from (internal) equilibrium. As this is an inductive process, we need to first understand how it is organised before getting a better idea of the rules that govern the patterns and processes being observed and what the organising principle(s) may be.

At best, we can only provide a general interpretation. With respect to the first law, the overall energy balance of the climate system is influenced by changes in radiative forcing, which acts as a changing boundary condition rather than directly forcing the atmospheric climate. This is because physical constraints prevent the atmosphere warming independently of the ocean. To address the second law with respect to entropy, we would have to solve the issue of what compliance to that law means in terms of entropy in nonequilibrium complex systems, which is currently a work in progress and

unresolved (Endres, 2017; Zia & Schmittmann, 2006). The 2nd law governs the dissipative process that transfers heat vertically and meridionally, the main role of the heat engine. Under forcing, a heat storage and release system in the shallow ocean results in complex system responses producing rapid shifts in climate on decadal timescales.

Without generalisable rules of how the 2nd law governs this process, we first need to work out what the system is doing before being able to figure out how it works. Our aim is to understand the 'what' before the 'how'. Later in the comments we suggest that energy rate density may be the measure that serves as a unifying principle for self-organised behaviour. However, this is very tentative and would need to be tested.

Conventional theory maintains that radiative forcing by greenhouse gases in the atmosphere results in gradual warming mediated by variability. The development of radiative transfer functions linking forcing to temperature change (Hansen et al., 1997; Hansen et al., 2005) produces a monotonic curve. Our position is that the additional heat energy from greenhouse gas forcing is absorbed by the shallow ocean due to its greater thermal conductivity and capacity resulting in strong negative feedbacks. Heat is released in response to thermodynamic forcing as the build-up of heat exceeds the capacity of climate to dissipate that heat. The resulting regime change produce step-like warming that over sufficient timescales form a complex trend (Jones & Ricketts, 2017).

Fig. 1 is a simplified schematic of the major processes involved in the modelling and downscaling of climate information. Most assessments move straight from the first to the third chevron by either considering the second as broadly synonymous with the first and/or as a generator of stochastic variations superimposed on the first. Not shown in Fig. 1 are the feedback processes involved in self-regulation, which feed information back up this process at every level (and are essential parts of any complex system).

Figure 1: Schematic diagram showing the major climate the major processes involved in modelling and downscaling. Not shown are feedback mechanisms moving back up

this sequence.

Thermodynamic forcing is the missing link in the existing analytical structure. Moving from radiative forcing to dynamic distribution generally involves linearisation. Even if stochastic elements are introduced into a time series, the underlying forced response is assumed to be gradual. Linking effective radiative forcing to the climatic response formalises this relationship. The storage of heat in the shallow ocean means that the radiative forcing component becomes indirect and the thermodynamic forcing aspect becomes direct. This treats the climate system as two separate but interacting components as per Ozawa et al. (2003), where the radiative component is linear and additive, and the thermodynamic component involved in the dissipation of heat, is nonlinear.

The role of thermodynamics in contributing to complex behaviour in climate is often acknowledged in principle, but in practice is largely overlooked in favour of dynamics; i.e., the act of subtracting interannual and decadal-scale behaviour from climate time series is to separate the dynamic and forced components of climate. We are proposing that thermodynamic forcing plays a largely unrecognised role in climate change. The mechanisms involved are the subject of the paper: climate acting as a self-regulating complex dynamic system, with the Pacific Ocean heat engine playing a governing role.

Regarding the suggestion to take a more scientific approach by quantifying energy and entropy budgets according to the 1st and 2nd law of thermodynamics. We do not believe that in a complex, dynamic system that working from first principles will bear fruit. Arguably, it has not succeeded to date. An empirical understanding of complex system behaviour is important when diagnosing governing equations and gaining an accurate description of how the system works is our primary goal. We also consider this to be a rigorous scientific approach if accompanied by sufficient testing and review.

This problem is described by Champion et al. (2019): The traditional derivation of governing equations is based on underlying first principles, such as conservation laws and symmetries, or from universal laws, such as gravitation. However, in many modern

systems [read complex], governing equations are unknown or only partially known, and recourse to first-principles derivations is untenable. Instead, many of these systems have rich time-series data due to emerging sensor and measurement technologies (e.g., in biology and climate science). This has given rise to the new paradigm of data-driven model discovery.

Champion et al. (2019) describe a methodology that uses sparse identification of non-linear dynamics to diagnose parsimonious governing equations from data measured in complex dynamic systems. It can also incorporate partially known physics and constraints. Models diagnosed in this way include those containing Lorenz attractors, a reaction diffusion system and nonlinear pendulum (Champion et al., 2019). We believe that to utilise this type of methodology, the basic structure of how the system works needs to be understood first. That is our goal.

Complex system dynamics informed the thinking behind part 2 of the original discussion paper, but this was not openly declared. Instead discussion points were taken from different parts of the literature and the result was scattered and confusing. A revision would introduce the key features of complex dynamic systems such as networks, emergence, state changes, symmetry, attractors and boundary conditions and discuss them in the light of our results. This would be largely descriptive, supported by quantitative methods and tools where available.

For example, introducing the network approach currently in Section 6.5 earlier will provide the context for analyses describing the behaviour of ENSO, PDO and AMO, and free and forced modes. Emergence is also an important issue. Most of the phenomena we deal with are emergent, affecting how causality can be analysed and interpreted. Appendix II discusses some of the arguments as to why emergence is important.

Taking up the suggestion of Referee 1, we used Granger causality methods to analyse temperature changes on monthly and annual timescales for 29 regions covering the hot and cold heat engine nodes, zonal and hemispheric regions to global means

for land-ocean, land and ocean. However, Granger causality cannot in itself identify externally-forced nonlinear change because the test requires a stationary time series. A stationary time series can only describe internally generated influences. However, by comparing stationary with nonstationary results and being able to independently verify the nature of change, we can infer what processes and regions heat is moving through by assessing changes in temperature. The method and some results are described in Appendix 1.

Another important aspect of complex systems are boundary conditions and the manifold, or operating space. In the discussion paper we were unsure of what driving the shifts – an oversupply of heat at the surface, an imbalance in the heat engine itself, or Earth's energy imbalance as measured through the top of the atmosphere energy deficit. We now believe the main role of the heat engine is in governing meridional transport from the equator to the poles, with the geostrophic limits to meridional heat transport imposing very strong boundary conditions for total transport. The heat engine sits within a network that combines various oscillating and circulating systems around the globe acting as nodes via teleconnections. More detailed descriptions are provided in the responses to both referees.

Change to manuscript Both referees suggested dividing the paper into two. The additional work carried out has made this the most appropriate step, which would address their concerns about the focus on the analyses of heat engine behaviour. The first paper would concentrate on the mechanics of the heat engine based on the analysis of observations. The second paper would describe climate as a complex system governed by responses to thermodynamic forcing using the findings from the first paper, incorporating analyses from climate models and more widely from the literature, along with considerations of complex system behaviour.

The following structure is proposed.

Paper 1
1. Introduction 2. Context and general approach 2.1 Previous research and measuring nonlinear change 2.2 Limitations of the standard approach 2.4 Data, methods and tools 2.5 Physical setting 3. Results 3.1 Relationship between TEP, TWP and broader climate 3.2 Tracking changes 1: the heat engine 3.3 Tracking changes 2: teleconnections 3.4 Tracking causality: Granger analysis 4. Discussion and conclusions

Paper 2

1. Introduction 2. Context and general approach 2.1 Climate as a complex dynamic system 2.2 Data, methods and tools 3. The hidden climate 3.1 Emergence 3.2 Representation of the Pacific Ocean heat engine in climate models 3.3 Regime changes in energy transport 4. Complex system behaviour in the coupled atmosphere-ocean 4.1 Boundary conditions (e.g., meridional heat transport) 4.2 Attractors and regime changes 4.3 The heat engine and self-regulation 5 Discussion and conclusions

Note that responses to comments 1 to 31, technical notes and two Appendices are in the attached supplement. We apologise on the length but we have presented some challenges for orthodox approach to understanding how the climate changes and many of the points we make are barely covered in the existing literature.

Please also note the supplement to this comment:
https://esd.copernicus.org/preprints/esd-2019-72/esd-2019-72-AC2-supplement.pdf

[Figure]

Figure 1: Schematic diagram showing the major climate the major processes involved in modelling and downscaling. Not shown are feedback mechanisms moving back up this sequence.

**Fig. 1.**

**Supplement:**

**The Pacific Ocean heat engine: global climate's regulator**

Response to Referee 1  - comments and appendices

Roger N Jones roger.jones@vu.edu.au
James H Ricketts

**Specific comments**

**Comment 1. Line 33**: I do not understand why the authors oppose the approach of emerging trend-like evolution of the forced climate response, to approaches where the non-linear interactions between natural variability and forced response are explicitly taken into account.

**Comment 1. Author's response**
We were unable to convey why we oppose the trend-like evolution of the climate response in the introduction convincingly. Lines 40–76 summarise why we do not endorse the standard approach and are pursuing a different kind of nonlinear interaction. These briefly cover the results of Jones and Ricketts (2017), where we showed that gradual change did not pass severe testing and that the response of climate to radiative forcing on decadal timescales is step-like, or nonlinear. Section 6.1 also outlines why we think the standard approach is physically implausible.

Severe testing, developed by Mayo (1996, 2005, 2018) and (Mayo, 2010; Mayo & Spanos, 2011), works by using probative criteria to show that *'a hypothesis needs to pass tests that its rivals fail with a high probability'*. The addition of probative criteria makes it more powerful than straightforward statistical induction, which climate science relies upon heavily. Severe testing also draws on the philosophies of science and statistics.

Severe testing subjects a hypothesis and its plausible rivals to the same sets of probative criteria. Many studies aim to do this, but the outcomes often depend on initial assumptions. For example, information criteria that prioritise statistical model simplicity will prioritise models that represent linear behaviour against those that represent emergent nonlinear behaviour because of the additional variables required to describe that behaviour. Processes that 'clean' data may remove information about complex system behaviour. However, some complex processes requiring complex statistical descriptions can be described by parsimonious governing equations (e.g., see Champion et al. (2019)). Discounting this possibility may mean complex system behaviour is being overlooked.

Our starting point, based on JR2017 is, if warming, or a large proportion of it is, a gradual response to forcing (i.e., following linear response theory), the six tests conducted in JR2016 (summarised in lines 56–67) could not have been passed.

This is illustrated in the following example. The IPCC concludes that it is *extremely likely* (95–100%) that human activities caused more than half of the observed increase in global average surface temperature from 1951 to 2010 (Stocker et al., 2013, p. 60). The IPCC's conclusion also meets the criteria for severe testing, because the alternatives (solar, internal variability, cosmic rays etc) have been well tested and ruled out. The conclusion is also conservative in order to provide a <5% level of error.

Fig. 2 shows GMST 1880–2018 from the NCDC data used in the paper. Time series are separated into fast (shifts) and slow components of change (internal trends). Shifts are calculated as the distance

between internal trends; i.e., the gap between the dashed lines in Figure 3. Over the period 1951–2010, shifts add up to 0.55 °C.

[Figure]

Figure 2: Anomalies of GMST for the NCDC record 1880–2018 showing internal trends (dashed brown lines) and the simple trend 1951–2010 (dotted line). Anomaly period 1880–1899.

This is compared with three measures of total warming over the same period: the change on the shift and trend process (0.63 °C), a simple trend (0.76 °C) and the difference between five-year averages around 1951 and 2010 (0.64 °C). The proportion allocated to shifts (0.55 °C) ranges from 72% for the simple trend to 86% for the shift and trend process (Table 1). Only one of the internal trends in GMST achieves a $p<0.05$ (1997–2013), so giving internal trends preference over shifts when measuring both penalises shifts.

Table 1: Measurements from Fig. 3.

|  | Start | End | Change | Ratio of shifts to total change |
|---|---|---|---|---|
| Shift & trend | 0.19 °C | 0.82 °C | 0.63 °C | 86% |
| Simple trend | 0.02 °C | 0.80 °C | 0.76 °C | 72% |
| Simple difference | 0.16 °C | 0.81 °C | 0.64 °C | 85% |

We attribute most of the anthropogenic component of this change to shifts, consistent with the conclusions in IPCC (2013). In JR2017, we tested shifts and internal trends against equilibrium climate sensitivity (ECS). Updated data provides 93 model runs with available ECS estimates for the 107 RCP4.5 CMIP5 ensemble members analysed. Total steps 2006–2095 have an $r^2$ of 0.66 with ECS, total shifts 0.53 and total internal trend 0.18. On this basis, shifts are 2.9 times more effective than gradual warming in representing positive feedbacks in climate models. When related to per degree ECS, gradual warming is 26% as effective as rapid warming.

If we transfer those estimates to the warming of 0.63 °C in Table 1 with a ratio of shifts to total change of 86%, a contribution of 0.30 °C can be attributed to shifts and 0.01 °C to trends contributing to atmospheric feedback, 0.18 °C to intermodel uncertainty and 0.14 °C to internal model uncertainty, largely stochastic behaviour. In the models, the shift to trend ratio is uncorrelated with ECS, reflecting stochastic uncertainty. The more trend-like model runs therefore do not exhibit a stronger relationship between internal trends and ECS.

Part of the intermodel uncertainty is due to model structure. For example, the lower ECS values are dominated by two ensembles from the GISS laboratory that have different ocean schemes and shifttrend ratios (JR2017). The higher ECS values are represented by a greater variety of different models. We discuss potential model bias further in the response to Comment 24.

During periods when the heat content of the ocean is increasing while SST remains in steady state, there will be no atmospheric feedback. When thermodynamic forcing produces step changes in SST occur, the atmosphere overhead also warms, with immediate positive feedback. The observational record shows that step changes in SST ranging from 0.15 °C to 0.30 °C are shortly followed by step changes of 0.25 to 0.60 °C over land. This explains the strong relationship between shifts and ECS in the climate models.

If we accept the 21$^{st}$ century relationship between fast and slow warming and ECS in the models, and apply that relationship to observations, a trend-like response on decadal timescales cannot explain the observed changes.

**Comment 1. Change to manuscript**
An introduction and previous research subsection that address these points and explain the findings of JR2017 that are relevant to this paper will be prepared (see table of contents above).

**Comment 2. Lines 33–34.** As found in the IPCC 5th AR, at sect. 11.3.1.1 (I suggest that, when possible, the authors refer to the original publication, rather than the chapter of the assessment report where the information is taken from, e.g. Hawkins and Sutton 2009, in this case), "The evolution of the S/N ratio with lead time depends on whether the signal grows more rapidly than the noise, or vice versa". This does not rule out the possibility that the noise itself is affected by the forcing; it just says that, if the forced signal grows faster than any other change due to the internal variability of the system, the latter is treated as noise. The rationale behind the S/N approach is not to be found in the radiative-convective model, rather in the fact that the near-surface layers of the atmosphere, and particularly the land surface, are heated over timescales comparable to the inter-annual and decadal timescales of the natural variability. Furthermore, one may add that the forced response should be treated as a trend-like response specifically in the case of climate models, where the internal variability (which is related, but is not coincident, in the model world, to the concept of actual natural variability of the system) is (almost) insensitive to the forcing over a wide range of timescales. In addition to that, I believe that it is nowadays commonly accepted that the regional forced response is unavoidably affected by changes in the statistics of the natural variability due to the forcing, so that the trend-like approach alone is not a viable option.

**Comment 2. Authors' response**
Thank you for the comment. This comment contains many of the points associated with the current paradigm of how climate changes. We see these as covering the cognitive values the support the paradigm, the methods that express it and the theory that informs it. In this and the following responses we will cover some of the history and philosophy of climatology as a branch of the earth sciences and its contribution to theories of gradual change, the role of the IPCC in endorsing methods that flow through to the characterisation of risk and touch on some of the evidence to the contrary (more evidence is presented in subsequent responses).

The reason for citing the IPCC directly is because it guides latest practice. The relationship between theory, methods and scientific values and how that affects the take-up of new knowledge or ways of doing things is often underestimated (This is discussed in detail by Laudan (1984)).

Historical contingency is a major influence on the long-standing preference for the model of trend-like change. Irrespective of the theoretical positions taken in AR5 and in the broader literature, the practice of measuring trends dates back to the scientific enlightenment, preceding the development

of climate theory by over a century. The historical role of trend analysis and the signal-to-noise model, and scientific values attached to them, is discussed in Jones (2015). This is summarised and updated here.

The notion of a mechanical universe subject to regular change is a cosmological value (Baker, 1998). When Laplace and Gauss introduced statistical methods to measure change via least squares linear analysis, this became the accepted norm because it matched existing values (Laplace's System of the World (1830) makes this clear) (Stigler, 1986). At the same time, the earth sciences saw an intellectual battle between the French catastrophists and the English, culminating in Lyell championing gradual change through uniformitarianism (Baker, 1998; Rudwick, 2005, 2010). The English establishment saw off the challenge of the catastrophists by linking them with biblical catastrophism, a literal reading of the bible, therefore deemed unscientific (Baker, 1998). Their cosmological position of the universe as a regulated and rational clockwork mechanism handed down from Newton, was religious but involved design rather than divine intervention (Baker, 2014).

Gradual change has underpinned statistical climatology since, even as statistical methods have become more sophisticated. This cosmology removed nonlinearity from the natural sciences until disciplines such as physics, chemistry, ecology and punctuated equilibrium in biological evolution brought it back. In climatology, much of palaeoclimatic change is now widely considered to be dominated by nonlinear responses to gradual forcing (e.g., Milankovitch cycles), and nonlinearity associated with global climate regimes such as snowball and hothouse earth (Ghil & Lucarini, 2020; Kypke et al., 2020; Lucarini et al., 2010; Peltier et al., 2004).

Trend analysis dominated climate change analysis throughout the 19[th] and 20[th] centuries. For example, during the dust bowl droughts and warm conditions of the 1930s in the USA trend analysis was used to ascertain whether the climate had changed (Kincer, 1937). Reading Kincer's 1937 article "Has the Climate Changed?" is reminiscent of articles on climate being written by climatologists in the 1980s and 1990s, when the question of attributing warming to anthropogenic causes was in open debate.

The signal to noise label comes from detecting radio and later radar signals, which provides cognitive values that are widely applied. The signal is considered to contain useful information and noise is to be filtered out and discarded. This term was picked up by the climate community during the mid-20[th] century, accelerating after 1970. The value weighting between signal and noise has been transferred to the policy distinction between anthropogenically-forced change in the atmosphere and natural variability as articulated in Article 4 of the UN Framework Convention on Climate Change. The anthropogenic signal is of policy value, whereas the accompanying noise of natural origin is not. This has encouraged the scientific community to pursue the separation between the two.

In terms of theoretical development, the temperature response to radiative forcing (radiative transfer) in the atmosphere is often traced back to Arrhenius (1896) who estimated the amount of near surface warming heating the ground due to the measured atmospheric carbonic content (Tyndall and Foote's earlier calculations can also be converted int his way). Despite significant advances in the theory behind radiative forcing, it is often remarked how the core theory advanced by Arrhenius, Callendar and others has remained robust (Anderson et al., 2016), even though Arrhenius got the right result from the wrong numbers (Easterbrook et al., 2018). These theories and early work on radiative-convective schemes reference the land surface, either explicitly or implicitly. Early measurements taken by scientists were on or over land, either measured directly or by balloon. Despite almost 70% of the earth's surface being ocean, even today the sea surface is rarely mentioned in papers discussing radiative transfer and atmospheric heating.

As climate models developed, atmosphere-ocean energy exchange gradually became more sophisticated until coupled ocean-atmosphere schemes became available. The models eventually provided better coverage than observations. Ease of analysis is a factor influencing model use – if data from models is more easily analysable than observations, then the models become more influential. If models are biased to observations, those biases may be introduced into practice.

The earliest climate models represent climate change as a gradual process and have become the simple models of today. As models became more sophisticated, climate variability and features such as ENSO emerged (see Appendix 2). However, the methods used to analyse the outputs of these models continued to be dominated by trend analysis, even if those analyses became more sophisticated (e.g., empirical orthogonal functions and pattern analyses).

Although statistical climatology has always been dominated by linear methods, during the mid-20[th] century there was considerable debate on the fundamental theory of how climate changed. Whether the climatic response to forcing was linear or nonlinear was a key topic of discussion at the conference *The Physical Basis of Climate and Climate Modelling*, where both linear (Leith, 1975a) and nonlinear (Lorenz, 1975) theories were discussed. These discussions continued over the next few decades. The linear response theory was further developed by Hasselmann (1976) who introduced stochastic elements, receiving widespread uptake. We discuss this further in response to later comments. The literature continues to reflect interest in nonlinear responses – "abrupt climate change" as a phrase currently attracts 24,200 mentions on Google Scholar whereas "gradual climate change" has 1,820. "Abrupt" and "gradual" "climate change" attract 228,000 and 408,000 hits, respectively.

However, when comparing potential risk and practice the weightings given to linear and nonlinear change in are very different. Even though scientists' views range from those who see radiative forcing as being gradual and monotonic to those who see climate as being complex, nonlinear and unpredictable, in practice, almost all numerical projections provided to end-users are based on trend-like change. This is despite significant research interest in abrupt change and tipping points (Alley et al., 2003; Bathiany et al., 2018; Drijfhout et al., 2015; Lenton et al., 2008).

IPCC findings on both theory and practice and how they relate to each other, we therefore consider to be critical. They have major influence on current scientific practice, and their endorsement of linear change over decadal timescales. The IPCC is hampered in how it deals with the history of science because it is instructed to concentrate on references produced since the last report, and authors may not always be in a position to address the full historical legacy of issues they are assessing.

Hawkins and Sutton (2009) were not the originators of the signal to noise model, the use of which has arisen largely due to historical legacies and the scientific values that support them. The papers by Hawkins and Sutton (2009, 2011); Hawkins and Sutton (2012) and similar works are methodological, seeking to narrow uncertainties around monotonic projections from model ensembles – their identification of the window of 30–50 years where the linear signal outweighs the noise of variability and the policy uncertainty of different decisions affecting emissions combine with climate sensitivity uncertainties are at a minimum does exactly this (Hawkins & Sutton, 2012). Using transient sensitivity constricts this range of uncertainty even more.

The comment that the forced response from models should be treated as trend-like seems to refer to the principle behind pattern scaling regional change per degree of global warming, which is linear in most instances (Mitchell, 2003; Santer et al., 1994; Whetton et al., 2012), making them suitable for exploring a wide range of model and policy uncertainties. If variability is insensitive to forcing

(and it is often not) then probability distribution functions (pdfs) are also transferrable between observations and projections, with little further need for downscaling. However, this is a methodological statement.

If these methods mischaracterise risk in any meaningful way, they should be treated with caution. The framing of climate change as a gradually evolving risk is critiqued in Jones et al. (2013), where methods to adapt to rapid change are explored.

The application of p-values to the signal to noise model, where the signal is considered to be confirmed above a given threshold (e.g., p<0.05 for null hypothesis) is a form of behavioural statistical induction (Mayo, 1996), or mechanical reasoning (Gigerenzer & Marewski, 2015). Referee 1 writes "if the forced signal grows faster than any other change due to the internal variability of the system, the latter is treated as noise". That is not our understanding as to how this is applied in practice. Most of the time classifying the variability of the system as noise is a starting assumption.

As part of our research portfolio involves characterising risk, we feel the emphasis on these methods gives the wrong message. The largest risks on these timescales (and shorter) are from rapid shifts in climate regimes affecting the magnitude and frequency of extremes. Rapid transitions between regimes have serious implications for managing climate-related risk, where regional risk profiles can change abruptly (Ebi & Hess, 2017; Jones et al., 2013). In the same way that economists were unprepared for the GFC, these smoothed-over projections do not prepare strategic planners for the rapid shifts that will eventuate, albeit in the other direction.

For example, the 2019–20 fires in Australia and those in the seasons preceding it are much more severe than predicted by models. Our analysis shows that there was a regime shift in fire climates in SE Australia in 2002–03 that was thermodynamically driven, marked by a shift in relative humidity affecting southern hemisphere (SH) land. This produced shifts in relative humidity and fire season maximum temperature that shifted the forest fire danger index by 22% for the annual index and 52% by days of severe to catastrophic fire danger. The size of the shift is close to the upper limit of projections handed down in 2015 for 2030, but came three decades early (Fig. 3) (Jones and Ricketts 2020, unpublished paper).

[Figure]

Figure 3: ΣFFDI anomalies for SE Australia set to the baseline 1980–81 to 2009–10 (CCIA2015 offset by six months, magenta dashed line) showing the baseline 1972–73 to 2006–07 (Lucas07, brown dashed line) and most recent regime (dark blue dashed line), shown with the 2030 projected range of change (mid grey lines) and 2090 (dark grey lines). FFDI – forest fire danger index.

These changes are linked to regime shifts in the tropical Pacific in 1997–98 providing the heat, and following a succession of wetter La Niña years 1998–2001, sustained drier conditions leading to a downward shift in relative humidity over southern hemisphere land (Based on an analysis of HadISH

(Willett et al., 2014)). Climate models seem to be limited in their ability to produce the same degree of tropical expansion as already seen in observations (Garfinkel et al., 2015; Mantsis et al., 2017; Staten et al., 2018). Some studies interpret observed changes as partly being due to climate variability because their nonlinear responses coincide with decadal variations in tropical SSTs (Mantsis et al., 2017; Staten et al., 2018). Rather than being due to climate variability superimposed on a trend, we interpret it as nonlinear thermodynamic forcing superimposed on decadal variability resulting in shifts in steady-state regimes.

The IPCC's conclusion from the AR5 Technical Summary, cited in the first paragraph of our introduction, is the first time IPCC Working Group I has explicitly endorsed a linear response to forcing over decadal time scales. This is an important conclusion that supports and strengthens the status quo.

Because of the strong methodological support for linearising the warming signal, hypotheses that integrate non-linear elements with gradual change will be more palatable than those that reject it outright as we do. Arguing for a non-standard theoretical or methodological approach is also made difficult by current definitions. In AR5, trend is defined as 'a change, generally monotonic in time, in the value of a variable'. This is non-technical and flexible, allowing the term to be used in different ways ranging from technical to colloquial. This is consistent with how trend has been used historically (in climatology and more generally when measuring any change in natural or human systems). Using the term in a variety of ways normalises linear change because it passes from scholarly usage to everyday language. While this may have been appropriate when natural systems were considered to be linear by default, it is no longer the case.

Referee 1 also commented that the warming process is not found in the radiative-convective model but in near surface warming. However, the description given by Manabe (2019) of the radiative-convective model is consistent with this description:

> *The first process involves the increase in the downward flux of longwave radiation*
> *that increases the temperature of the Earth's surface. Over a sufficiently long*
> *period of time, the surface returns to the overlying troposphere practically all the*
> *radiative energy it receives, with thermal energy being transferred upward*
> *through moist and dry convection, longwave radiation, and large-scale circulation*
> *in the atmosphere. Thus, temperature increases not only at the Earth's surface*
> *but also in the overlying troposphere.*

Manabe (2019) describes how this process is represented in a series of complex to simple models, using a simple radiative-convective model as an example. The technique of using models of decreasing complexity to explain physical processes, first described in detail by Schneider and Dickinson (1974) breaks down with emergent processes when simpler models fall below the level of emergence. Purpose-built idealised models, such as simple ENSO models that can reproduce the emergent process, are then needed. Limited work has been carried out on decadal variability in this regard, with most of the recent examples focusing on prediction (See Appendix 2).

We disagree with the comment that near surface heating occurs directly over timescales comparable to the inter-annual and decadal timescales of the natural variability. Heat is trapped in the lower atmosphere but is rapidly absorbed. In ocean regions, heat content is surface controlled by the ocean's ability to maintain steady state. The heat flux over the ocean is overwhelmingly negative (Frankignoul et al., 2004; Park et al., 2005), so the only way that atmosphere can warm gradually is if the ocean surface warms gradually, and it does not. Temporary heating will occur over land but once the atmosphere circulates over the ocean, it is brought back into steady-state. Instead, we propose that warming principally occurs via a storage and release mechanism governed by coupled

interactions between the atmosphere and ocean. There may be some trend-like components within time series, but these are minor when compared to the nonlinear components in terms of how they influence climate and associated climate-related risks.

These arguments were contained in the discussion paper, mainly at the beginning of part 2 (Section 6.1), but we plan to make this quite clear in the introduction and a context-setting section. The overall context of the two papers would will focus on complex system dynamics. As shown in Fig. 1 we propose that the current understanding of cause and effect in climate change is missing an important step that involves thermodynamic forcing resulting in nonlinear dissipative processes. The Pacific Ocean heat engine is put forward as the key mechanism governing this.

**Comment 2. Change to manuscript**
See the changes to the table of contents proposed above intended to clarify our approach the scope of the paper(s).

**Comment 3. Lines 41-42:** showing that a signal responding to a monotonic trend has a not strictly monotonic behavior does not necessarily imply that there is some sort of interaction between the forcing and the internal variability of the system, contrary to what seems to be suggested here.

**Comment 3. Authors' response**
We do not quite understand this comment – the subject of the sentence is that variability and change interact, not that the lack of monotonic behaviour means that some kind of interaction has occurred. Our position is more sophisticated than that. Perhaps if we said (on line 40) 'an alternative position' or 'our position' instead of 'the alternative position' if the definite article is the issue. The next line 'How this could manifest has been the subject of much speculation, given the many possible outcomes' refers to the uncertainty as to how interaction between forcing and internal variability could manifest.

Given both referees' comments on this first section, we have not communicated the critical point, which is that our previous work shows that the majority of change cannot be trend-like. From Fig. 1 we argue that there is a missing step of nonlinear thermodynamic forcing that follows radiative forcing and governs the dissipative process.

**Comment 3. Change to manuscript**
These points would be covered in a rewritten introduction and section on previous research that does not repeat the findings of JR2017 but interprets them in the context of these papers.

**Comment 4. Lines 63-64**: is this result from the 2017 paper relevant in this context? Does it relate to the results shown in this manuscript?

**Comment 4. Authors' response**
This comment combined with Comment 3, shows we have not managed to convey the starting point for the paper well enough. A context section will outline the previous work and how it informs the view that warming cannot take place in a monotonic trend, and that it is dominated by shifts on decadal timescales.

**Comment 4. Change to manuscript**
As for comment 3.

**Comment 5. Line 61:** it is not clear to a reader who is not aware of the 2017 paper, what "explanatory power" means, especially given that a measure for that seems to be implicitly adopted here;

**Comment 5. Authors' response**
We agree this was unclear – too much shorthand. We have changed how the previous research would be presented to better illustrate key findings relevant to this paper. This point highlights the strength of the relationship between shifts and internal trends in explaining the relationship between global mean surface temperature (GMST) and equilibrium climate sensitivity (ECS) in the CMIP5 RCP4.5 climate model ensemble. It is also discussed in the Comment 1 response.

**Comment 5. Change to manuscript**
Example text: For RCP4.5-forced warming 2006–2095, regressions between the total shift and total trend components and equilibrium climate sensitivity (ECS, n=93) showed that total shifts ($r^2$ 0.53) had 2.9 times the explanatory power than total trends ($r^2$ 0.18). The total GMST-ECS regression 2006–2095 has an $r^2$ value of 0.66, so shifts explain most of this relationship.

**Comment 6. Lines 63-64:** is this result from the 2017 paper relevant in this context? Does it relate to the results shown in this manuscript?

**Comment 6. Authors' response**
Sorry, this was not clear. In these three regions, attribution showed that the change from stationarity to nonstationarity occurred abruptly. The timing differs for each region but is consistent with the shift from free to forced mode in the heat engine and broader climate discussed later. We have now collected more analyses relating to the changes from free to forced, so describing these will strengthen the paper.

**Comment 6. Change to manuscript**
Example text.
Regional attribution carried out for SE Australia Central England and Texas, showed an abrupt change from stationary to nonstationary conditions in 1968, 1989 and 1990, respectively. This is counter to the gradual emergence of the signal-to-noise model narrative (Hawkins & Sutton, 2012; Mahlstein et al., 2011; Santer et al., 2011). An earlier nonstationary shift to warmer conditions was detected for Central England temperature in 1911 or 1920, depending on the length of the record used. This may have been due to combination radiative and solar forcing over the NH (Egorova et al., 2018; Hegerl et al., 2018; Wang et al., 2018).

**Comment 7. Lines 81-82:** in my opinion it is a bit misleading to look at the Earth's energy imbalance (EEI; I think this is what the authors are referring to here) as an atmospheric energy deficit. The role of the atmosphere is determining the rate of absorption/ scattering/emission of solar/infrared radiation through its chemical composition. The atmosphere has a very small thermal inertia, limiting the storage of radiative heat in its interior to (less than) 1%, whereas all the remaining EEI goes into the ocean, warming the surface, melting sea-ice and glaciers (cfr. Von Schuckmann et al. 2016).

**Comment 7. Authors' response**
Thanks for the comment. This is referring to the EEI and perhaps should have been written as top-of-the-atmosphere energy deficit, with dashes to make it clear. Physically, it is the surface to top-of-the-atmosphere deficit even though it is referred to as Earth's energy imbalance. This comment refers to amount of heat that is assumed to remain in the atmosphere (roughly 1% of the EEI or toa

energy deficit) when the remainder not taken up by land, land surface processes and melting ice and snow is taken up by the ocean.

A recent estimate is 0.9±0.3 (Trenberth et al., 2016) with non-ocean climate system components estimates of 0.04 W m$^{-2}$ for 2004–08 (Trenberth et al., 2009), 0.07 W m$^{-2}$ in the 2000s (Trenberth et al., 2014) increasing from about 0.03 W m$^{-2}$ in the 1990s (Hansen et al., 2011), and 0.03 W m$^{-2}$ from 1993–2008 (Church et al., 2011).

In pointing out the small amount of heat retained by the atmosphere, we are highlighting the physical incongruity between heat uptake into the ocean, which has much greater thermal conductivity and capacity than the atmosphere, when large heat anomalies produced by ENSO face no physical barrier to being reabsorbed.

Recall Lorenz' butterfly that flaps its wings and causes a tornado. This is the opposite – expecting the butterfly to hover in the same place during a tornado. On lines 546–550 we refer to anomalies of >1 W m$^{-2}$ due to ENSO events being returned to mean conditions within months. Expecting the atmosphere to retain this tiny amount of residual energy while the surface is in steady state is physically unrealistic.

We agree with von Schuckmann et al. (2016) that having a good understanding of Earth's energy imbalance, especially over the historical period, is needed. At the time of writing we thought there may be a threshold level of EEI that sees a switch in the heat engine and climate network from free to forced mode. At the time it was not clear to us whether the issue is energy oversupply at the surface, top of the atmosphere demand or a dynamic relationship between both with the latter seeming most likely. However, further work identifies a surplus in energy at the surface as the most likely factor. We go into more detail in response to later comments.

**Comment 7. Change to manuscript**
The above fluxes and overall concept would be introduced Section 2 of paper 1 and addressed in more detail the proposed Section 3.3 of paper 2. Although we did not cite von Schuckmann et al. (2016) in the discussion paper we did cite some of the source papers. It will be cited in any revision. Paper 2 will also review and discuss the literature on meridional heat transport, which plays an important role.

**Comment 8 Lines 97-98:** I do not see which literature reference introduces this definition and, more importantly, which evidence is behind it. Certainly, the paper by Kjellson et al. 2014 does not address this definition.

**Comment 8. Authors' response**
Thank you. We agree this was unclear and the citation from Kjellsson misplaced.

To our knowledge, there is no single origin for referring to the warm pool as a heat engine. An early reference was in Simpson (1988) in the Report of the Science Steering Group for a Tropical Rainfall Measuring Mission, who referred to the tropical system as the firebox of the atmospheric heat engine via cloud clusters and implicitly with the warm pool as the largest source of these. This wording may have come from an earlier article on the 1982–83 El Niño in Weatherwise (Rasmusson & Hall, 1983). Rasmusson was a member of science steering group in the 1988 report. Similar references to tropical convection as a heat engine were also made during the TOGA program investigating tropical rainfall in the warm pool region. Webster (1994) likewise uses it for the whole climate, while paying particular attention to the role of the warm pool. This terminology gradually moved to referring to the warm pool specifically.

The palaeoclimate community also identified the long-lasting role of the warm pool somewhat independently, mostly through coral and ocean core analysis. An abstract by Quinn et al. (2002) was the first specific reference we could identify with the full paper published in 2006 (Quinn et al., 2006). The climatic and palaeoclimatic importance of the warm pool as a source of energy was progressively recognised, and by the mid-2000s, it was being specifically identified as a thermal heat engine. The paper by Kjellsson et al. (2014) does not address this directly, so we will remove it from the list, however they describe results that are consistent with recent results on the changing nature of circulation that we will cite.

**Comment 8. Change to manuscript**
We are not sure whether this history is necessary other than to briefly state that the terminology and specific identification of the warm pool as the global heat engine has evolved over time, with contributions from references already cited and addition of several from above.

**Comment 9. Lines 117-118**: this up-gradient flow of heat might well be compensated by subsurface return flow, to my understanding. If this is the case (although I have no reference in mind for that), does it make sense to consider the upper part of the ocean as a closed system, given also its heat exchanges with the deep layers in the subsidence and uplift regions?

**Comment 9. Authors' response**
This is the case; there are several return flows. The north and south Pacific equatorial currents and subsurface currents about which little is known (Li et al., 2020). East-west exchange also varies on a seasonal basis and with ENSO events. There are also meridional return flows, mainly in the NH, which have a longer circulation time.

We don't consider the system to be closed but it is governed by boundary conditions. For energy exchange with the rest of the climate system, heat energy above the thermocline is the most active component. Slower meridional circulation occurs but we are interested in the most active thermodynamic response. We have conducted analyses in temperature and heat content of the top 100 m of ocean and this shows similar regime-like behaviour as for SST. However, with greater depth, the more gradual warming becomes, and is mostly trend-like below 700 m. The rate of loss to the deeper ocean is largely concentrated to specific regions of subsidence and the rate from the tropical Pacific is mediated by the PDO (Balmaseda et al., 2013; Chen & Tung, 2014). The exchange between the shallow and deeper ocean mainly affects the frequency of shifts, with less heat available at the surface, so the time gaps tend to be longer during negative PDO phases.

Our analyses of monthly and interannual interactions suggest the TEP-TWP link has become more tightly coupled and less connected by direct circulation to the rest of the climate system over time (and more teleconnected). As part of a network it also appears to be multiscalar – exhibiting interactions on weekly to multi-yearly timescales. Further analysis of correlation on monthly and annual timescales and Granger causality on monthly and annual timescales indicates that the link between TWP and TEP has strengthened and appears to be close to critical limits. Not included in the paper, we also conducted running correlations on satellite sea level data between Indonesia and Niño 3.4. Data was collected at roughly weekly intervals and running correlations of 23 data points (covering about three months) for +1, real time and -1 lags shows random switching between leads and lags up to several times in a single year. Correlations also vary between positive and negative. These are more regular and sustained during major ENSO events, when large amounts of water are transported between TWP and TEP (e.g., Peyser et al., 2016).

On the timescales we are working with, the well-mixed later of the upper ocean coupled with the atmosphere is the most important area of energy exchange. The connection to slower modes of circulation via the Southern Ocean and within the Pacific itself probably form boundary conditions that keep the system stable via negative feedbacks. This is supported by the capacity of the climate system to switch between fast and slow modes of dissipation in order to maintain meridional heat transport.

**Comment 9. Change to manuscript**
With the additional analyses carried out and recent literature, we can improve the description of the heat engine structure in the proposed context section 2.5. The proposed section 3.1 describes the evolving relationships between TWP, TEP and the broader climate network. The results will mainly focus on correlation, regression including Granger causality and the analysis of short-term noise (white noise and autocorrelation). Section 3.4 will have a more comprehensive description of heat engine structure, identifying the remaining uncertainties.

**Comment 10. Line 124**: I do not agree with the statement that internally the system is not in equilibrium with incoming radiation from the Sun (if I interpreted correctly what the authors mean here). Climate is not in thermodynamic equilibrium with its surroundings, rather it is in a statistically steady state, meaning that, in the absence of any forcing, the net energy input equals the net energy output over sufficiently long timescales. More appropriately, it can be said that the system is a non-equilibrium dissipative steady-state system. The atmosphere is on average in energetic balance with its surroundings, and same can be said for the oceans, even though the time needed to achieve such a balance greatly varies depending on the subsystem that is taken into consideration.

**Comment 10. Authors' response**
Thank you. This is not what we were saying and we will rephrase, omitting any reference in the text to equilibrium of the whole climate system to incoming radiation. (It probably never completely is due to time adjustment in slower processes such as the Milankovitch cycle; e.g., Bathiany et al. (2016)).

**Comment 10. Change to manuscript**
Suggested text: Internally, Earth's climate is in thermodynamic disequilibrium (Kleidon, 2009, 2012; Leith, 1975b; Trenberth & Hurrell, 1994). The tropics are a region of energy gain and the high latitudes a region of loss, so heat moves from the equator to the poles on a continual basis. Heat engines within the climate system are involved in dissipating this energy. They can be stable, periodic, non-periodic or event-based (e.g., tropical storm systems). Two major stable heat engines on each side of the central Pacific are the north and south Hadley Cells.

**Comment 11. Lines 136-139:** I think that the authors might discuss to what extent the efficiency in the meridional heat transport is maximised in the climate system. One can have meridional transports simply as a consequence of the differential in diabatic heating (cfr. Lucarini et al. 2011, JAS), but energy can be transported (and transformed) in many different ways, and the moisture is a critical feature, in this respect (cfr. Yang et al. 2014, Clim. Dyn.).

**Comment 11. Authors' response**
Thank you for this. Meridional heat transport (MHT) was largely overlooked in the paper – mentioned but not assessed. As pointed out it plays an important role as a major boundary condition and latent heat transport is the key vector in anthropogenic climate change.

Geostrophic controls on MHT provide an important boundary condition for the dissipation of heat from the equator to the poles. Both vertical and horizontal dissipation occur in the atmosphere;

vertical dissipation contributes to horizontal dissipation through mechanisms such as Hadley Cells and along the pressure gradient from the equator to poles (Wallace et al., 2015). Vertical dissipation is the larger component but when the two are divided and analysed separately they were positively correlated in the CMIP3 climate model archive (Lucarini et al., 2011).

A range of studies show the energetic constraint provides similar estimates to those gained from dynamic analyses, doing so by relating top of the atmosphere and surface energy without needing to know the specific atmospheric motions in between. This property of Earth-Sun geometry was first identified by Stone (1978) for total ocean and atmosphere heat transport providing a constraint where ocean and atmospheric transport must compensate for each other, as first proposed by Bjerknes (1964). It can also locate the latitudes of maxima and minima in transport.

Trenberth and Stepaniak (2003a, 2003b) examined the budgets and atmospheric transports of moist and dry heat from the NCAR-NCEP reanalysis 1979–2001. Trenberth and Stepaniak (2003a) describe the distribution of moist and dry heat from the equator to the poles while Trenberth and Stepaniak (2003b) examine the transports. Based on top of the atmosphere measurements, they divided heat into quasi-stationary and transient, and dry static, latent and kinetic heat energy. The latter is fairly small. Total energy transport from pole to pole produces a smooth sinusoidal curve but the contribution of each varies both spatially and temporally (Fig. 4). The interannual variability of both dry static and latent energy is much greater for each than the variability of the two combined. This led to the authors calling the relationship 'remarkably seamless' (Trenberth & Stepaniak, 2003a, 2003b). When ocean meridional transport is added, the relationship is even more smoothly sinusoidal (Fig. 4)(Fasullo & Trenberth, 2008; Yang et al., 2015).

Total energy divergence is at a minimum (slightly positive) near the equator and positive either side, becoming negative above 40° (negative divergence is convergence, or inward transport). This combines positive dry static energy and negative latent energy. Divergence of dry static energy in the tropics is positive due to the intense convection and release of latent energy in rainfall, so the descending arm of the Hadley Cell is dominated by dry static energy. Air descending over the ocean results in positive evaporation/precipitation ratios, and much of that water vapour is returned to the tropics. As a result, the region from about 15–40° is a zone of dry energy convergence and moist energy divergence. Latent energy divergence is larger in the SH because of the greater expanse of ocean available for evaporation. The high latitudes are regions of convergence for both dry and moist heat.

Armour et al. (2019) investigated meridional atmospheric heat transport (AHT) from (1) a dynamic perspective, assessing the transport of moist static energy from the equator poleward via circulation structure; (2) in an energetic sense, linking AHT to the relationship between surface and top of the atmosphere energy fluxes; and (3) in a diffusive sense, in terms of downgradient energy transport. They describe the dynamic perspective as being the result of circulation starting in the tropics, progressing through different meridional structures, beginning with stationary eddies and moving to transient eddies with very different structures that produce a smooth outcome. Although it explains atmospheric circulation, largely through geostrophic forces, it is energetically unconstrained (Armour et al., 2019).

[Figure]

Figure 4: Upper: The northward transport of heat as a function of latitude and its major components. The total heat flux is in black; Schmitt (2018) adapted from Yang et al. (2015). Lower: Estimates of zonally integrated meridional freshwater transport (1 Sv = $1 \times 10^6$ m$^{-3}$ s$^{-1}$) in the atmosphere and the ocean (Schmitt, 2018).

Armour et al. (2019) assessed each of the above perspectives using a CMIP5 model ensemble comparing a pre-industrial baseline, a time slice 100 years into an abrupt 4 x $CO_2$ experiment and an equilibrium mixed layer model experiment. Donohoe et al. (2020) compared the extremes between the last glacial maximum and abrupt 4 x $CO_2$. The effect of the latter on total meridional heat transport (MHT) is shown in Fig. 5. This shows marginal changes on MHT, amounting to 2% of the total for the difference between 4 x $CO_2$ and pre-industrial in an ensemble of 7 models (Donohoe et al., 2020).

Armour et al. (2019) modelled dry and moist diffusion separately, finding they both produce the broad structure of AHT, but the diffusion of moist static heat produces more realistic patterns of change, especially of polar amplification. An increase in heat at the equator increases the meridional gradient. The diffusion of dry static heat results in a small decrease in temperature gradient while moist static energy produces a large decrease, resulting in polar amplification of temperature.

[Figure]

Figure 5: (a) Ensemble-averaged total (ocean plus atmosphere) annual-mean meridional heat transport in preindustrial (PI, black), abrupt carbon dioxide quadrupling (4 × CO$_2$; red), and Last Glacial Maximum (LGM; blue) CMIP5 simulations (n=7). (b) The changes in MHT between the 4 × CO$_2$ and PI simulations (red), and between the LGM and PI simulations (blue) (Donohoe et al., 2020).

Changes to AHT are largely independent of internal climate dynamics. Instead, broadly similar patterns of forcing, feedback and heat uptake produce the meridional structure of AHT with very different spatial patterns of warming (Armour et al., 2019; Donohoe et al., 2020). The meridional structure of AHT is strongly constrained by energy and mediated by the diffusion of moist static energy, via surface energy fluxes. This imposes strong boundary conditions that set tight constraints for models, and presumably, the real world. Armour et al. (2019) conclude that the tropical to polar diffusion of latent energy is the defining process for changes in AHT, but are unclear as to what role atmospheric dynamics plays. For example, transient eddies can be invoked to explain extratropic change, but have a limited contribution in the deep tropics.

Like the majority of researchers, Armour et al. (2019) see warming as occurring directly in the atmosphere, the rate being mediated by ocean heat uptake. If we turn this around and invoke the heat storage and release model governed by the Pacific Ocean heat engine, temperature shifts are a nonlinear thermodynamic response to meridional energy imbalance. There is still a role for in situ and rapid adjustments to radiative forcing within the atmosphere (see Smith et al., 2018 for details), but the amount of energy involved is secondary to that involved in the storage and release process.

The increasing imbalance in the meridional heat gradient results in heat stored in the shallow ocean being released. The energy limits and compensatory nature of meridional heat transport, where oceanic and wet and dry atmospheric transport substitute for each other provides a robust manifold for the thermodynamic forcing of the dissipative process. Dynamic constraints include stationary dissipative processes such as the Hadley and Walker Cells, which are relatively fixed but can expand, contract, speed up or slow down. Transient dissipative processes are much more flexible.

As a set of largely stationary processes, the deep tropics stays warm enough in most climates for to convection to reach the top of the atmosphere, especially above the warm pool (De Deckker, 2016). This can be increased or decreased by expansion or contraction through thermal forcing (Schneider, 2006), allowing the vertical loss of heat from the tropical zone. Tropical processes also initiate meridional transport. However, if climate subject to increasing greenhouse gases is dominated by a storage and release process centred in the tropics, mechanisms for increasing meridional heat transport are required. Therefore, in addition to entraining heat transport in the tropical Pacific to make it available for dissipation, the Pacific Ocean heat engine governs the broader climate response expressed as MHT.

The limits for MHT provide robust boundary limits for the nonlinear dissipative processes required to transport heat from the tropics to the poles. This links the slower processes of oceanic heat transport with the faster processes of AHT. Regime changes not only warm or cool regions to increase/decrease MHT but can also switch between fast and low processes depending on whether heat transfer is increasing or decreasing. Regime shifts are transmitted via teleconnections that can be associated with both ENSO and decadal oscillations. Teleconnections during ENSO events are driven by the latent heat of precipitation (Mayer et al., 2013; Trenberth et al., 2002). Decadal oscillations are being increasingly understood as forming a teleconnected network (Cassou et al., 2018) that undergoes synchronous changes in both forced and unforced contexts (Neukom et al., 2019; Tsonis & Swanson, 2012; Tsonis, 2018).

**Comment 11. Change to manuscript**
We will add a summary of the above stressing the importance of meridional transport and present some of the new results showing examples of changes in meridional transport planned for Section 4.1 of the second paper.

**Comment 12. Lines 170-171**: are these values provided with an uncertainty range? If so, is it possible to have them shown?

**Comment 12. Authors' response**
These have been calculated.

**Comment 12. Change to manuscript**
The uncertainty ranges will be included in the main manuscript and details in the SI.

**Comment 13. Lines 174-175**: given that the existence of step-like changes in the TEP and TWP and the mechanisms underlying them are at the core of the results, statements like this one shall be corroborated by more quantitative arguments. At Line 202 the authors claim that they start to track changes after 1947, because of the poor quality of data prior to WWII. Then why caring of shifts happening in 1937-1942?

**Comment 13. Authors' response**
To provide a full account of heat engine behaviour we need to understand the how and why of any shifts detected in the historical record.

To place this particular analysis and result in better context, shifts in the broader climate system will be described in the context section of paper 1 before focussing on TWP and TWP. This would update the data from JR2017, using the additional four years to 2018. This adds another shift in 2014–15 to the overall record. It will outline how we measure the fast and slow components of change emphasise the relative stability of TEP compared to the ocean more generally, resulting in larger steps.

We also need to provide the reason for selecting ERSST and NCDC data and only using a single dataset rather than HadSST and HadCRU data in a revised methods, data and tools section (it was briefly mentioned and detailed in the SI). This is because the method of calculating HadSST involves independent sampling of each data point, removing some memory of the previous state, thus a loss of information (documented in the SI).

The main discussion about free and forced mode will come later, but it is a key consideration in the behaviour of the heat engine and network, so the description of how the climate data whitens abruptly will help set the scene. This involves moving lines 380–388 to the context section updating

the previous research. Additionally, all 29 regions in the NCDC data set have been tested for autocorrelation and white noise. Almost all regions whiten substantially, and only individual SST zones fail to whiten fully, although global average SST does. TWP itself shows shifts in autocorrelation from 0.32 1881–1922, -0.31 1922–1940, 0.62 1942–1971, -0.04 1972–2018, all at $p<0.05$. The tracking model only works when the data are white – we hypothesise that this is a feature of the change from free to forced mode. Unfortunately, the inhomogeneities in the data around WWII do not allow us to detect the nature of the change with a lot of confidence.
We report on shifts 1936–42 because they were detected, so we need to document why some may be rejected. Based on comparisons between ERSTTv3 with ERSTTv4 and ERSTTv5, the shift in 1937 has been moved later in the tropics and SH, and WWII temperatures have been raised and are now more of an outlier.

We are interested in shifts through the whole of the time series, especially to understand where those before WWII may have originated. The widespread shift in the late 1930s appears earliest in the 0–30N ocean in 1936 global ocean, 60–30S ocean and SH land in 1937, SH ocean 1938 and 20S–20N and 30S–0 in 1939. This was the start of the long period of stable temperature marking the end of the early 20$^{th}$ century warming, so it is of interest. If we reject the timing of a shift due to suspected inhomogeneities, we feel that needs to be added as a caveat, having reported it.

**Comment 13. Change to manuscript**
Whitening of temperature in the climate system will be discussed in data updates before the heat engine tracking and related modelling is introduced. Lines 380–388 will be moved to the context section updating the previous research. Free and forced mode will also be introduced at that time but discussed in more detail in Section 3.1.

**Comment 14: Figure 2d**: the authors compare here anomalies in GMST, TWP and TEP with TWP-TEP difference. The visual effect is that variability in the anomaly time series are damped by the temperature gradient offset about 2 K. I suggest that the gradient and the anomalies are shown in different panels.

**Comment 14: Author's response**
We will show them in different panels. As flagged earlier we may show the GMST chart in an earlier figure when updating the previous research

**Comment 14: Change to manuscript**
Alter figure as suggested.

**Comment 15: Line 190**: I do not understand the relevance of this statement. The GMST are roughly stationary from 1880 to 1920. If we consider the 1900-1920 instead of 1880-1899 as baseline period, the warming in TEP and TWP would probably be consistent with GMST warming.

**Comment 15: Authors' response**
The choice of baseline from pre-industrial is usually to 1899 or 1900. This is because of the general belief that forcing produces a proportional response in temperature and that selecting as close to pre-industrial climate can meet both policy (UNFCCC) and physical criteria. We do not know what caused the cooling around the turn of the century but our interest is in nonlinear response to forcing no matter when they happen. It is true that TWP has roughly warmed the same as GMST, but as we argue later, we think that up to 1937, TWP is catching up to previous levels. It then remains fairly stable until the late 1960s.

The point that TWP resembled GMST was raised around line 170 and is shown in Figure 2d. This section is mainly descriptive, with possible mechanisms discussed in Sections 3.1 and 3.2. We could add to this. Downward shifts in the SH coincide with the shift in the PDO (1895) and in the SH ocean, 30–60 °N and TWP in 1902. We presented monthly timing in Table S8 and that indicates the SH preceded the NH by 5–6 months, 30–60 N preceding the AMO by four months and TWP following a month later. Other regions shift in 1903, a year later. The TWP-TEP temperature differences were consistent with the PDO phases during this period, so they were maintaining the ~2 °C difference.

We do not know enough to attribute cause to these changes around 1900, but the downward shift in ocean temperatures provided the oceans with additional capacity to take up heat. They did not return to pre-1900 levels until 1936–37. We don't think this is roughly stationary – cooling interacted with decadal variability. Volcanic eruptions may have also influenced the timing of changes. The choice of baseline is to be as close to the pre-industrial period as possible, and to understand the evolution of climate since that time.

One question is whether TWP is temperature (heat) taker during the first part of the 20[th] century (in free mode) and temperature maker during the second half (in forced mode). The warming is consistent with GMST but we do not want to adjust the baseline, merely understand whether or not the TWP may be acting passive then actively.

**Comment 15: Change to manuscript**
We will edit the manuscript to reflect these issues a little better. One important matter is whether we can distinguish between radiative and thermodynamic forcing of temperature, and whether warming over NH land in the early 20[th] century due to solar forcing is combining with tropical ocean warming due to increasing greenhouse gases. This will be discussed.

**Comment 16: Sect. 3.1**: the title and the first sentence in this section suggest that at least qualitative arguments are provided to explain a causality mechanism connecting shifts in TEP/TWP with regime changes in natural variability (namely, PDO and AMO). The section is rather a collection of insight descriptions of step-like changes, in which the propagation of the signal is argued in terms of their coincidence with regime shifts and impacts over various aspects of climate variability. I also struggled with the definition given of "tracking model", given that the approach here shown seems to me rather an interpretative framework of the observed time series. It is interesting to infer causality links locally and remotely (the latter is partly accomplished in sect. 5). Nevertheless, there are several rigorous methods that might better serve the scope (e.g. Granger causality and derivatives) and I think that they should be addressed here.

**Comment 16: Authors' response**
Thanks for these comments and we would like to tighten up this area as suggested. However, we do believe that in the circumstances, showing whether shifts are synchronous, random or display some kind of regulated process is valuable information.

The tracking model: the initial rationale for setting up a tracking model was for several reasons. Given the correspondence between shifts in GMST and TWP we wanted to test whether TWP had any role initiating shifts. We also wanted to test the possibility that shifts were being caused by a build-up of heat energy that was not being removed by normal processes such as convection, warm pool expansion and ENSO events. In that situation, using temperature as a proxy for heat might be able to signal a regime change – this is based on having a steady-state attractor acting within strong boundary conditions that needs to switch to a new level because it passes a critical threshold.

An assessment of heat content in the warm pool 1982–2014 shows that changes in surface variables (warm pool volume) precede its central movement by 3 months, its depth by another 4 and heat content by 5 months (Kidwell et al., 2017). Warm pool volume is largely influenced by temperature >28 °C. Over this time, the warm pool moved 2° west, became 3 m deeper, and expanded by about 0.75 million km$^2$ and stored more heat (Kidwell et al., 2017). In this study change was measured as a trend but was nonlinear for depth, movement and heat. They also noted some response to phase change in the PDO (Kidwell et al., 2017). These findings suggest that TWP most closely follows volume changes.

The other reason was to explore the capacity to identify a shift soon after it had occurred, using a tracking process. We did not include those calculations within the paper, but the most recent shifts around 2014 were large enough to register within 2–5 years. If a step change is large than one standard deviation, it will register with 3–4 years at the $p<0.05$ level and those around one standard deviation or a little less will take longer. In a system that has whitened, detection periods tend to be shorter because variance is reduced. Understanding the precursor signals would be preferable, but without this, a detection process is the next best thing without investing in model predictions.

We took up the suggestion to investigate Granger causality. Rather than use large panel approaches, we applied the paired approach from the Real-Statistics package for Excel (we can call it cross-Granger analysis, given its similarity to cross-correlation analysis). This method applies a cumulative approach to Granger causality where all the lags up to lag n are included in the regression. This applies an information-first approach where correlations may be maintained over time, but if no new information is being added, Granger causality declines.

This was carried out for both monthly and annual data, de-stepped to produce stationarity and as observed for the 29 different NCDC temperature time series. The record was divided into free and forced periods 1880–1967 and 1968–2018 (May 1968 marking the period when TWP shifted after the mid-century quiescent period). The choice of change-over date is not highly sensitive within the limits of 1957–1973, which span global whitening for ocean and land. Where possible, lags up to n=24 were tested for both monthly and annual data, but annual data 1968–2018 only allowed to n=15 without increasing sampling errors. The results are to be summarised in their own section (3.4). A description of the technique and the results is presented in Appendix 1.

The Granger analysis has not added a great deal to the issue of timing of shifts. This is because steps have to be removed to produce a stationary dataset. However, by comparing the observed (with steps) and stationary data (steps removed) it does provide valuable results in showing how shifts propagate through the climate system. It identifies the central Pacific and ENSO as a major source of the heat fuelling shifts, which is consistent with the results of the tracking model. Between free and forced mode direct circulation reduces and teleconnections, mainly associated with ENSO, increase. This is consistent with Kjellsson (2015) who found that in climate models under forcing mass transfer declines but energy transfer increases.

These results will be linked to complex system dynamics, where we track emergent processes and teleconnections (see Appendix 2).

**Comment 16: Change to manuscript**
The description of the tracking model would be improved by summarising the rationale given above to explain why it was developed and the findings will be explained better.

We will summarise the results of the Granger causality and correlation work in an updated section (3.4) that describes how the heat engine process interacts with the climate network. A great deal of

data was generated, warranting a separate publication, so only the most pertinent findings can be summarised in any revision.

**Comment 17: Figure 3:** I found this figure very difficult to read. This is probably because it spans two pages, and also because the captions are not very informative, especially referring to panels (c)-(g). Please consider expanding the caption and/or splitting it in several figures.

**Comment 17: Authors' response**
Thank you for the comment. We will split the figure and expand the caption. Showing it in landscape mode might also be an option.

**Comment 17: Change to manuscript**
We will make sure the figure is easier to interpret.

**Comment 18: Lines 231-257**: the description of selected shifts is linked to several events that happened in different regions of the Earth. The connection seems basically motivated by the timing, but there is no specific argument for these links to be descriptive of large-scale processes occurring into the system, so I wonder why it is relevant to consider them.

**Comment 18: Authors' response**
We are puzzled by this comment. These are all shifts in temperature – most of the observed warming has occurred during these events – they have to be detected before they can be interpreted. Perhaps the intent of the comment is to encourage interpretation instead of straight description, which we will do. Shifts are behaving like teleconnections and coincide with teleconnections in decadal oscillations. This is mentioned a number of times in the discussion paper. Some regime shifts may be associated with tropical expansion but the timing of change in tropical expansion are difficult to measure (Nguyen et al., 2018; Nguyen et al., 2015).

For example, shifts in the midlatitudes may be due to increasing intensity in the subtropical ridge or the tropical edge. Shifts in SST adjacent to the warm pool may be current-driven (this seems to have influenced SST changes around Australia) but that does not explain cascading shifts in other ocean basins. It is possible there may be several mechanisms in play. Atmospheric teleconnections may be driven by changes in air pressure but it is not clear what is driving the thermal aspect of shifts in SST. Reid and Beaugrand (2012) identify tropical origins of shifts in SST and propagation via circulation, especially boundary currents. This is one area where sparse-data methods may be able to identify changes the behaviour of attractors that are otherwise maintaining steady-state conditions.

The order of the shifts described here contributes to the conclusion that shifts initiated in the tropics spread to the ocean elsewhere then to the atmosphere on lines 796–800. This summary statement needs to be covered earlier. It is also linked to the concept of a climate network, discussed in Appendix 2. See also response to Comment 20.

**Comment 18: Change to the manuscript**
We have already flagged a more proactive start to paper 1 highlighting the importance of shifts. The context setting will discuss the importance of teleconnections and we will integrate the different parts of the proposed section 3 and provide a more solid argument in support of thermodynamically-forced regime shifts moving through the climate network.

**Comment 19: Line 274**: do the authors refer to an increase in the incident solar radiation, or are they referring to net solar radiation? This shall be specified.

**Comment 19: Authors' response**
Incident solar radiation, but we will use incoming because incident is often used for increase at the surface, whereas the references cited are referring to an increase in solar forcing.

**Comment 19: Change to the manuscript**
The NH shifts 1920–25 may have been partially driven by an increase in incoming solar radiation.

**Comment 20: Sect. 4.2**: as in my comment about ll. 231.257, the authors propose here a connection between changes in TEP, TWP and TEP-TWP gradients with the AMO. They provide a survey of results available in literature about AMO phase shifts and link them to the step-like changes found in sect. 3.1, but nowhere is suggested that the two changes are correlated, nor any process responsible for it is indicated. While I understand the motivation for the arguments about PDO (in sect. 4.1), I do not see the reason for this discussion about AMO.

**Comment 20: Authors' response**
Thank you for the comment. Teleconnections are nominated in the text (line 366). We will try to make this clearer by adding more detail. Even though teleconnections go back to time of Walker, the processes around how they work is not very well understood (Rousi et al., 2018), although according to Mayer et al. (2013), they are driven by the latent heat of precipitation. There are different types of teleconnection, some can be identified via correlation, some operate via different mechanisms such as reversals, node changes and event synchronisation. The most straightforward way to understand complex interactions qualitatively is to gather as much information as possible, and use that to identify what type they might be using a general understanding of complex system characteristics.

We do not think that either the PDO or the AMO can be linked via simple correlation, but in Section 5.1, we describe reversals in correlation between the AMO, ocean and land that coincide with AMO phase shifts. These mark changes in circulation from ocean to land and land to ocean. Phase shifts in the PDO involve a change between El Niño-like and La Niña-like modes. On Line 350 we mention studies (Swanson & Tsonis, 2009; Tsonis & Swanson, 2012; Wang et al., 2009) that have conducted network analyses, finding periods when the decadal oscillations combine statistically, then use this to remove the linearised influences of each from the trend. The periods when these are thought to oscillate in unison are not very precise, but coincide with the dates when we detect shifts. These periods could indicate when the oscillating systems are close to a critical threshold.

Decadal oscillations occur within networks, and may be described as steady-state attractors, acting as potential wells. The AMO and PDO have two stable states and oscillate between these. Normally, a phase shift will end up in roughly the same level of potential energy, with the alternate pattern of circulation. If there is too much or too little energy available to drive that oscillation within its boundary conditions it may speed it up or slow it down, and during a phase change may drive it to a higher or lower potential. This would allow shifts up or down in temperature to coincide with the instability that accompanies a phase shift.

There is a growing literature on the connections between the AMO and PDO. For example, Sun et al. (2017) identify a link between the AMO and decadal scale variations in the warm pool that expresses a positive feedback between the Atlantic Ocean and TWP.

**Comment 20: Change to the manuscript**
The major features of complex system dynamics would be introduced in the context section. This would allow sections 4 and the decadal variations in 5.1 to be integrated in a discussion that is

focussed on the elements described above. Some new literature in support will also be introduced (e.g., Sun et al. (2017)).

**Comment 21: Lines 372-376:** the statements in this paragraph also seem to suggest that the arguments about the timing in AMOC changes, PDO and TEP/TWP/TEP-TWP changes are rather speculative and a rigorous analysis is missing, relating these evolutions. Given that the AMO and AMOC are barely mentioned elsewhere in the manuscript, the authors might consider simply withdrawing these paragraphs.

**Comment 21: Authors' response**
We do agree that these lines are speculative, but we disagree that it is overly speculative to track the timing of changes in a complex system. Given that complex systems have induced structures it is necessary to first understand what those structures may be.

Hopefully the above discussion has highlighted some of those links. We are working with the concept of a network where thermodynamic, physical and radiative processes transfer information in a number of ways. When the rate of forcing increases, the network exchanges slow dissipative processes for faster ones. AMOC is one of the slowest (with the Southern Ocean deep overturning). The HadISST AMOC index switches at around the same time as TWP undergoes a small shift affected regions tot the south (1968). The discussion around the different modes of MHT will clarify this. It is also discussed in Appendix 2.

The correlation and Granger-type analyses that illustrate the move from physical circulation towards teleconnection when the climate shifted from free to forced mode. This has also been linked to an increase in moist static energy, which results in reduced mass transport (Wu et al., 2019). The idea that climate is a nested system of nonlinear processes ranging from weather timescales through to millennia is not controversial. Here, we are just trying to illustrate how that might work in practical terms.

**Comment 21: Change to the manuscript**
In the response to previous point, we proposed that these issues can be better integrated into the paper by describing a network capable of switching from slow to fast connections up front then providing these as examples.

**Comment 22: Line 388**: I am a bit confused about what "tightening of the heat engine" mean in this context.

**Comment 22: Authors' response**
The proposed restructure should help to explain this. When the climate system moves from free to forced, both it and heat engine moves from being loosely to tightly coupled. Autocorrelation within temperature reduces because the coupling becomes too tight for random walks influenced by decadal variability. This is reflected in data becoming whiter. The heat engine itself tightens as circulation with the NH and with land declines and more energy is routed through the central Pacific.

**Comment 22: Change to the manuscript**
The restructure will lay out the evidence for the climate and heat engine moving from free to forced mode and illustrate the concept of 'tightening' much better.

**Comment 23: Lines 389-390**: this could be related to the definition of forced and free fluctuations described in Lorenz, 1979 (the authors cite it elsewhere in the text).

**Comment 23: Authors' response**
Totally agree. Lorenz (1979) is cited on line 140 with reference to free and forced but to the difficulty in distinguishing between the two. This point should come up in the context section, as it is material to the whole analysis, particularly as we have accumulated more evidence to show the distinction between them.

**Comment 23: Change to the manuscript**
See previous entries – the description of free and forced behaviour of climate will become more central and will be introduced earlier. We have also gathered more evidence to describe it better.

**Comment 24: Sect. 5.2**: given that the authors refer many times to the co-variability of GMST, TWP and TEP, I do not understand why this section is only at this stage of the manuscript, and not before (same for Figure 4b). I would suggest that it is moved before sect. 3.1, after Figure 2 is described.

**Comment 24: Authors' response**
Agree and thank you for the suggestion.

**Comment 24: Change to the manuscript**
Move section 5.2 to section 3.1 and expand in light of the changes proposed.

**Comment 25: Lines 433-435:** I wonder why reporting the number of detected shifts is important here, especially given that the authors claim at l. 437 that many models show a "pronounced decadal variability". Typically, the multi-model analysis is ill-posed, if one does not provide a weighting scheme or qualitative arguments discerning the models. This shall be addressed when providing multi-model averages.

**Comment 25: Authors' response**
We disagree with the assertion that model weighting schemes are this important. This comes from experience in developing multi-model projections for impacts and adaptation studies, and testing whether weighting improves salience, especially the ability to change decision-making. We understand this is more for assessing reliability and performance, but the principles are much the same. The law of errors suggests that unless there is a large systematic bias in a sample, errors will average out. There are few studies that have tested weighting against non-weighting in potentially influencing decision making (we have, but in technical reports only). However, we did follow up on this point and concluded that models with higher ECS (>3.5 °C) show lower skill (see below).

In analyses for JR2017 we constructed a skill score for individual models based on the timing of shifts detected against observations. The ensemble showed skill but there was no meaningful relationship between individual model skill and other diagnostic values. This was partly because, as our analysis showed, there is a strong stochastic component to shifts in the models. Model structure influenced this as shown by the GISS models that contained two different ocean schemes discussed in JR2017. These influenced ocean-atmosphere coupling, producing different sets of shift/total warming ratios. If forcing affects the periodicity of the PDO, as might be inferred from observations, the assessment of what is deterministic and what is stochastic will be better informed.

Another question is "skill at what?". Testing for shifts is a relatively novel undertaking and the behaviour of shifts is not well understood. Seeing whether models have any skill in this area at all the first step. For the 20[th] century part of the CMIP5 assessments there is no statistical link for steps or trends between the model ensemble and ECS, whereas for the 21[st] century the relationship is quite clear. We have shown this resides in the rapid component of change. However, using observed

TWP and TEP magnitude and the relationship between model TWP, TEP and GMST shows some skill in estimating ECS in the real-world climate.

When in the exploratory stage of assessing relationships where the concept of skill has hardly been explored, expecting weighting schemes is itself somewhat ill-posed.

Whether the models reproduced any aspect of the heat engine was investigated for the following reasons:
- A sizeable proportion of the climate science community believes an observed phenomenon does not exist unless it has been reproduced by a climate model, so it was important to test this.
- We wanted to see whether climate models could produce the basic structure of TEP and TWP and their relationship with GMST; especially given the limitations of climate model performance in the tropics.
- Emergence is an important part of complex system behaviour, so we wanted to see which phenomena are emergent and to what degree.

With respect to decadal variability, we viewed its presence as positive. As the ENSO cycle is emergent in models, we were hopeful they showed decadal variations also. Based on the analysis in section 4.1, we see the PDO as an integral part of the heat engine behaviour, so some kind of decadal variability would indicate that decadal scale processes are emergent. We did not expect the models to produce internally-generated shifts solely due to internal variability because they too small to do so in observations and the models are less responsive. Testing mode changes in 'PDO'-like oscillations would require indices to be extracted and analysed and for the time being we have only extracted TEP, TEP and GMST.

The only multi-model averages we calculated were the numbers and size of shifts in TWP and TEP, and then mainly because we are interested in the relationship between those numbers and ECS. The rapid transfer of heat from the shallow ocean to atmosphere will result in rapid positive atmospheric feedbacks. This exercise is about collecting general statistics and then seeing how well they reproduce basic aspects of the heat engine structure.

The results are of interest because TEP shifts occur less often but are roughly twice the size of shifts in TWP, which are twice as frequent. The TWP–TEP gradient is also constant in the models but is usually less than observations, only two falling within a standard error of the mean and only two larger of the 28 in total.

Comprehensive sets of model scores for CMIP3, CMIP5 and CMIP6 have recently been made available (Fasullo, 2020; Fasullo et al., 2020). An initial exploration shows a positive relationship between longitudinal grid size and success rate for TEP and TWP coinciding with GMST; i.e., the larger the zonal grid size, the better the results ($p<0.05$). Due to the relatively low sample size, this could be coincidental, or if real may be related to coupling and representation of zonal processes.

Correlations between a number of measures of model skill in representing observed climate are negative for some measures linked to TWP and TEP; these are based on the 21 models with skill scores and ECS both being available. These relationships are negative for overall energy, ENSO, precipitable water, sea level pressure, net shortwave toa and cloud, and surface relative humidity. The relationship between average TWP size and skill scores is negative while for TEP size is neutral. However, TEP frequency is strongly negative – models with high skill scores had fewer shifts in TEP. There was no relationship between skill and the number of shifts in TWP and GMST. Based on the

relationship between ECS and step size, we estimate an observed ECS of 3.2 °C for TEP and TWP separately (p<0.01 and p~0.05) and 3.1 °C for them jointly (p<0.01).

The sample was divided into two groups above and below ECS=3.5 °C. For the lower ECS group most of the highly negative correlations are lowered or even reversed. The major changes are for the TEP-TWP gradient in °C, and the shift/trend ratios for TEP and TWP, which are more step-like at lower values of ECS. The results that retained high negative correlations are highly correlated with ECS. This suggests that the higher ECS models are being allocated lower skill scores, and these continue through to correlations with shifts in TEP and TWP. The exceptions are the number of shifts in TEP, which is strongly negative for the whole distribution, and the step-size in TWP. A strong positive correlation in the TWP-TEP difference emerges for some variables (6/23). We view this as a strong diagnostic because the gradient between TEP and TWP is a key part of the heat engine interaction.

ECS calculated using the <3.5 °C sample of 2.9 °C was obtained at the p~0.05 level for the TEP and TWP joint regression. Regressions for TEP and TWP individually are the same, but not statistically meaningful. As a curiosity, we estimated the skill scores for observations from the <3.5 °C ECS group in our sample. For annual skill it is 94.6 ($r^2$ 0.41) and overall skill is 93.8 ($r^2$ 0.22).

**Comment 25: Change to the manuscript**
More emphasis will be given to emergence, especially to the order of emergence within climate models. In Appendix 2, we show that shifts in climate model output emerged at the same time as ENSO, presumably due to the same set of processes. If the overall structure of the heat engine is reproduced by models without necessarily being able to reproduce the same detailed dynamics as seen in observations, then that structure, based on a stable cold reservoir dominated by negative surface feedbacks and a warm reservoir dominated by positive surface feedbacks with a constant gradient between the two, is a robust feature.

It is possible that ENSO, related decadal variability, attractors and the capacity to shift in response to thermodynamic forcing are related to (i) ocean-atmosphere coupling and (ii) reproducing the basic structure of the tropics with the thermocline and stationary coupled processes such as the Walker and Hadley Cells. This would imply that the whole climate network is emergent with coupling, not just ENSO. However, this does not imply any specific skill level.

We will add a summary of the skill issue along the following lines:
Model skill data is now available (Fasullo, 2020; Fasullo et al., 2020) and combining ECS and skill scores provides a sample of n=21 for testing. Correlations between a variety of measures linked to TWP and TEP and model skill are negative for overall energy, ENSO, precipitable water, sea level pressure, net shortwave toa and cloud, and surface relative humidity (Table in SI). This would suggest that heat engine performance is linked to models with lower skill.

However, the poorer skill scores and negative correlations are associated with models with ECS>3.5 °C. For those with ECS<3.5 °C, positive correlations are linked to shift/trend ratios in TEP and TWP, and the TWP–TEP gradient. The results that retained high negative correlations for both groups are highly correlated with ECS. Higher model ECS is associated with more frequent shifts and greater connections between TEP, TWP and GMST, which may be a response to increasing thermodynamic forcing.

Average TEP and TWP from the whole model sample (n=28) is 0.55 °C and 0.28 °C for TEP and TWP respectively, but from the <3.5 °C sample (n=11) is 0.46±0.17 °C and 0.26±0.10 °C, respectively, almost the same as for observations (0.46 °C and 0.24 °C). What constitutes skill with respect to the timing and magnitude of shifts and representation of thermal forcing still needs to be resolved.

**Comment 26: Sect. 5.4**: the aim of this section is supporting results shown in previous sections but, in order to do so, authors provide a survey of previous analyses (Andrews et al. 2015; Andrews and Webb, 2018; Dong et al. 2019). I think that it would be more appropriate if the authors would extend this discussion describing the correlation of their data with actual measures of the atmospheric feedback, such as OLR, the cloud radiative effect (CRE). These are available either as satellite data (e.g. CERES EBAF), Reanalyses (ERA5) or climate model outputs (CMIP5, CMIP6).

**Comment 26: Authors' response**
We still think these reviewed papers are very useful because they show different atmospheric feedbacks from east to west that counterbalance the surface feedbacks. This provides feedback counterbalances between the ocean surface and the atmosphere. This implies the presence of 3D attractors even if they are usually mapped in 2D. We investigated top of the atmosphere variables to see whether they provided further insight. This involved data that was relatively accessible, including the NOAA Interpolated Outgoing Longwave Radiation (1979–2019), and selected variables from the NCAR-NCEP Reanalysis 1 (R1 1948–2019) and NOAA/CIRES/DOE 20th Century Reanalysis (V3 1836–2015).

Globally, top of the atmosphere LW shifted up by 2.4 W $m^{-2}$ in 2003 from 1979–2002 and down by 1.7 W $m^{-2}$ in 2014. This was driven by changes in the NH tropics and is partially related to phase changes in the PDO. There were increases 60–90° in both hemispheres from around 2000 and 30–60 °N. The region 30–60 °S has remained fairly stable. The shift upward 60–90N was 4.2 W $m^{-2}$ in 2002 and 2.2 W $m^{-2}$ in 2016 and 30–60 °N by 3.1 W $m^{-2}$ in 2001.

The reanalyses show quite different patterns of change for the same period. They do not capture the large increases in the high latitudes and for most regions, correlations with observations are <0.5, with the tropics being poorest. The highest correlation is 0.76 for 60–90N between observations and NCAR-NCEP v1 increasing by 3.8 W $m^{-2}$ compared to 6.4 W $m^{-2}$ for observations over the same period. Over TWP and TEP, correlation was also high. This may be due to OLR partially being estimated from surface heat content and the small area. Largely stationary diffusive processes and characterisation of surface temperatures and feedbacks also contribute to higher correlations (Gastineau et al., 2014; Trenberth & Stepaniak, 2003a). There was a small reduction in OLR over TWP in 1999 and small increase over TEP in 2002, both statistically minor. The reanalyses reproduce the direction of change but not the magnitude.

The tropics (30S–30N) and extratropics (30–90 SH and NH) 1979 – 2019 are shown in Fig. 6. For the tropics, there is no net change in OLR and correlation with reanalyses is low (0.23 R1, 0.11 V3). For the extratropics, correlations are higher (0.54, 0.49), but the change before and after 2003 differ (2.2 W m-2 compared to 0.5 (R1) and 0.3 (V3). Longer term, the records are very different. NCAR-NCEP R1 increases substantially from 1948, which looks like a start-up issue and the 20[th] C V3 declines by about 1 W $m^{-2}$ in the tropics and increases only after 2010 in the extratropics by 0.5 W $m^{-2}$. Note also, the large differences in interannual variability between the observations and reanalyses by a factor of 2.6 and 6.6.

[Figure]

Figure 6: Comparison of observed (satellite, NOAA Interpolated Outgoing Longwave Radiation) and reanalysis (NCAR-NCEP Reanalysis 1, NOAA/CIRES/DOE 20th Century Reanalysis V3) 1979–2019 shown as a 1981–2010 anomaly.

These cover just one satellite record and two reanalyses, but they reflect the problems of reconciling different measures of energy-related variables discussed in the wider literature. Most measures cannot be directly observed, depend on either remote sensing, modelling or both. Observations are affected by instrumental inhomogeneities and changes in technology. Furthermore, when they are assessed, the researchers are most interested in analysing spatial patterns, trends or both. The expectation of researchers is that the forced change is trend-like, so decadal variations are routinely assumed to be due to climate variability, and apparent inhomogeneities are assumed to be measurement related.

The detection of shifts requires high quality, homogenous data from records as long as possible. Combining adjusted and infilled data from remote sensing with modelled data is an apples and oranges exercise if they are not measuring exactly the same quantity or have been derived using different methods.

We investigated modelled interactions drawing from the CESM1-CAM5 model where we conducted a tracking exercise, testing the relationship of TWP, TEP and shifts in temperature elsewhere. Global top of the atmosphere longwave radiation (outgoing longwave radiation; OLR) was analysed with the bivariate test (Fig 7, top panel). Negative shifts in 1883, 1963 and 1982 can be associated with volcanic forcing. Positive shifts occur in 1996, 2013, 2030, 2043, 2064 and 2080. This timing matches shifts in GMST of 1998, 2014, 2029, 2043, 2054, 2066 and 2081, with only 2054 missing. From 2000, they also fall within one year of shifts in tropical ocean (1970,1998,2013,2029, 2043,2065,2081).

Shortwave radiation increases then decreases in almost mirror image, increasing in the 20th century in 1883 and 1959, then shifting downward in 2016, 2033, 2053, 2062 and 2079 (Fig. 7, second panel). The 21st century component is dominated by gradual changes, with shifts playing a secondary role. When net radiation is analysed, the two almost cancel each out, leaving a very regime-like response (Fig. 7, third panel).

[Figure]

Figure 7. Shifts in variables from CESM1-CAM5. Top of the atmosphere longwave radiation (yellow), toa shortwave radiation (purple) and net toa radiation (green). Surface sensible heat flux (red) and surface latent heat flux (blue).

Surface heat flux is represented by sensible and latent heat components. Sensible heat flux declines gradually over the century from about 1930 (Fig 7, fourth panel). Change in latent heat dominates surface heat flux (surface latent heat flux; SLHF, Fig. 6, bottom panel), declining gradually over the 20th century and increasing in the 21st. The record is dominated by regime shifts, increases occurring in 1998 (p<0.05), 2014, 2030, 2043, 2064 and 2080. These are clearly related by timing to changes in top of the atmosphere longwave radiation. The decline over the 20th century in OLR to 1995 is almost three times the surface latest heat flux (0.14 W m$^{-2}$ per decade compared to 0.05 W m$_{-2}$ per

decade) and the increase after 1995 is 3.8 W m$^{-2}$ compared to 4.4 W m$^{-2}$, so the surface flux increases faster than OLR during this period. The difference is mediated by absorbed shortwave radiation.

Lucarini and Ragone (2011) identify a number of internal inconsistencies within the CMIP3 models with respect to meridional energy transport and energy balance. We could not identify any studies that updated this for CMIP5, but given the similar performance between CMIP3 and CMIP5 models in terms if energy dynamics, these shortcomings will be present in that archive as well.

We tracked the timing of shifts and TWP excess temperatures using this model and two others, NorESM1-M and MIROC-ESM. All models showed shorter-lived periods of regime shifts dividing more stable periods. None showed a consistent approach to which regions led or lagged. TEP led in some instances, TWP less often, sometimes ocean and sometimes land. We tested the timing of surface latent heat flux and OLR from Fig. 7 and they followed surface shifts, sometimes by only a month. Shifts in surface latent heat flux generally lagged, but led once. In this, the models show more co-ordinated changes than the reanalyses – this may because they have a simpler representation of these processes.

Regime changes in variables such as temperature, humidity and surface heat fluxes will rapidly change dissipation rates. These in turn will affect atmospheric feedbacks, also very rapidly (Sherwood et al., 2015). Figure 7 shows that surface latent heat is clearly regime-like in the CESM1-CAM5. Its timing closely follows temperature changes. This shows that both dry and moist static energy in the atmosphere are responding to forcing in a stepwise fashion.

Shifts in both latent and sensible surface heat fluxes are also present in both reanalysis data sets looked at. More analysis is needed, but heat flux changes do not closely match shifts in temperature within the reanalyses. The direction of change resembles those in the models, with surface latent heat increasing in the higher latitudes and responding to decadal variability in the lower latitudes, especially over the ocean. Surface sensible heat flux mostly shifts negatively.

If the bulk of the warming response is not directly radiatively forced, but is a storage and release process, the changes being attributed to radiative forcing will need to be divided into direct radiative forcing, thermodynamic forcing and radiative response. This is a causal attribution separate to considerations of effective radiative forcing, which was developed to understand the global temperature response to forcing (Boucher et al., 2013; Hansen et al., 2005; Myhre et al., 2013; Shine et al., 2003).

The causal process moves from trapping of heat by radiatively-active gases mostly near the surface, its storage in the shallow ocean as the coupled ocean-atmosphere system stays in steady-state due to the strong negative feedback or damping effect of the oceans. This is part of the oscillatory system dominated by the PDO and AMO which allows shorter lived oscillations (e.g., ENSO and probably the North Atlantic Oscillation) to return to mean conditions. This feedback is dominated by the central Pacific as seen in TEP, but is also anchored by longer-term oceanic processes in the Atlantic and Southern Oceans. As such, these oscillations are acting as attractors.

The dissipative process appears to be thermodynamically driven. This term has not been used for the wider climate but has been used for short-lived phenomena such as tropical storms (Chan, 2009; Wilcox & Ramanathan, 2001) and also to ocean-ice edge dynamics (Kusahara et al., 2017; Shine & Henderson-Sellers, 1985). Thermal forcing is much more commonly used, but this generally refers to when heat is applied to part of a system. The dynamic part of the oscillatory system is largely wind-driven as shown by the many studies that initiate oscillatory behaviour. However, the boundary

limits and positive and negative feedbacks involved in the oscillation themselves are thermodynamic. With respect to the regime changes temperature and related variables, dynamically-driven changes will result in a mode shift, but the overall energy balance will remain unchanged. However, if there is sufficient thermodynamic forcing, a regime shift to higher/lower energy may occur. It is likely that this combines both dynamic and thermodynamic components, hence thermodynamic forcing.

The radiative response is the response to warming, and involves cloud feedback, changes in absorbed shortwave radiation due to factors such as cloud-driven responses and changes in atmospheric water vapour content. To date these have mainly been looked at as part of radiative forcing.

Our conclusion is that the climate models have significant shortcomings in being able to reproduce observed heat engine behaviour. Based on our initial analyses, the reanalysis models do too. If regime changes are due to coupling between the atmosphere-ocean they may not capture the more detailed behaviour of the Pacific Ocean heat engine until the deep tropics can be better represented, which includes the TEP-TWP relationship, the intertropical convergence zone and the ability to 'tighten' from free to forced mode under sustained forcing. Until they can do so, it is unlikely that measures such as regime changes in high latitude OLR, or the evolution of toa radiation balance in the tropics, can be properly captured.

This reinforces the view that the real world should be the primary focus of interest and not the control–perturbation relationship represented by climate models. Although the latter is important, the real world is the best climate model we have access too, and we should rely on it more strongly.

**Comment 26: Change to the manuscript**
Combined with the additional arguments regarding emergence and MHT, this provides a much more comprehensive description of how the heat engine works and the key boundary conditions governing it. The order of emergence from models suggests that shifts are the result of ocean-atmosphere coupling, but that specific behaviour relates to how detailed the model structure is particularly in the deep tropics.

This favours the two-paper structure, where the first part paper focuses on climate and heat engine behaviour and the second focuses on the heat engine structure. The major role for the heat engine in a changing climate is the meridional transport of latent heat. Thermal forcing occurs within the limits of the meridional heat transport curve, which is largely geostrophically-defined. The paper will be structured to propose the following.

Dissipative processes range from slow to fast, and the whole system is a self-governing network dominated by the Pacific Ocean heat engine. Heating moves the system from slow to fast processes and cooling the other way. Historically, we have identified two modes: free and forced. In free mode, location-based forcing such as changes in incident solar radiation in specific latitudes, or aerosol emissions can result in a regional forcing response. This may cause regime shifts elsewhere to rebalance the system. For example, warming in the early 20[th] century largely in the northern hemisphere prompted a regime shift in the tropics. The main effect of radiative forcing due to greenhouse gases is on heat storage in the tropic oceans. During periods of warming, the ocean whitens, which we take to be an increase in dissipation rates, and a sign that the climate system is undergoing thermodynamic forcing due to the build-up of heat.

The largest increase is in the transport of moist static energy at the expense of slower ocean heat transport. Circulation also makes a transition from direct circulation to being more teleconnected,

ENSO plays an increasingly important role and the behaviour of the PDO is more pronounced (shifts are more distinct). Upward mass transport is reduced (Wu et al., 2019). Shifts in temperature in the ocean transmitted from the tropics move into the extratropics and from ocean to land. These result in rapid adjustments to atmospheric feedback and absorbed solar radiation subject to cloud feedbacks and increases in water vapour. The increased transport of atmospheric heat poleward results in polar amplification. The changes in top of the atmosphere radiative balance is the product of increased ASR and greater availability of longwave radiation in the upper atmosphere. Significant improvements in modelling capable of reproducing observations will be required.

**Comment 27: Line 462**: what do the authors mean by "positive spatial variations in atmospheric feedbacks"?

**Comment 27: Authors' response**
This is what we say "Spatial variations in atmospheric feedback are strongly positive in areas of subsidence and weak to negative in areas of uplift."

**Comment 27: Change to the manuscript**
Replace with "Atmospheric warming feedbacks are strongly positive in areas of subsidence and weak to negative in areas of uplift."

**Comment 28 Lines 533-558:** the authors claim here that treating the forced response and the internally-generated nonlinear variability in a separate way has no physical foundations, and they give arguments to explain that. Provided that I tried to explain in my comment to l. 33 that this sharp separation is due to a partial misinterpretation of the theory and modelling of the climate response and forcing attribution, I will try here to problematize some of the arguments. The main underlying argument they provide is that the fact that the ocean uptake of the energy imbalance determined by the anthropogenic forcing implies that the response of the system is modulated by the regime changes in oceanic variability. The authors claim at ll. 505-507 that the warming ultimately determined by the greenhouse gases is not affected by their conceptual model of the Pacific thermal engine. Rather, the model seeks to better characterize the paths through which the warming can be achieved. The S/N approach is indeed aimed at determining the overall response of the system to the anthropogenic forcing at timescales for which most of the modes of natural variability, including oceanic multi-decadal variability, is treated as noise. When looking at decadal variability or shorter timescales, the way variability is affected by the forcing is the main subject of investigation in many fields, ranging from North Atlantic weather regimes (e.g. Strommen and Palmer, 2018), ENSO regime changes (e.g. Kim et al. 2014, Cai et al., 2015, Kohyama et al. 2018), to the impact of resolution in climate models (e.g. the European project PRIMAVERA H2020). At ll. 549-550 they argue that the marginal uptake of heat by the atmosphere is soon to be uptaken by the ocean. I do not see why this should be relevant, given that the atmosphere exchanges heat mainly through latent heat and isotropic emission of LW radiation: the latter, mainly affects the exchanges of heat between the atmosphere and the continents, and is only partly emitted towards the ocean's surface. Given that, the atmospheric marginal heating is the net result of all these exchanges, thus it should not be accounted for as a heat source for the ocean.

**Comment 28: Authors' response**
This is a substantial comment and one reason why we have engaged in lengthy responses to the points made. We made two points earlier about theory and method associated with the standard approach. One is that method overwhelmingly treats the two as separate. The second is that the theory is much more equivocal about the relationship between forced change and internally-generated variability. However, if practice favours only one alternative, what is the point of exploring theoretical alternatives other than an intellectual past-time for scientists? We understand

that most climate scientists are of the view that climate projections will not change much if external-internal processes interact strongly, because this will add stochastic behaviour into a long-term trend. This view is articulated in Comment 30. In this case, the basic approach evolves. However, there has been almost no incorporation of nonlinearity into the standard approach from when it was first quantified via pattern scaling in the early 1990s, almost 30 years ago. At best, random nonlinear behaviour is incorporated into pdfs, which do not characterise risk adequately (Carter et al., 2007).

The referee is correct in pointing out that on lines 505–507 we say that the ultimate long-term warming is consistent regardless of the nature of the atmospheric dissipative process. We covered this in JR2017 (it was the topic of the Discussion), so it may need to be restated in any revision. The referee then moves onto the signal-to-noise model, overlooking the arguments presented in Section 6.1, which maintain that gradual warming is physically implausible. We also make the point that the signal-to-noise model contains cognitive values that preference one set of theories to another.

The S/N approach is a method of analysing measurement in order to make inferences. It may shed some light on the physical processes involved, but its very nature means that nonlinear dissipative processes will be dissipated in the process of extracting a signal. There may be a large literature investigating decadal and shorter variability, but these studies are overwhelmingly carried out within the signal-to-noise paradigm.

Returning to section 6.1 in reference to lines 549-550, the argument is put that atmospheric marginal heat exchange is the result of latent heat exchange and isotropic emissions and that it should not be accounted for as a heat source for the ocean. This begs the question as to how the ~93% of EEI can be measured as heat stored in the ocean. The radiative effect may be isotropic at the molecular level, but due to the concentration of greenhouse gases, especially water vapour and $CO_2$, the greatest concentration of trapped heat is near the surface, especially in the presence of high $pH_20$, so its flux is biased downwards.

Our argument is essentially about energy balance. The atmosphere cannot warm independently of the ocean, because of the superior heat conductivity and capacity of the ocean and the active coupling between atmosphere and ocean that has a powerful negative feedback or damping effect. Gradual warming in the atmosphere would require a gradual increase in SST. Fig. 8 shows global average SST, which has a shift/total change ratio of 85% (0.69 °C of 0.81 °C).

Despite the argument for greater radiative transfer over the land surface per area, the coupled ocean-atmosphere system also maintains steady state over land, with the net transfer of energy from ocean to land, mainly as latent heat. For example, Dommenget (2009) argues that 80–90 % of continental warming driven by anthropogenic forcing is indirectly forced by ocean warming, not locally by radiative forcing. Ocean warming can be used to predict changes over land (Byrne & O'Gorman, 2018).

[Figure]

Figure 8: Step change break points and internal trends global average SST ERSSTv4 1880–2018.

When shifts are factored in, our tracking model results are consistent with this, as are regime shifts in Tmin over land often preceding those in Tmax (JR2017 and this paper). The latter has a strong land-surface moisture feedback. Finally, the Granger analyses show that circulation through the heat engine (influencing TEP and TWP) has increased substantially. Meanwhile, TEP retains a strong steady state and has shifted twice, and TWP is more responsive, relating to the changes in Fig. 8 (1987–88 being the exception).

**Comment 28: Change to the manuscript**
As per our discussion above, we do not accept that our position on the theory has been mis-conceived and see no reason to retreat from our previous position and plan to advance it (see responses to other comments).

**Comment 29: Line 575:** the authors might be more specific and explain what they mean in a scientific context when they say that the system "does not 'flip-flop"'.

**Comment 29: Authors' response**
This was in reference to a fuller quote from Hasselmann (1976) in his model of stochastic climate variability who said "A basic difficulty of unstable feedback models (apart from – or possibly because of their high degree of idealization) is that they tend to predict climatic variations as flip-flop transitions and therefore fail to reproduce the observed continuous spectrum of climatic variability." This was edited out of the manuscript due to space reasons but in doing so, the term lost its context.

Hasselmann (1976) was referring to simple positive feedback models that produced spontaneous transitions in climate state such as those of the daisy-world type. Otherwise, he maintained that climate forms continuous climate spectra on weather to millennial time scales. He continued: "In order to obtain a statistically stationary response, stabilising negative feedback processes must be invoked." The stabilising negative feedbacks were discussed in Frankignoul and Hasselmann (1977) and their capacity to dampen perturbations discussed in Frankignoul (1985); Frankignoul and Kestenare (2002) and Park et al. (2005).

Combined with the linear response model, this would result in a trend-stationary process where forcing produces the trend and negative feedbacks return any perturbations to the trend. Such perturbations would range across the spectrum ranging from weather to decadal scales and longer. If such variability was shown to be partly deterministic, the stochastic element would become more

predictable along the lines suggested by the referee in Comment 30. We agree that stabilising negative feedback are present, but these are an essential component of steady-state attractors.

This model is led by its assumptions. The two most important are that climate forcing follows the linear response theorem and that climate variability forms a continuous spectrum. Decadal scale "flip-flops" introduce discontinuity into the variability spectrum, which is not seen as a barrier by Ghil and Lucarini (2020). However, we have also rejected the linear response theory in favour of a storage and response process. We also argue that the strong negative feedbacks in the ocean used to justify the stochastic climate variability model constitute a major reason why the atmosphere cannot warm independently of the sea surface. Nor does the land control this process because the net energy flux is from ocean to land.

**Comment 29: Change to the manuscript**
See the changes proposed elsewhere.

**Comment 30: Lines 578-583**: surprisingly, the authors seem to ignore that significant rigorous results have been achieved on characterizing the impact of the forced variability on a chaotic system. This includes approaches based on studying the parametric smoothness of minimal QG models (Lucarini et al. 2007, Physica D), the crisis of the attractor by means of Koopman operators (Tantet et al. 2018, Nonlinearity), stochastic perturbations of edge states (Lucarini and Bodai 2017, Nonlinearity). The authors refer to Bartsev et al. 2017 to motivate the statement that the climate forcing projection over the leading modes of natural variability is inadequate to explain the nonlinear response of the system, given the inevitable presence of multistable regimes. Bartsev et al. 2017, though, use conceptual models to demonstrate that multiple stable state possibly exist in a complex system, such as the climate system. Rigorous arguments have to be provided (which is deliberately not the aim of Bartsev et al. 2017), to explain how this applies to the real system. In fact, a survey of available literature would probably convince the authors that, despite the lack of sufficient observations, precise constraints have been provided on to what extent the Hasselmann-type response dominates and when the chaotic nature of the system emerges, leading the system towards critical transitions (cfr. Ghil and Lucarini, 2019, arXiv:1910.00583 for a review).

**Comment 30: Authors' response**
We have spent a great deal of time considering this comment and cannot agree with the Referee on the relative importance of these two approaches. We are applying rigour in a different way. When writing the discussion paper, a major concern was whether the literature had identified or explored complex system behaviour consistent with our analyses. We were aware of the work on weather states and long-term critical transitions affecting global climate (e.g., global climate states; (Lucarini & Bódai, 2017; Lucarini et al., 2010; Peltier et al., 2004)). We accepted that mega-climate transitions do occur and that they can be triggered dynamically. Furthermore dynamically-driven state changes have been simulated on a wide range of scales including decadal. We were surveying the literature to see what studies applied to what we had detected in observations and to a lesser extent in climate models. Especially relevant were externally-forced regime changes, at decadal timescales.

In contrast, according to their own conclusions, the results explored by Tantet et al. may be applicable to some models, but are too detailed for observations. The Lucarini et al. (2007) study looks at responses to the equator-pole atmospheric temperature gradient identifying changes that move beyond stable circulation to periodic behaviour, finding that they vary smoothly with the gradient and follow a power law. This is consistent with the general increase in complexity with forcing (Gent & McWilliams, 1982). However, the range over which this was tested extends far beyond the range that would occur in the presence of a deep ocean. In any case, observations show the equator -to-pole gradient itself is subject to regime change, decreasing in steps with forcing. For

example, the shift in temperature at the equator in 2014 was about 0.25 °C whereas the increase in the Arctic was about 0.6 °C.

The studies mentioned above explore complexity within the broad framework of linear response theory, aiming to find generalisable rules and a comprehensive theory of climate. Regarding the final sentence in Comment 30, a survey of available literature then and now has not convinced us that a Hasselmann-type response dominates the forced change. Hasselmann introduced the concept of stochastic forcing to linear response theory but beyond providing the means to apply it, never defined it (in terms of the physical process driving it). Stochastic forcing has been applied in many different ways to many different types of forcing. We see it as a useful tool for representing uncertainty but a solid understanding of actual process in each case is preferable. With respect to the Ghil and Lucarini (2020) review, we have followed this (impressive) paper through its various iterations. Not all topics could be covered in the review and these are briefly covered in the concluding remarks. Several of these are relevant to the issues at hand (e.g., network theory, beyond linear response theory and detection and attribution).

Idealized studies of complex system behaviour face the problem of methodological underdetermination where many phenomena have a number of non-unique solutions (see Appendix 2 on model underdetermination). The same issue faces competing explanations with statistical models. This is a major reason why we applied a severe testing framework in JR2017, concluding that warming cannot be gradual and is dominated by step-like change consistent with heat releases punctuating steady-state regimes. As we outlined in the response to Comment 26, the strong boundary conditions and degrees of freedom within models suggests that many roads can lead to Rome, but only one of them can be the correct one (and none may be).

With respect to the comments about rigour in the real system. Bartsev's group (via Pavel Belolipetsky (2014; 2013), Chris Reid (2012; 2016) and us have independently identified step changes in temperature. We knew there was a thermodynamic component to those shifts and a method to attribute thermodynamically-driven changes on land had been published (Jones, 2012). This method distinguishes forced from unforced shifts using a linear inverse methodology on annual average Tmax and Tmin to detect changes in sensible heat by subtracting variability due to changes in latent heat. Bartsev et al.'s (2017) approach was reasonable in the light of that.

An earlier study that may be relevant is from Nicolis (2000) who investigated quasigeostrophic limits in a reduced Lorenz model forced by the thermodynamic component with the output measured as horizontal entropy and kinetic energy. At low forcing, the stable solution is a Hadley Cell, followed by two asymmetric stable solutions consistent with models of this type (Gent & McWilliams, 1982). Increased forcing leads to higher horizontal entropy and kinetic energy, leading to weak chaos, periodic limit cycles and chaos (Gent & McWilliams, 1982; Nicolis, 2000). The Lucarini et al. (2007) study also revealed a similar order. Thermodynamic forcing does not introduce new fluxes but influence dissipation indirectly. Under geostrophic limits, dissipation in the periodic regimes lies between the energetically expensive Hadley circulation and inexpensive asymmetric stationary circulation Nicolis (2000). Because increased forcing led to outcomes that were less dissipative than symmetric patterns but more dissipative than stable asymmetric patterns, she concluded that an organising principle was in play. She also discounted strict limits because the system could increase or decrease dissipation in moving towards an attractor. This includes both straightforward efficiency and entropy (Nicolis & Nicolis, 2010).

The Nicolis (2000) paper was located after we updated the conceptual heat engine and network model and we found many aspects in common. The separation of thermal effects and dynamics, horizontal and vertical entropy was of interest. With increases in forcing, both entropy production

and kinetic energy increase, monotonically at first, but as new patterns of circulation emerge, both shift nonlinearly, with horizontal entropy responding first and vertical with more chaotic conditions (Nicolis, 2000). The change in dissipation rates towards the attractor also accords with climate regimes obeying steady-state conditions. Nicolis (2000) concluded that the results were not due solely to the model but generic properties of the atmosphere, which contains all the phenomena included in the model, but over a wider range of timescales from hours to millions of years.

As to rigor, we follow the statement from Steve Brunton (articulated in Champion et al. (2019) that deriving complex system behaviour from first principles is difficult to impossible, and that most derivations, such as attractors, will be descriptive. Such derivations are ideally based on observations and used to build models. If we were to speculate on how the heat engine operates in oscillatory mode, the relationship between TEP and TWP, which move heat from a cold to a hot reservoir on a gradient that acts something like a see-saw rather than pendulum, the exchange between the two acts as a chaotic bridge oscillating nonperiodically (ENSO) and quasi-periodically (PDO). This longer-term oscillation involves input of upwelling cold water from the Southern Ocean, surface heat exchanges extending from the east to the western Pacific along the surface and thermocline and concentration of heat for vertical and meridional dissipation. It involves both dynamic perturbation (e.g., surface winds) and thermodynamic forcing. The latter depends on the relationship between heat available for dissipation, the potential for the heat engine to dissipate that heat and the meridional energy gradient between the equator and the poles.

Once a reasonably comprehensive empirical description of the heat engine has been developed more idealised studies could proceed with greater focus.

**Comment 30: Change to the manuscript**
See the changes proposed elsewhere.

**Comment 31:** Lines 641: the authors might want to provide references on the land warming leading the oceans (if it's not just a matter of pace of the warming).

**Comment 31: Authors' response**
Thank you. It is pace warming, so we will make this clearer. One issue is that in non-scientific communication the land warming first is often stated as the land warms first, but most of the literature falls into two groups: simultaneous or ocean first.

**Comment 31: Change to the manuscript**
This point will be drawn into the description and discussion of the Granger results and ocean to land tracking.

**Comment 32 Lines 668-675**: I do not understand how this mention of the maximum entropy production principle fits the remainder of the discussion in this paragraph.

**Comment 32: Authors' response**
Accepted. There has been quite a lot of speculation about the energetic boundary limits of dissipation in the literature. This was a nod to that but is not needed. We have identified two modes for the heat engine: free and forced. The forced mode reflects an excess of heat in the tropics resulting in thermodynamically-forced shifts being initiated in the tropics. In free mode internal climate variability dominates with a greater chance of shifts being initiated in extratropical regions if regional forcing is high enough. This suggests that there is an upper and lower limit to thermodynamic forcing and that the issue of entropy production and related matters such as switching between dissipative processes needs further exploration. It may be that both maximum

and minimum measures of entropy and/or efficiency are in play. In Appendix 2, we build on the propositions of Chaisson (2011) and Lovejoy (2019) that energy rate density, the energy flux as a function of mass, may be the unifying principle that is being sought for self-regulating thermodynamic systems.

**Comment 32: Change to the manuscript**
This point will be drawn into the description and discussion of the Granger results and ocean to land tracking. The last point will be proposed in the discussion of Paper 2 as described above.

**Comment 33 Lines 681-683:** the authors seem to suggest that the Pacific heat engine can be treated as a Carnot cycle. Clearly, the Carnot approach only provides an ideal constraint to the efficiency of the heat engine, and this should be clarified here.

**Comment 33: Authors' response**
Thanks. Following Laliberté et al. (2015), this constraint may be set by the efficiency of the hydrological cycle. The ideal water holding limit of the atmosphere defined by the Clausius-Clapeyron relationship is a 7% increase per °C, but atmospheric constraints limit this to around 2% (Held & Soden, 2006; O'Gorman et al., 2012; Richter & Xie, 2008) and aerosols may be limiting this even further (Salzmann, 2016).

We had not really addressed what the limits might be (see the previous comment and the response to Comment 11), but further investigation suggests the capacity for the meridional transport of latent heat holds the key. Warmer oceans support the transport of greater amounts of water vapour in the atmosphere. A global observational data set of land and sea humidity data has just become available (specifically, the marine and blended data (Willett et al., 2014)), which indicates upward shifts in specific humidity in 1995 and 2014–15 and a northern hemisphere shift in 1987. However, the most recent changes to relative humidity have been negative, with the SH leading in 2002–03. This may demonstrate some of those limits.

We believe that the limitations in the climate models regarding the hydrological cycle, fire danger (the hot and dry end of the hydrological cycle), the simulation of the deep tropics, tropical expansion and limitations in the representation of the heat engine in models are all related. In a complex system it is reasonable to consider that heat engines themselves can be nested across time and scale. The operation of a self-governing system therefore requires a network that bridges these scales.

**Comment 33: Change to the manuscript**
The updated conceptual model for the heat engine is proposed as the subject of a second paper.

**Comment 34 Sect. 6.5:** this section contains a very long review of a few previous works using network analysis. This includes ll.715-735, where results from Tantet and Dijkstra 2014 are extensively discussed. No original analysis is provided, supporting the consistency between mentioned literature and the results here shown. Given that this is manuscript is submitted as an original research article, I would suggest that the authors either provide an application of the network analysis in this context or remove this part.

**Comment 34: Authors' response**
We do not think providing a separate analysis is a reasonable request because (A) the timing of shifts via teleconnections makes it clear that this is a network we are dealing with and (B) it is reasonable to be able to draw on the literature to explain a new and potentially important set of findings.

Climatology is a natural science. A major task of the natural sciences is to understand how the world works. It was once the case that papers would combine original research and elements of review to speculate on the broader implications of a set of findings. The increasingly narrow definitions surrounding scientific scholarship tends to trivialise it, resulting in incrementalism that risks science devolving into little more than stamp collecting.

We can draw more strongly on point A, the timing of shifts, in a revised paper structure.

**Comment 34: Change to the manuscript**
The additional work carried out highlights the network connections more clearly, so the addition of literature that supports and sheds further light on how that network behaves should be appropriate because it will rely more strongly on our results.

**Comment 35 Lines 769-771:** again here, I think that the dichotomy "linear-nonlinear" is unreasonably emphasized, whereas the two frameworks are actually complementary. Therefore, I do not think that there is a lack of vocabulary for the nonlinear context, only because terms like "trends", "rate of change" etc. are extensively used.

**Comment 35: Authors' response**
We disagree. The frameworks are competing, both from a theoretical point of view and from a risk characterisation view. If we add the scientific values attached to terminology as part of paradigms (see Laudan, 1984) there is yet another layer of meaning (e.g., signal to noise as discussed earlier).

The referee argues that they are complementary because the referee believes they are complementary.

We have comprehensively surveyed the literature exploring the thermodynamics of climate change finding little agreement in the detail (pace response to Comment 30). If complex system science is thrown into the mix, there is little consensus and a number of possible approaches (see the section on model underdeterminism in Appendix 2). It may seem to a subject specialist that their language and terminology is consistent, but this is not the case. The discussion paper by Westhoff et al. (2019) has been withdrawn, but their point about a lack of a common language for thermodynamics stands. For example, the role of the second law of thermodynamics in representing climate can be and is represented in a number of different ways.

In contrast, when referring to trends, the main sticking point is to whether we are referring to a statistical trend or one that is also physically representative. This became a talking point of the so-called climate wars over whether the 1997–2014 hiatus in temperature was a physical thing, a perturbation due to climate variability (Lewandowsky et al., 2015; Risbey et al., 2018) or an artefact of the data (Karl et al., 2015). The overall consensus is that the underlying forcing produces a gradual response and that a combination of the above were involved (Marotzke & Forster, 2015; Medhaug et al., 2017). We wrote a working paper that claimed both sides were wrong and that the hiatus period is better characterised as a steady-state regime punctuated by rapid increases in temperature at either end (Jones & Ricketts, 2016). This was submitted and rejected by reviewers who cleaved to the noise generated-statistical artefact argument.

When referring to nonlinear phenomena, regimes continue to be variously defined (Overland et al., 2008) and break-points can be shifts, steps, trend-changes, bifurcation, thresholds, tipping points etc. climatology has a well-accepted tool-kit for analysing and interpreting one (trends) and a large set of contexts and methods for the other. Perceived ease of prediction and familiarity are two other factors that influence their relative ease of acceptance.

**Comment 35: Change to the manuscript**

As discussed, we have settled on complex systems science as offering the most suitable scientific context for describing the heat engine behaviour and conceptual model. This will settle on one broad set of terms. These will be accompanied by a glossary. This will overcome the confusion noted by both referees where terminology was taken from different subject areas and the discussion moved from one concept to another in a way that was hard to follow. The latter will also be dealt with by a more straightforward structure that has been informed by subsequent work carried out to address referee comments.

**Comment 36: Line 838:** I am not really sure what the authors mean by "disrupting" the climate system.

**Comment 36: Authors' response**

See the following clarification

**Comment 36: Change to the manuscript**

Change to 'which could lead to greater instability, such as ice-albedo feedbacks'

**Technical corrections**

The following corrections will be made if the text in question survives any revision.

Lines 390-392: the straight line the authors refer to in the text is not visible in Figure 5a.
Line 484: I believe that the authors refer here to 'CMIP5'.
Line 514: replace "additional" with "addition"
Line 544: remove "provides".
Line 693: add "be" after "may".
Line 831: remove "shift".

**Appendix 1: Granger Causality**

Annual and monthly temperatures from TEP, TWP, GMST and other regions (29 in total) were tested for correlations and Granger causality. Timeseries were de-stepped to produce stationary data and tested as observed for both the free and forced periods (1880–1967 and 1968–2018). De-stepping, an alternative to detrending, removes the change in mean during intervals identified by the bivariate test. If the time series is dominated by shifts, as we have shown, de-stepping is the more appropriate action.

Granger causality uses lagged regression. In a stationary timeseries this will test the effect of underlying periodicity, lags affected by phase changes, unit root behaviour and similar. Correlation at time t=0 reflects direct influences such as those due to circulation (monthly, annual) and direct teleconnections (annual). Correlation does not show the direction of flow but the Granger causality data at time t=1,2, … n can show the likely direction at time t=0, indicated by the direction of short-term lags and through lagged correlation. These can be one-way or two-way.

The Granger tests are used to analyse influence for up to lag-24 when possible, and lag-15 for annual data from 1968. For example, t=1 tests lag-1 whereas t=10 test all lags 1 to 10. If lag n+1 has a higher f-test result than lag n then it is considered to have greater influence, implying a causal relationship. For monthly data, the f-test results are preferred to p values because of the large number of samples. Because n is so large, a p-value of 0.01 can be quite weak, so the order of f-test results from large to p<0.01 is a better guide. Also, the tests are not measured against the null hypothesis but are conducted to gauge relative influence and how that changes over time. For annual data, p-values above 0.05 represent weak to no influence. Figure 9 shows the p=0.01 and 0.05 curve for f-test values n=139 (the full annual record). P-values are insensitive to n unless n is small.

[Figure]

Figure 9: P=0.01 and 0.05 curves for f-test values n=139 (the full annual record 1880–2018).

Testing the observed nonstationary time series disobeys the design rules of the Granger test when assessing it against the test null. Nonstationarity violates the principle of data independence, invalidating p-values. However, our main goal was to compare stationary against nonstationary

results and from that infer the non-stationary or forced contribution to change. The stationary results will mainly reflect circulation and teleconnections in steady state.

We conducted sensitivity tests with random data containing steps or trends and with various degrees of autocorrelation to aid in interpretation. Simple trends and steps produced almost identical results– a high lag-1 value for the f-test statistic, followed by a hyperbolic decrease to low $pH_0$ values. Introducing multiple lag effects makes the decrease less hyperbolic. We tested it for n=100 and n=1,000 as a proxy for annual and monthly and it produced a similar pattern of results for both, f-test values increasing for larger n.

The testing was done in pairs, rather than for the whole panel at once. This was for transparency, but generates much data that needs to be interpreted. The correlations produced 29 x 29 correlation matrices, but for the Granger tests we analysed each pair by interchanging the x and y variables to assess the direction of influence. For example, if we wanted to test the influence of TEP on all other variables, then 28 results would be produced. When testing all other variables against TEP, another 28 results would be produced. We tested six main pairs: GMST, land ocean, TEP, TWP and tropical ocean (20S–20N), influences to and from. As described below, this was done for periods of free and forced climate behaviour, stepped and de-stepped, annual and monthly, adding up to 2,688 pairs in total.

We tested pre- and post-1968 periods representing free and forced behaviour of climate. Monthly data contains 1,668 data points in total, 1,056 to 1968 and 612 from 1968. The de-stepped annual data was stationary using the ADF test and for monthly data was stationary for both the observed and de-stepped data. The de-stepped annual data was tested for stationarity and whiteness. All de-stepped records were stationary according to the ADF test. Using the Box-Pierce test, 23 of 29 passed for the forced period and 6 of 29 for the forced period, for the Ljung-Box test, 22 of 29 free and 5 out of 29 forced. This is consistent with our testing for GMST, and global land and ocean. The observed data was mainly nonstationary and non-white. The monthly data tested as stationary, but as we show below, was nonstationary with respect to the Granger test.

If shifts are the result of external forcing, removing shifts from timeseries will remove most of this influence. The testing on shifts using ANOVA, ANCOVA and unit root presented in the original Supplementary Information support the interpretation of external influence, as do the probative tests in JR2017. The Granger test treats both steps and trends as a trend, common to most time series analysis methods. We therefore used the de-stepped time series as a control and the observed time series as a measure of forced response, where the difference between the two patterns of change potentially show the main influence on the shift.

Correlations of monthly data at t=0 show how connected regions are on weather timescales. The Granger data shows lag effects and the direction of influence for up to 24 months. Annual data follows similar patterns. Testing TWP, TEP, global, hemispheric and zonal data for land-ocean, land and ocean allows different thermodynamic units to be compared. Lagged correlations will show the direction of change. For example, a negative correlation is where cool (warm) temperatures in one region are related to later warm (cool) temperatures in another. Combined Granger and Pearson (correlation) data can show whether temperature flows change direction, strengthen or weaken over time, how much influence TWP and TEP may have on other regions and how other regions affect them.

These interpretations are:
- If an x-y pair shows a large lag-1 effect with a following sharp hyperbolic decrease the influence is immediate and is usually preceded by a high correlation. For monthly data, this

shows direct circulation. The direction of influence can be tested by comparing opposing pairs' influence on each other, either with the Granger test or lagged correlations.

- If the difference between a stationary pair and the observed response shows a large lag-1 effect with a following hyperbolic decrease, the outcome can be interpreted as a simple shift. The direction of forcing can be tested by comparing opposing pairs' influence on each other combined with lagged correlations.
- If the effects decrease according to region size from large to small, the large region is the average of smaller regions that undergo a common change. If a small region shows the largest shift, it has a strong individual influence, especially if the larger region does not share that influence.
- Very low f-test results when the variables have changed by a similar amount over time suggests they are not linked, either physically or through teleconnections. This is evidence they are subject to different physical processes.
- When the stationary data shows strong influences, this may be a sign of continual flux if one-way (e.g., ocean to land) or may be due to variability moving in either direction over time. For example, ENSO effects in a stationary climate and other oscillations will do this.
- In some cases, the stationary time series may exhibit stronger Granger causality than the observed time series. This reflects a relationship that contributes strongly to variability and weakly to change.
- Patterns showing delay and evidence of periodicity (large peaks at lags 2–3 or long trailing influences) are likely to be genuine, even if the f-test and p values are inflated due to nonstationarity, especially with monthly data. Lagged correlations may be needed to test the direction of influence.

Given the breadth and complexity of data generated, we can only summarise the main findings.

The average monthly correlations in the correlation matrix for de-stepped free mode (1880–1967) range between 0.07 for TWP and 0.48 for GMST. Correlations show regional variations where land areas are more highly correlated than ocean areas and high latitudes the least. In forced mode (1968–2018), these correlations are slightly lower, showing a reduction in physical connectivity (from 0.07–0.48 to 0.01–0.42). For annual data, higher correlations in free mode reduce by a greater amount in forced mode (0.19–0.60 decreasing to -0.02–0.42). A t-test on ranked monthly and annual correlations shows the same relationship persists for each from free to forced mode, indicating no change in basic circulation structure, while their t=0 influence decreases.

TWP is the lowest ranked for correlation and TEP the most stable for average correlation when moving from free to forced, GMST the most stable for rank with the highest correlation. Less connected regions are 30–60S, 60–90N and 30–60N in that order (recalling that 60–90S has been left out).

The following summaries are for selected regions.

To TWP stationary (Plate 1): Monthly TWP shows little influence from outside before 1968, but afterwards gets short-term inputs from SH land and the SH. Annually, before 1968 TWP is influenced by the SH and SH ocean decreasing from lag-1, followed by the NH ocean and land at 2–3 years. After 1968 there is minimal influence, consistent with the record being dominated by white noise.

To TWP observed (Plate 1): Monthly TWP before 1968 is strongly influence by a lag-1 influence on all three ocean domains (NH, SH and global). After 1968, influences are stronger, extending to land and the tropics, showing a wider catchment area. Annually before 1968, TWP shows as similar pattern to the stationary series but stronger – the NH and SH ocean lag-1 and declining and a NH, SH and global

influence on a 2–3 year ENSO timescale. After 1968, most land and ocean regions have a strong lag-1 influence declining slowly over a handful of years. This is consistent with widespread contemporaneous shifts in temperature.

From TWP stationary (Plate 2): monthly TWP to 1968 strongly influences the tropical ocean and TEP with a 1-month lag. After 1968, monthly TWP follows a similar pattern, stronger lag-1 for tropical ocean and TEP and influencing the SH and NH ocean. Annually, TWP has little influence before 1968 but afterwards the tropical ocean has minor lag-1 influence and the SH, SH land and SH ocean a weak lag-2 influence.

From TWP observed (Plate 2): monthly TWP to 1968 strongly influences the tropical ocean and TEP with a 1-month lag, similar to but weaker than the stationary data. After 1968, monthly TWP influence is strongest on NH land and tropical ocean. strengthens in lag 3–4 and tapers over 12 months, with short-term weaker influences over the tropical ocean and land. Annually, before 1968 TWP had no influence, showing it was a net absorber of heat during this period. After 1968, annual data shows a strong lag-2 influence on the tropical ocean, tropical land and descending influences on the SH and NH regions. This is likely to be an indirect ENSO influence.

To TEP stationary (Plate 3): To 1968, monthly TEP gets its greatest influence from TWP lag-1 and rapidly declining and from the tropical ocean, a lag-3 peak tapering slowly. After 1968, the lag-1 influence from TWP is very strong and other regions have lesser lag-1 contributions, GMST and SH, then NH. The tropical ocean has limited influence. When large regions have greater f-test results than small regions, influences are widespread and homogenous. Annually before 1968, there is little incoming influence. After 1968, the tropical ocean and land have strong lag-1 influences followed by the SH. There is a NH peak at lag-14 that may be real with a sudden spike in correlation (0.34).

To TEP observed (Plate 3): To 1968, monthly TEP was most influenced by TWP lag-1 hyperbolic, and tropical ocean lag-2 tapering as in Plate 2 but changing places in importance. After 1968, tropical ocean has a lag-1 effect and TWP a sustained effect over most of the 24 months. Annually before 1968, there is no incoming influence, and a minor lag-2 peak afterwards.

From TEP stationary (Plate 4): To 1968, monthly TEP has a strong lag-1 hyperbolic effect on tropical land and ocean, the NH, GMST and SH. This show widespread short-term influence. From 1968, there is a large lag-1 hyperbolic effect on SH land, tropical oceans, GMST and global ocean. Annually before 1868, there is a lag-2 peak on SH land and tropical ocean. After 1968, there is a large lag-3–7 year peak on tropical land and GMST and a 2-year peak on NH and SH temps reflecting peaks in SST.

From TEP observed (Plate 4): To 1968, monthly TEP has as strongly hyperbolic influence on the tropical and SH land and tropical oceans and land affecting GMST. There is a 12-month exchange with TWP as also seen in Plate 2. From 1968, monthly TEP has a limited effect except for a short-term on tropical regions but the exchange with TWP is stronger. This suggests that short-term circulation through TEP has narrowed and is focused on TWP. Annually to 1968, the pattern resembles that for the stationary time series but is slightly stronger, suggesting that TEP and ENSO are not so much a source of warming but of stronger redistribution. After 1968, the lag-2 peak has strengthened, with a second 7-year peak for tropical land and more widespread tapering effects elsewhere lasting up to a decade.

Plate 1: To TWP Monthly (top) and Annual (bottom)

[Figure]

**Plate 2: From TWP Monthly (top) and Annual (bottom)**

[Figure]

Plate 3: To TEP Monthly (top) and Annual (bottom)

[Figure]

Plate 4: From TEP Monthly (top) and Annual (bottom)

[Figure]

Plate 5: To Ocean Monthly (top) and Annual (bottom)

[Figure]

Plate 6: From Ocean Monthly (top) and Annual (bottom)

[Figure]

Plate 7: To Land Monthly (top) and Annual (bottom)

[Figure]

**Plate 8: From Land Monthly (top) and Annual (bottom)**

[Figure]

**Plate 9: To GMST Monthly (top) and Annual (bottom)**

[Figure]

To Ocean stationary (Plate 5): To 1968, monthly global SST is influenced by TEP, TWP and tropical ocean in order of importance, the three declining but persistent. From 1968, monthly SST is much more strongly influence by TEP which persists and TWP which is short-term hyperbolic. Annually before 1968, SH and tropical lands have a hyperbolic influence. After 1968, peaks on lags-2 and 3 include TEP, SH land, tropical land and TWP show an ENSO influence.

To Ocean observed (Plate 5): to 1968, tropical ocean has an influence from month-3 that is persistent and TEP from month-12. After 1968, TWP also adds a persistent influence through month 24. Annually before 1968, TEP shows a year 2 peak and tropical ocean declines from year-1 with little influence after year 3. After 1968, there is a strong lag-2 peak in TEP, tropical ocean, GMST, land and TWP, where TEP persists at lower levels to year-9.

From Ocean stationary (Plate 6): to 1968, short-term hyperbolic patterns influence tropical and SH land, NH ocean, SH and GMST to lag-3. After 1968, monthly influences are slightly stronger and hyperbolic on GMST and NH (mostly land) and TWP. Annually, to 1968 there is a lag-3 influence on SH land and lag-2 on TWP. Annual influences following are minimal, with GMST the only variable registering. This shows the ocean has little effect on global climate during steady-state conditions.

From ocean observed (Plate 6): to 1968, monthly SST produces a strong lag-1 hyperbolic effect that has its strongest effect on GMST, expressed regionally though the NH, SH, SH land, tropical land and land regions except 60–90 N, and the NH ocean. After 1968, the strongest monthly influence is a lag-1 hyperbolic influence on all areas of land, SH strongest and very strong on 30–60S ocean and weak elsewhere on ocean regions. The exception is an acute lag-2 peak for TWP. Annually to 1969, the strongest effect is a lag-1 influence on TWP, lag-2 on SH land and lag-1 on the tropical ocean. After 1968, there also strong influences lag-1 on all areas of land, integrating into an effect lasting for up 7 years for global land; lag-1 over some areas of ocean and lag-1 for the tropics and 0–30N all combining for a moderate lag-2 effect for GMST.

To land stationary (Plate 7): To 1968, monthly influences are TEP and tropical ocean lag-1 and declining, with lesser influence from the SH and NH oceans. From 1968, monthly influences are minimal. Annual influences to 1968 are minor with some delayed influence from the NH ocean and following are TEP and tropical ocean on 2–3 year lag, along with NH ocean.

To land observed (Plate 7): To 1968, monthly influences show a similar order as for the stationary record, but with much greater influence, led by TEP, the tropical and NH ocean, all ocean areas contributing except the high latitudes. Lesser contributions come from land regions. From 1968, all ocean regions contribute a lag-1 hyperbolic influence except TWP and TEP and all land regions except 60–90 N. To 1968 annually, the only region contributing is 30–60N lag-1 declining. Following, contributions are strongest for global and NH ocean lag-1 declining, then lag-2 influences from TEP, 0–30N and 20S – 20N ocean, then lag-1 declining influences from TWP and other ocean regions. The only land contribution is a declining lag-1 from 60–90 N.

From land stationary (Plate 8): To 1968, on a monthly timescale, land mainly influenced other land, mainly the SH and tropics, tapering from lag-1 but sustained over 24 months. The overall cumulative effect of these changes has the largest influence on GMST, then NH temps. There is a lesser but sustained monthly effect on ocean, especially 60–90N. After 1968, there is no discernible effect on any land or ocean region but GMST and the NH have a strong lag-1 followed by a sharp decline, then a long taper. Annually to 1968, no regions are influenced except TWP at lag-1 but they also add up to a small influence on GMST at lag-1 tapering quickly. After 1968, SH and tropical south land have an influence with a slight aggregate year-2 peak on the ocean. These combine to a modest year-2 peak on GMST.

From land observed (Plate 8): To 1968, on a monthly timescale, land mainly influenced other land, mainly the NH influencing the SH, tapering from lag-1 but sustained over 24 months. The overall cumulative effect of these changes has the largest influence on GMST, the NH then the SH. There is a lesser but sustained monthly effect on ocean, especially 60–90N. From 1968 monthly, all land regions are affected and taper to over 12 months. For ocean, 60 – 90 N has a strong lag-1 effect tapering to 12 months and a weaker effect on tropical oceans. These add up to a large lag-1 effect on GMST and NH and a lag-3 effect on TWP.

Annually to 1968, the largest influence is a lag-1 effect between NH zones that does not add up to an affect for NH land, 60–90N land ocean lag-1 tapering and a slight year-5 peak on the NH. After 1968, the largest affects are lag-1 hyperbolic effects on NH extratropic land, a lag-3 peak on tropical land and a slightly stronger lag-2 peak on tropical, SH and global ocean. This we interpret as heat from land moving back through the ENSO cycle. The largest influences are a lag-1 effect on TWP to year 4, and a year 2–3 peak on NH and GMST tapering to year 7.

These records reveal two different stories over the historical period. Before 1968, the heat engine through TEP and TWP are much more integrated into the tropics, while the higher latitudes are quite separate.

Land shows little outside influence on a monthly basis during steady-state conditions, except for minor ocean inputs, weak before 1968 and slightly stronger afterwards. Under warming the strongest short-term input comes from ocean regions, much stronger after 1968, with some recirculation from land. Annually before 1968, there are no meaningful influences but afterwards in stationary conditions there is some immediate influence from 30–60 N land and stronger ENSO influence from the ocean led by TEP. Under warming there is a direct effect from the global and NH ocean, and tropical land followed by a TEP-led ENSO effect from the tropical ocean having an influence for about 7 years. The ocean is clearly warming the land.

For the ocean, the largest monthly influences on stationary conditions are TEP and TWP in that order, hyperbolic but persistent over the full 24 months. In stationary conditions, the tropical ocean had a smaller but persistent influence before 1968. After 1968, TEP and TWP influences strengthened and 30–60 N ocean and land have a hyperbolic influence. Land also had a feedback affect after about 12 months, mainly global and NH – lagged correlations suggests this is negative.

Under warming both TEP and TWP lose their short-term influence in the first few months but maintain the 24-month influence, the land feedback remains and the tropical ocean has a persistent influence. Annually, before 1968, the only influence under stationary conditions was SH and tropical land with a short-term hyperbolic effect. After 1968, this increases to include TWP and tropical N land and TEP and tropical ocean on a year-2 ENSO cycle. Under warming, the land had less of an influence before 1968 and TEP a small 2nd year influence. After 1968, tropical and SH land and tropical S ocean have a hyperbolic influence and TEP, tropical ocean and NH lands a year-2 ENSO influence.

[Figure]

Figure 10. Granger test results for TWP – TEP pairs stationary and observed (top), and related lag-correlations for the first fifteen years (second row); Granger test results for TWP, TEP – GMST pairs stationary and observed (third row) and related lag-correlations for the first fifteen years (bottom).

The greatest influence on GMST is TEP, followed by the various ocean regions. Under stationary conditions from 1968, TEP and the tropical and NH oceans have a multi-year influence on temperature over a period that matches the ENSO cycle to 7 years. This is supported by lag correlations between TEP and GMST, which is positive lags-1 and 4, neutral lag-3 and 7, and negative lags 2, 5 and 6. Under warming, TEP leads a strong lag-2 line-up followed by the N tropical, tropical, NH and global ocean then global land. In this instance, the lagged correlations between TEP and GMST are positive. TWP does not have as strong a direct effect on GMST, but it does influence both land and ocean on a strong 2-year lag, so we infer that is has an indirect influence through TEP. From Plate 1, we can see large amounts of heat flowing through TWP during warming periods although it retains little memory of this.

TWP and TEP themselves show an interacting opposite phase relationship (Fig. 10), which is clearest for the stationary data after 1968. Adding the lagged correlations (second row) shows clear reversals in the relationship (for reference the p=0.05 level is 0.27 and p=0.01 level is 0.35 for the post 1968 data, and 0.21 and 0.27 for the pre-1968 data). When both are compared to GMST, the opposite phases are even clearer with a year-2 peak. The pre- and post-1968 data suggest that coupling is weak before 1968 and stronger afterwards, supporting the separation of the two periods into free and forced. The reduction in correlation between the free and forced stationary data combined with a strengthening of an ENSO influence after 1968 indicates a decline in direct circulation and an increase in teleconnections.

This, along with Plate 9, indicates that in stationary conditions TEP is responding mainly on an ENSO cycle of 3–7 years, the positive and negative lags alternating between recharge and discharge behaviour. During warming periods, the ENSO cycle is strongest on a 2-year lag. The El Niño phase of ENSO broadly coincides with regime shifts, but removing steps does not remove all ENSO events, only some. TEP has only two shifts removed, so apart from those is remarkably stable.

This assessment supports the results from the tracking model and the analyses diagnosing free and forced mode in the heat engine. It emphasises the role of TEP and its importance in distributing heat. TWP is acting as a thermostat and a temporary storage, but the cold reservoir represented by TEP is the major distributor of heat. The intensification of ENSO in the late 20[th] century relative to pre-industrial periods prompted Timmermann et al. (2018) ask whether its evolution and amplitude are responding to external forcing. Based on these analyses, we see a key role for both the heat engine and ENSO in governing the climatic response to thermodynamic forcing.

**Appendix 2: Emergence**

**Shifts**

The evidence points to regime shifts in climate being emergent, similar to the emergence of ENSO and other oscillatory behaviour. The emergence of ENSO in climate models was the product of atmospheric and ocean coupling with sufficient resolution to distinguish features such as the tropical Pacific thermocline (Kim et al., 2014; Timmermann et al., 1999). Regime shifts appear to have a similar origin. For example, some models dating from the late 1990s and early 2000s (the generation that contributed to the IPCC AR3) display stationarity and step changes in temperature whereas others show continuous gradual warming from the outset. In Fig. 11, GMST from the HadCM3 model shows the most pronounced regime-like structure and the CCC model the least, producing an almost monotonic curve. Of a suite of five, the MPI model was the only one that produced stationary conditions at the regional scale in SE Australia. The two models showing shifts were amongst the first generation to produce an identifiable ENSO signal.

[Figure]

Figure 11: GMST from three GCM runs from the late 1990s, early 2000s. Top HadCM3, centre MPI ECHAM3 and CCCma CGCM1 model.

If regime shifts were emerging at roughly the same time as ENSO, the simplest explanation is that they are being produced by a common set of phenomena, the coupling of the ocean and atmosphere. This raises the question as to what other phenomena can be considered along with ENSO. The presence of attractors, teleconnections and the like would suggest that network behaviour is also emergent at that level. Emergence is a property of system complexity, influencing

both systems and models of systems. In modelling a complex system such as climate, the order of emergence is important. The features represented are also important for understanding what is driving and shaping emergent processes.

**The role of ENSO**

Dedicated models have been used to test whether phenomena such as ENSO display complex system behaviour but the question as to its emergence is less often addressed.

For example, the Zebiack-Cane (ZC) intermediate complexity ENSO model (Zebiak & Cane, 1987) is capable of representing chaotic behaviour (Ramesh & Cane, 2019; Tziperman et al., 1994). Ramesh and Cane (2019) reconstructed variations in Niño 3 15-year standard deviation as an attractor using observations (Kaplan SST), the ZC model of the tropical Pacific and two climate models from CMIP5: GFDLv2.1 and CCSM4. They constructed the simplest possible shadow manifold from a time series using a set of delays and settling on the pair with the lowest error. This technique is a simpler version of the method described by Champion et al. (2019) for data-sparse machine learning of complex system behaviour.

Their results displayed an inner and outer orbit with a bimodal distribution indicating high and low standard deviation. The points within the shadow manifold described by the attractor space were used to derive physical terms, which included predictability at the decadal scale. The ZC model provided reliable statistics (reliability was aided by very long simulations). Both observations and the CCSM4 model showed similar patterns, but the 160-year historical record was statistically the least reliable due to its limited length. The GFDL model (v 2.1) showed a different pattern (Ramesh & Cane, 2019). The authors attribute this to Pacific decadal variability being noise-driven in the GFDL model, rather than having a chaotic component (Wittenberg et al., 2014), which separates it from most of the CMIP5 models. The CCSM4 model is neither noise driven or chaotic.

The GFDL v2.1 model does produce shifts in GMST (it was analysed for JR2017), showing that such shifts can be produced by models dominated by dynamically-driven noise. This indicates that regime shifts may be a basic response to ocean-atmosphere coupling, as is ENSO. A key question is whether these emergent phenomena are separate from each other or are part of a larger system with unifying principles. A second question addresses the relative involvement of dynamics and thermodynamics.

In Appendix 1, ENSO is nominated as a major vehicle for warming shifts, especially after 1968. The El Niño–Southern Oscillation link was first proposed as a positive ocean-atmosphere feedback in the eastern Pacific (Bjerknes, 1969) but this was not recognised until the 1980s (Timmermann et al., 2018; Wang et al., 2017). This positive feedback needs a negative feedback to bring the system back to mean conditions and four oscillator models have been proposed: delayed, recharge, Western Pacific and advective-reflective (Wang, 2018; Wang et al., 2017) and one that integrates all four (Wang, 2001). Most attention has been focussed on the recharge-discharge oscillator focuses on in the eastern equatorial Pacific; specifically, eastern Pacific subsurface temperature and western Pacific thermocline change (Jin, 1997a, 1997b). A number of idealised modelling studies have been constructed that build upon this simple recharge model but the complex nature of ENSO remains unresolved.

Two current theoretical frameworks for ENSO describe an unstable, self-sustained oscillatory mode or a stochastically-forced stable mode (Timmermann et al., 2018; Wang et al., 2017). Wang (2018) favours the integrated oscillator model and Timmermann et al. (2018) propose a linear oscillator with nonlinear feedbacks or multiplicative stochastic forcing. Timmermann et al. (2018) also propose a unifying dynamic framework that identifies the following as key factors for ENSO complexity: two

primary ENSO eigenmodes that map onto the eastern Pacific (EP) and central Pacific (CP) El Niño modes, excitation processes, nonlinearities and cross-timescale interactions. The latter extends from monthly to decadal timescales.

Timmermann et al. (2018) calculated Empirical Orthogonal Functions on detrended SST 25° S–25° N and 140° E–80° W 1920–2016 showed for EOF1, the standard ENSO pattern of a warm anomaly along the equator in the eastern Pacific and a cool bow-shaped anomaly extending across the equator with ends facing east in the western Pacific (48.5% of uncertainty) and for EOF2 a bow-shaped warm anomaly in the central Pacific (same orientation) and a small cool anomaly in the SE (10.4%). The first has a frequency of 3–7 years and the second is quasi-biennial and decadal.

In a modified ZC model, the EP-type event is prominent when the mean thermocline is deep and the trade winds are weak, with strong thermocline feedback; whereas the CP-type event is dominant when the mean thermocline is shallow and the equatorial trade winds are strong, with zonal advective feedback (Xie & Jin, 2018). Negative feedbacks are dominated by thermal damping (Kim et al., 2014).

As CP El Niño has become more prominent, El Niño has also undergone nonlinear dynamic heating, promoting El Niño events and suppressing La Niña (An, 2009; Timmermann et al., 2018). A nonlinear dynamic heating response associated with such feedback is associated with the vertical displacement of water eastwards from the warm pool east of up to 200 mm in 1997–98 and 2014–15 (Peyser et al., 2016; Yin et al., 2018).

Timmermann et al. (2018) nominate 1968, 1994 and 2010 as the strongest recent CP events and these happen to coincide with shifts influencing SST south of the warm pool affecting SST around Australia. Each has also been followed by a conventional El Niño event associated with larger-scale shifts affecting ocean and land masses: 1972–73 (mainly SH), 1997–98 and 2014–15 (both global).

Comparing the periodicity of EOF1 and EOF2 with the Granger results, from Plate 4 after 1968 the influence from stationary TEP matches the EOF1 periodicity (3–7 years) and observed TEP matches the biennial aspect of EOF2. The same pattern is seen on Plate 9 where after 1968 the strongest influence on GMST is 3–7 years under stationary conditions and 2 years under change.

Appendix 1 also shows that El Niño has intensified in recent decades, the pre- and post-1968 data displaying different patterns. The reduction in cross-correlation with pairs in the stationary annual data after 1968 shows that direct circulation has decreased. The Granger to be taken up by the ENSO cycle. Comparing the stationary and observed Granger results post-1968 suggests that the shifts are mainly being driven on a 2-year lag, while dissipation under stationary conditions shows a 3–7 year lag. The intensification reflects the excess of heat in the tropics, producing more frequent regime shifts and an increase in dissipation rates.

According to Timmermann et al. (2018) both CP and EP El Niño operate in states of near criticality. It is likely that the upgradient relationship between TEP and TWP also operates in a state of near criticality. Fig 10 suggests that the two are coupled, flipping between opposite phases and the Granger analysis suggests they are the major contributor to changes in GMST. ENSO interactions therefore form an essential part of the heat engine.

Timmermann et al. (2018) propose two eigenmodes for ENSO, CP and EP El Niño, but +PDO and -PDO may also serve as eigenmodes on decadal timescales. The PDO has remained reasonably constant over the historical period, despite intensifying and decreasing in periodicity under forcing. In the discussion paper we raised the possibility that the heat engine operated within a broader

network of nested phenomena spanning space and time. We tentatively suggested that steady-state climate regimes form the basic physical climate unit. This allows the possibility for climate to be defined physically rather than statistically. This description has formed in this response with the identification of geostrophic controls on MHT as an important boundary condition.

The intensification of ENSO in the past fifty years relative to historical and reconstructed pre-industrial records also focuses attention on the role of external forcing on ENSO (Freund et al., 2019; McGregor et al., 2013). According to Timmermann et al. (2018), this may be due to long-term changes in background state modified by subsurface decadal variability. However, if both ENSO and decadal variability are fundamental parts of the dissipation of energy from the equator, a unifying thermodynamic framework may be more appropriate.

**A self-governing climate network**

The presence of teleconnections, switching behaviour and other phenomena suggests that the Pacific Ocean heat engine sits within a broader climate network facilitating the dissipation process. In the discussion paper we tentatively proposed this was the case and that it involved switching between slower modes of dissipation, such as the AMOC to faster modes in the atmosphere. The analyses presented here support that, showing a changing emphasis from circulation to teleconnections in observations from free to forced mode.

There has been a lot of emphasis on the potential for entropy-limited dissipation in the atmosphere, both maximum and minimum (Kleidon, 2009; Lorenz, 2002; Lucarini et al., 2010; Lucarini et al., 2011; Ozawa et al., 2003). A more general and less committal approach is to invoke efficiency or even more fundamentally, the principle of least action (Nicolis, 2000; Paltridge, 1978). Given that shifts in warming and cooling are both possible, depending on the direction of forcing, there is an upper and lower critical limit that can lead to regime shift. The identification of free and forced modes also suggests that different states exist. We interpret these as when the heat engine is tightly or loosely coupled into the broader network. Forced states are associated with high rates of dissipation producing white noise, whereas in free mode the system is 'relaxed' allowing land and ocean temperatures to go on random walks subject to the various modes of climate variability, principally the AMO.

One suggestion has been to use energy rate density for the atmosphere where it shows scaling properties over space and time (Lovejoy, 2019). Energy rate density has also been proposed as a more general metric for complexity (Chaisson, 2011). Such a principle would be consistent with moving from low density slow-moving fluxes, such as deep ocean circulation, through surface and boundary currents, dry air, moist air and atmospheric feedback processes where the medium of adjustment ranges from thermal and physical through to teleconnection (thermal to kinetic to telekinetic).

Induction is an important aspect of system behaviour that results in emergence. Patterns, including networks are induced, usually by an underlying set of unifying rules or principles. Self-regulating systems are invariably governed by a set of unifying principles, so if climate is self-regulating, there will be such a set.

The other aspect of energy transport via this network is symmetry. If either of the hemispheres generates an imbalance with the other via feedbacks or other processes, cross-equatorial energy flows will compensate. The capacity of the ENSO cycle to intensify or relax depending on the underlying thermal intensity will also display a similar symmetry. In the case where energy supply declines, which happens during glacial periods, we could predict a process of nonlinear dynamic cooling focused in the far eastern Pacific, where upwelling increase and the warm pool shrinks and

becomes less prominent. The idea of a self-regulating network has been proposed by Rial (2012) who investigated synchrony between the N and S polar systems in past climates.

**The order of emergence in climate models**

The appearance of emergent phenomena during the process of model development is not a smooth process. In the introductory comments, Referee 1 writes "*the propagation of the signal (cfr. Sect. 3.1) is discussed in terms of timing across different time series, and this qualitative argument severely undermines the robustness of the results. The authors seem to claim that conclusions similar to those based on observational-based datasets can be drawn from investigation of CMIP model outputs. A visual inspection of the model results shown in the supplementary material, though, does not seem to support this conclusion.*"

This raises two issues. One is that pattern matching in complex systems is perceived as soft evidence. We agree that pattern matching is an inferential process, so needs strong supporting evidence to be robust, however, we do not agree that qualitative arguments are necessarily weak, especially when dealing with complexity. Networks are inherently synchronistic. Part 2 of the discussion paper aimed to provide evidence for a sequential process of diffusion, where regime shifts originated in the tropics and propagated through climate system. In this response we provide stronger evidence.

The second issue refers to climate model performance, where we asked: (1) whether GCMs contain a Pacific Ocean heat engine and (2) whether the behaviour of those heat engines can be linked to shifts in GMST. The criteria we used for (1) were the presence of TWP, TEP, a constant gradient between the two and comparison of relative TWP and TEP shift size and frequency. We then used average shift size to estimate observed ECS of 3.2 °C. If the system was not internally consistent, this outcome would not have been possible. One criterion for (2) was based on the probability of random matches occurring within ±1 year between shifts in TWP, TEP and GMST in the models resulting in an average probability of p=0.001 and median $p=1.1 \times 10^{-8}$. In light of this, a 'visual inspection' carried out by the Referee does not seem very robust in itself, failing its own criteria.

The order of emergence of these phenomena in climate models needs to be better understood. The emergence of shifts appears to be a product of ocean-atmosphere coupling but from there the relationship is unclear. We carried out tracking exercises in three models, extracting regional data for up to nine variables, fewer than the 29 investigated for observations. For observations, two sets of regional shifts in the modern era were not associated with an initial shift in TWP. 1987–88 may have originated in the tropical Atlantic and affected mainly land in the NH and 2009–10 in the Australia region appears to be an expansion of the warm pool.

In the tracking exercise, some shifts were clearly initiated by TWP or TEP but others were not. Of total shifts in TWP and TEP all the models tested, 47% matched with GMST within one year and 48% of total GMST shifts were matched. For example, NorESM1-M has TWP shifting in Nov-1997 and TEP in Feb-1999, but NH land shifted in Dec-1995. We then looked to the N Pacific Ocean 30–60 °N and detected a shift in Oct-1995. Therefore, the model shifts in the mid-latitudes before the tropics, in different order to observations. Further investigation would be needed to assess which ocean basin was involved and whether there was a tropical origin in that basin. The following shift is led by the warm pool in Jan-2010, the next by TEP in Jul-2037, the next in the 2050s is staggered and messy, the following in the global ocean in Aug-2069 and the last in TWP in May 2089.

The models do not show evidence of being tightly coupled in their ocean-atmosphere systems at any time during 1860–2100 as they are in observations after 1968. We could not detect a distinct whitening in GMST or ocean data in these three models with the commencement of sustained

warming in the late 20[th] century. For TWP, MIROC-ESM is autocorrelated 1944–1991 and from 2071; for NorESM1 before 1937 positive, 1907–1930 slightly negative, 1931–2043 positive, more neutral after 2043 and negative from 2089; for CESM1 TWEP decreases from slightly positive to neutral from 1928. For observations, TWP whitens during 1921–1940 and from 1976, the two periods when active warming was occurring.

The large deficits appearing in top of the atmosphere energy balance in these models also indicates looser coupling than in observations (see response to Comment 26, Figs 6 and 7). It also indicates that excess energy at the surface and energetic limits to dissipation are driving shifts in climate regimes rather than any energy imbalance

The following aspects are robust in the CMIP5 models:
- The TEP region is represented by shifts that are about half as frequent and double the size as those in the west. These broadly match the historical pattern.
- Shifts in TEP are more highly correlated with ECS. This fits in with patch experiments, which show that warming in the eastern Pacific has high feedback and in the western Pacific has low or negative feedback. Applying regressions between shift size in TEP and TWP and observations for TEP suggests a climate sensitivity of 3.2 °C.
- The gradient between TEP and TWP in the models remains constant under warming, showing that the models are self-regulating despite not mirroring observations in how they change.

However, until model performance is improved to better represent the Pacific Ocean heat engine, and that performance can be recognised in the representation of the hydrological cycle and realistic representation of latent heat flux and radiative fluxes at the surface and top of the atmosphere, their performance on decadal time scales needs to be treated with caution.

In any revision we will be very clear about the criteria we apply when comparing the models to observations by articulating the issue of emergence.

In climate models, we would expect basic phenomena to emerge well before they can reproduce ordered processes in reasonable detail as has been the case for ENSO. ENSO has been emergent in climate models for roughly 20 years, but had the advantage over other emergent phenomena because it had already been identified as important. People were looking for ENSO in climate models, analysing its behaviour and using simpler idealised models to understand that behaviour. They have not been looking at decadal regimes with anywhere near the same interest until comparatively recently.

Is it possible that when ENSO emerged from the models, a teleconnected global network also emerged? The presence of step changes implies that may be the case. What emergent behaviour might we expect from climate models as they become more physically realistic? Our analyses suggest that current representations of the heat engine are fairly crude. How well does the tropical Pacific need to be represented for a more realistic representation? Even if the CMIP5 ensemble under historical forcing is positively correlated with historical timing of shifts in the 20[th] century, might the sequence of shifts follow those in observations or not? Our analyses of three climate models suggests sometimes, but not as reliably as in observations.

**Model underdeterminism and system complexity**

The modelling of emergent phenomena in complex systems, both physical and statistical, is beset by the problem of underdeterminism. This due to strong boundary conditions and/or imposed limits where there are multiple pathways to potential solutions. Simple models may predetermine such

pathways, and then the issue becomes which simple models produce a realistic-looking solution. For example, evaporation over the land surface can be estimated using energy balance or wind speed (Dalton) methods, obtaining similar results. Idealised models than contain nonlinear processes intended to represent emergent processes can have similar issues. The reviews of ENSO studies cited above are replete with examples where the authors are attempting to select the most 'scientific' explanation from a set of candidates that have all passed scientific review. Similar problems accompany energy balance approaches to understanding climate and thermodynamic approaches utilising heat engines.

Another issue for emergence in model-based studies is the use of nested models representing a hierarchy from complex to simple. This framework was first described in detail by Schneider and Dickinson (1974) and is still endorsed as a way of breaking down the complexity of the climate system (Manabe, 2019). The problem with this approach is that it can bypass emergent processes because in terms of the governing concept, it is deterministic all the way down. In the standard model hierarchy, this concept is radiative forcing. The ultimate effect is to linearise the emergent processes in the climate system.

Underdeterminism in these areas (not to be confused with the philosophical debate over theoretical underdeterminism) is best dealt with by using a whole-of-system approach. Understanding the governing principles of a self-regulating complex system is often used in applied ecology which aims to determine the rules in play from observation rather than using a bottom up, deductive first principles approach. This invites comparison with the Gaia hypothesis (Lovelock, 2003; Lovelock & Margulis, 1974), which proposes a coevolutionary role for biology and the atmosphere that keeps the Earth in a habitable state.

Gaia hypotheses can be divided into weak and strong types (Kirchner, 1989). For weak Gaia, biology builds on pre-existing stabilising tendencies evolving feedback processes that make the climate more habitable, whereas in the strong Gaia hypothesis biology drives the planet towards a more stable climate and habitat. Kirchner (1989) argues that abiotic systems do self-organise but most assessments of the Gaia hypothesis focus on the biological aspect (see Rubin and Crucifix (2019) for a review). Rubin and Crucifix (2019) argue that the Gaia hypothesis goes beyond the question of self-regulation to self-maintenance and repair, or autopoiesis (Lovelock, 1987).

Margulis viewed Gaia as an ecosystem where the inorganic and organic aspects of the earth system are in a symbiotic relationship that is homeorhetic (Margulis, 1981), or following a particular trajectory. Part of what informed the Gaia hypothesis is the recognition that Earth's climate can fall into stable macro-states such as snowball earth, some of which would be unfavourable for large-scale ecosystems (Watson & Lovelock, 1983). These states can evolve in an abiotic environment but only a selected few are recognised; e.g., snowball, waterbelt, temperate and moist greenhouse (Lucarini & Bódai, 2017; Wolf et al., 2017). They are separated by abrupt climatic transitions.

[remaining 36,580 characters of this post omitted]

---

## Author Comment (AC3) · 20 Nov 2020

Overall summary: Referee 2

This work builds upon a previous study of the authors that showed that decadal scale warming as monitored by Global surface temperature is not "trend-like" but behaves like a progression of fast warmings ("shifts") and periods of stagnation. The present paper tries to relate this concept to processes related to ENSO and the "Pacific heat engine" by statistical analyses of warm-pool and cold-pool temperature time series, their relation to climate indices (PDO,AMO,AMOC...) in the context of shifts. A conceptual model of the heat engine is presented in a second part of the paper. While I find the concept very interesting and I appreciate the effort of detailed analyses, I found this

paper very difficult to read. It is very long, wordy and not very focused.

The many statistical analyses and detailed descriptions are difficult to follow. How the conclusions made in the text arise from the Figures is mostly difficult to see.

Overall summary: Authors' response

Thank you for the review and the time taken to consider the paper. We agree that the paper did not settle on a particular focus and could be made much clearer.

In our view, the previous work (Jones & Ricketts, 2017) showed that the current paradigm of the climate changing gradually over decadal timescales could not be sustained. When time series of GMST from the CMIP5 RCP4.5 ensemble 2006–2095 were divided into shifts and internal trends and separately regressed against ECS (n=93), shifts explained 2.9 times the variance of results than trends. Shifts also dominated internal trends during 21st century warming (average 0.73 °C vs 0.43 °C 2006–2095, n=107). For observations, shift/total warming ratios of up to 100% across some regions also preclude warming being trend-like.

This is consistent with a storage and release system, where the additional heat being trapped by greenhouse gases in the lower atmosphere is being absorbed and stored by the shallow ocean. If the shallow ocean–atmosphere is accepted as being in steady-state during the above-mentioned 'periods of stagnation', then there is little scope for gradual warming, because there is no barrier to the uptake of atmospheric heat Upon release, atmospheric feedbacks respond immediately. An analysis of temperature in the central eastern Pacific (TEP) and western Pacific warm pool (TWP) led us to believe the relationship between the two was central to this process, so we developed the tracking model to assess where and when shifts occurred.

Consistent with the severe testing approached taken in JR2017, we used statistical induction to interpret test results while aiming to develop probative criteria that could be used to assess those interpretations against plausible alternatives. The above ex-

ample of using ECS to test step-like against trend-like components of warming is an example. The main drawback is that by rejecting the current paradigm, we also reject the probative criteria currently in use. For example, criteria that justify the use of the signal-to-noise model as representing an underpinning physical process (i.e., the conversion of radiative forcing into atmospheric temperature mediated by ocean heat uptake).

Developing probative criteria for testing nonlinear deterministic change in climate time-series is complicated. The usual interpretation is to combine linear response theory with a stochastic model that has a deterministic component as advocated by Referee 1 (see Ghil, 2015; Ghil & Lucarini, 2020). However, we have already rejected this as an alternative because it over-emphasises the role of gradual change. While gradual change is a potential response to some types of forcing (e.g., anthropogenic sulphate aerosols, ongoing land-use and land cover change), we conclude that this is not the case for greenhouse gas forcing.

Without a unifying principle, evidence may be drawn from a variety of different sources, which include statistical climatology, hydroclimatology, complex system dynamics, fluid dynamics, statistical physics and idealised model approaches (e.g., energy balance models, attractor models). We drew from these hoping the reader could follow, but the lack of focus was a significant barrier.

We have prepared these responses by concentrating on complex system dynamics, specifically fluid dynamics (although qualitatively). Although this focus makes sense for a heat engine dissipating energy in a coupled liquid and gas and gas environment, it was unclear in the discussion paper as to what the controlling boundary conditions were: top of the atmosphere energy deficit, excess surface temperature, some measure of Earth's energy imbalance involving both, an imbalance between TEP and TWP or something else.

Our aim is to develop a qualitative description at the global climate scale that best fits
the evidence and can be subject to further testing. This method may not appeal to those who work quantitatively from first principles but is intended as a starting point. Champion et al. (2019) argue that complex systems are not amenable to approaches based on first principles and that many of the quantifiable relationships that can be derived from data will be descriptive. For example, attractors describing ENSO or decadal oscillations.

A key insight came from work on developing fire climates, and was prompted by Chadwick et al. (2016) describing changes in relative humidity over land as a thermodynamic response. We tested relative humidity time series from two data sets (Lucas & Harris, 2019; Willett et al., 2014), identifying changes over Australia, both hemispheres and globally as regime shifts. Changes in latent heat supply are regime-like and the timing shows they usually follow temperature changes (but occasionally change together). Model output also shows that energy transport is dominated by latent heat transport and that changes in surface latent heat flux are regime-like.

The main governing principle for the global heat engine is the meridional transport of energy from the equator to the poles (See the response to Referee 1 for more detail). Under positive forcing this mainly involves the atmospheric component, especially changing latent heat measured as the divergence of moist static energy. Geostrophic limits provide extremely tight boundary limits to the transport of meridional energy, where modelling studies show it changes little between glacial periods and 4 x $CO_2$ (Donohoe et al., 2020). This suggests that changes in variables such as temperature, humidity and latent heat transport are thermodynamically forced.

The Pacific Ocean heat engine itself is composed of stationary and transient components. The stationary components involved include the Hadley and Walker Cells and the transport of heat from cold reservoir to warm pool. These supply much of the vertical transport from the equator, and also are the source of meridional transport.

When the heat engine is working in forced mode, meridional transport is initiated by an
excess of heat in the tropics. The evidence from regime shifts suggests that on decadal timescales, such forcing is nonlinear, resulting in shifts in mean between steady-state regimes that alter dissipation rates, in the same way that an engine may shift gears. Meridional limits in heat transport are maintained while the mode of transport, being comprised of dry and moist static energy in the atmosphere, ocean heat transport and kinetic energy can alter to accommodate the required rate of dissipation to maintain steady-state.

In free mode, forcing in the extratropics could therefore result in shifts being initiated in those regions and propagating back to the tropics, as probably occurred in 1920 and possible at other times in the first part of the 20th century.

Radiative feedbacks are part of this adjustment process, with shifts in surface temperature heating the atmosphere, resulting in cloud feedbacks and altering absorbed shortwave and emitted long-wave fluxes. The heat engine therefore acts as a network, where in the early 20th century the influence of the Pacific Ocean heat engine was largely tropical and subtropical with the extratropics being able to do their own thing. The increase in forcing expands the influence of the tropics, increasing the strength of teleconnections but also resulting in increased spatial feedback from the high latitudes towards the tropics and between the hemispheres. Slow modes of dissipation decrease in influence, while rapid modes increase, reducing physical circulation at the expense of teleconnections.

The first part of the paper, therefore traces the behaviour of the heat engine and its global network, especially of the regime changes responding to thermodynamic forcing. While concentrating on temperature, atmospheric moisture fluxes are important while harder to measure, and radiative fluxes even harder still. Temperature analyses can rely on observational networks but other variables also require remote sensed, reanalysis and climate model output, which are limited by the capacity of those sources to reproduce historical change.

Our proposal is to split the paper into two as advised. The first paper would include the statistical analyses and tracking models of observed climate. Observations are the primary focus.

The second paper will elaborate on the above description, taking the findings from the first paper and adding some additional analysis as illustrations of specific points. It will also draw widely from the literature in order to construct a conceptual model of the global heat engine. We will argue that the current model of radiative forcing producing a climate response is missing a key component, a 'hidden climate' where thermodynamic forcing produces nonlinear responses on decadal timescales. Because these responses are boundary-limited by a gradual increase in radiative forcing they produce complex trends. These complex trends are similar in overall rate and magnitude to the simple, monotonic trends being used to measure the long-term climate signal, but on decadal timescales the signal is nonlinear.

Overall summary: Change to the manuscript

Revised contents for the two papers are presented in the response to referee 1.

Referee's comments

The following changes need to be made before it can go into a more rigorous review of the science:

Comment 1) The paper should be substantially shortened. I suggest to leave part 2 out and leave it for a separate paper.

Comment 1) Authors' response

As mentioned above, we agree with separating the paper into two, having generated some data to add to the first and a large amount of material for the second paper. One paper will focus on the behaviour of heat engine processes and the other on climate as a complex dynamic system, where thermodynamic forcing is the driving process behind how that system behaves.

Comment 1) Change to the manuscript

Revised contents provided in response to Referee 1's overall comments.

Comment 2) Introductory material is mixed-in with the results. Please clearly distinguish between introduction/discussion of previous work and unique results from the present study.

Comment 2) Authors' response

We agree. We propose to have a short introduction clearly stating the purpose of the paper(s), restate the case for regime-like warming and briefly describe the evidence that will be presented. This will be followed by a context setting section that will cover the previous work describing severe testing and updated conclusions, how nonlinear change is measured within time series, the approach taken to complex systems in the paper, a brief section sign-posting methods and tools used and links to the supplementary information and the physical setting.

Comment 2) Change to the manuscript

Changes detailed in response to referee 1.

Comment 3) I suggest a clear re-structure of the paper: Describe in the introduction in a focused way the starting point, also the state-of-the art with respect in relation to climate indices etc (a lot of this now is mingled in with the results sections). It should be clearly stated at the end what hypotheses you are investigating or which questions you are addressing. After the introduction, there needs to be a methods and data section where the statistical methods are explained and data described. At present, this is all in the SI, and e.g. the "tracking model" and how it is used is unclear. The methods section needs to lay out how the hypotheses will be tested with the statistical methods. The results sections are overloaded with detailed information but the storyline is not clear. Methods description and results need to be delineated. The text needs to directly refer to the Figure panels so one can track the conclusions drawn.

Comment 3) Authors' response

Thank you for these suggestions – this is now part of the suggested restructure

Comment 3) Change to the manuscript

Changes detailed in response to referee 1, comprised of new table of contents and reorganised sections.

Comment 4) Use present tense when referring to the results. It is very difficult to track whether described results are from the present study or referring to some previous results.

Comment 4) Authors' response

We agree, and apologise for the confusion. The revised structure will make this more straightforward.

Comment 4) Change to the manuscript

The appropriate changes will be made to reported results.

Comment 5) There are many unclear and unscientific phrases throughout the paper: e.g.: ... "the heat engine is networked within the climate system" (abstract)... Examples from the intro: ...the process is "regulated" by a heat engine spanning ... ...shifts are "linked to the wider system of climate oscillations"... ...the difference between "this" and the standard model "in terms of energy flow" .. ...how it interacts with "broader climate"... section 2: ...heat "being channeled and made available for dissipation"... page 11: ..."because of issues with".... page 17: ...the "tightening of the system" coincided with the heat engine moving from free to forced mode... etc. There is a need to go through the paper again and make statements as clear and scientific as possible.

Comment 4) Authors' response

Agreed and accepted – the lack of a focus for the science and a lack of agreement in

technical definitions and language when dealing with nonlinearity and complex systems made this difficult. By focussing on complex system dynamics and taking a conservative approach to thermodynamics we hope to make this much more straightforward.

The context setting planned as an extended introduction will provide a guide as to the general approach and define important terms. For some of the phenomena we will be describing, the thermodynamics as to how they distribute energy is uncertain. Our interest is in the effect in terms of measurement and how that might flow onto risk.

The additional work done to clarify points raised by both referees has helped to increase our understanding of how the climate system is distributing energy, which can flow through to clear descriptions of those processes. Some of the phrasing will remain descriptive, and therefore 'unscientific' because the system is complex, but such descriptions will be standardised and defined where needed.

Comment 4) Change to the manuscript

The text will be gone through with this in mind and language and meaning clarified.

Structural comments:

Comment 5) Figure 1a is something you find in any climate/ocean dynamics textbook. Instead of this rather basic Figure it would be better to redraft something related to the heat engine hypothesis, recharge-discharge theory with focus on your hypothesis.

Comment 5) Authors' response

We felt it useful to start with a fairly basic description because that's what people generally understand. We will add some details

Comment 5) Change to the manuscript

Some minor details added but we want to keep this fairly straightforward at the start

Comment 6) Section 1: Bullet points 1-6 read like the conclusions section from the

Jones and Rickets 2017 paper and are not needed in this way here. Line 77: what is meant by subsequently and elsewhere? Line 84: here, hidden in a side sentence, it is stated for the first time what is the subject of the paper. It is not clear though what is meant by "processes" because I find this paper to be mainly about statistical analyses and not necessarily the underlying processes.

Comment 6) Authors' response

We have discussed both referee's comments with regard to our failure to convey why these findings are so important and have proposed changes. Line 77 is just not clear and line 84 is not well sign-posted in the preceding description.

We intend to make this clearer in the context section, but the approach we took to testing the difference between gradual and abrupt change in JR2017 is error testing. Mayo (2018) defines an error probability as the probability that a method commits an erroneous interpretation of data, and error statistics the practice of exploring this. Error testing is described as both a philosophical and statistical approach to statistical induction (Mayo, 2010; Mayo & Spanos, 2011). In JR2017, we concluded that the evidence for steplike change was overwhelming. We will emphasise the key results here rather than repeating bullet points 1–6.

The intention of the paper is to use statistics to explore the underlying processes resulting in change on decadal timescales. The non-standard nature of the testing (i.e., testing for shifts rather than trends) is also because we are working with a complex dynamic system and trying to infer the mechanisms behind specific behaviours. This falls in to the area of mechanistic philosophy (Craver & Tabery, 2019), which has largely arisen due to the problems with understanding complex systems, drawing on and developing methods for understanding mechanisms that may not be tractable to the use of more traditional deductive methods.

Accordingly, we will be much clearer about processes, using more precise terminology that makes use of the concepts of emergence and organisation. Many of the tests used
are detection tests designed to measure timing and effect and we will be more precise about what those are. From those we wish to infer various mechanisms associated with complex system dynamics and how they may be influencing the climate system.

Comment 6) Change to the manuscript

Some of the material in the previous paragraphs will be incorporated into the context section describing the general approach, and how the statistical methods and tools will be used to inform the analysis and results.

Comment 7) Section 2 and the beginning of section 3 are introduction material. What exactly do you mean by heat engine "behaviour" in the title section 3? Line 158 suddenly results start. Lines 169 ff: too many numbers, difficult to follow.

Comment 7) Authors' response

We agree that the mixing of introductory material and results is confusing and the introductory material would go into an expanded context section. The proposed new section 3 is to be called results. The results from Line 169 will be woven into more of a narrative that discusses the timing of the various changes.

Comment 7) Change to the manuscript See above

Comment 8) Section 3.1: first sentence: where do you show this? explain tracking model in methods, it is otherwise very difficult to follow section 3.1

Comment 8) Authors' response

This first sentence belongs later on. We have other reasons for hypothesising that shifts in warming may originate in the Pacific due to the additional accumulation of heat from forcing, but the tracking processes needs to be shown before we can argue this properly. Agreed, the tracking model should be explained in methods and the reasons for constructing it described better.

Comment 8) Changes to manuscript

The tracking model and attached hypotheses being tested will be described in methods and tools. The delineation of free and forced, which influences how the tracking model works will be described in the new proposed Section 3.1 describing TWP, TEP and their relationship with the broader climate.

Comment 9) Section 4: First sentence: where do you show this? Lots of material and references to previous studies to be moved and discussed in introduction.

Comment 9) Authors' response

This was meant to be a mini abstract for this section but clearly does not work. Changes will be made as suggested, similar to those recommended for Section 3.

Comment 9) Changes to manuscript

The methods used for testing whether decadal oscillations may be linked to shifts in temperature will be introduced in the context section describing methods. The evidence for their being linked to the climate network and hypotheses that arise from this (shifts being propagated through teleconnections) will be introduced in Section 2.3, describing climate as a dynamic complex system. Section 3.3 would present the results.

Comment 10) Section 5.3: is this needed? It seems like a proper analysis and description of climatological data is enough.

Comment 10) Authors' response

There were three main motivations for doing the model comparison: (1) Because regime shifts are an emergent phenomenon, we wanted to see whether the basic functions of the Pacific Ocean heat engine were present in climate models, and if so, to what degree. (2) We are proposing that one of the most widely-held views about how the climate changes is incorrect – if our proposal is accepted then we would like to avoid charges of the science being fundamentally wrong. If models sustain our thesis by representing the basic characteristic of a heat engine, then only the interpretation of their output is misplaced, not the basic science itself. (3) Climate models are promoted as the primary source of information about future climate risk which, as we have argued in the past, understates that risk (Jones et al., 2013). If observations capture a degree of risk not present in models, then it is important that these are captured and used to inform future risk planning. There is also a line of argument in some assessments that if a mechanism is not present in climate models under specified boundary conditions, its existence in the real world is in doubt Mann et al. (2020). To address this, it is important to identify whether or not a particular mechanism exists in a model, especially if it is being overlooked.

In response to comments from referee 1, we have investigated model performance in more detail, extending to variables where observations are relatively brief (e.g., energy fluxes). Although these are preliminary as we have only looked at several models, they show where important links can be made when comparing observations, reanalyses and earth system models. These are now to be incorporated in paper 2.

Comment 10) Changes to manuscript

A revised section on the performance of the heat engine in models is planned for the second paper, where a brief summary of energy fluxes will be added. This will also support the discussion on the heat engine and self-regulation.

Comment 11) Conclusions: here I suggest to clearly distinguish what are new hypotheses and interpretations rather than conclusions drawn from the statistical analyses in this paper. A clear summary of the statistical results needs to be presented first.

Comment 11) Authors' response

Thank you for this suggestion. The proposed revision to the research context should set this up better. The conclusions from the first paper will present the results of tracking the evolution of warming through the system, key network interactions and put the case for there being free and forced modes during the 20th century.

The second paper will address the new hypotheses and interpretations using the findings of the first paper, with additional analyses from climate models backed up by more analyses from observations and reanalysis models. These hypotheses are now much clearer and focus on how thermodynamic forcing influences dissipation in the coupled ocean-atmosphere system. This system forms a self-governing network with the Pacific Ocean heat engine at its centre. Powerful boundary conditions, such as the geostrophically-limited meridional heat transport curve between the equator and the poles, the ocean subsurface and top of the atmosphere set the physical and energetic limits of the system, and changes in radiative forcing determine the amount of heat available for dissipation.

The behaviour of the whole system is based on the idea that as a complex system, the climate autonomously settles on the preferred modes and pathways for dissipating energy. These can be crudely lumped into what we consider to be 'climate variability'. When changes in forcing are very small, climate is dominated by dynamic processes. When forcing changes, thermodynamic processes become more prominent and climate will pass through a series of thermodynamically-forced steady-state regimes until it achieves a relaxed state. This model suggests that energy imbalance leading to thermodynamic forcing is an excess of energy within the meridional dissipative system, moderated by as yet poorly understood thermodynamic rules. In a positively-forced system, this will be excess of heat at the equator. Earth's energy imbalance is therefore a symptom of dissipative imbalance. Rebalancing the deficit (or excess) at the top of the atmosphere is an outcome, rather than a causal factor driving climate change.

Comment 11) Change to the manuscript

The two-paper proposal in the final response to Referee 1 should address these concerns.   References

Chadwick, R., Good, P. and Willett, K. (2016). A simple moisture advection model of specific humidity change over land in response to SST warming. Journal of Climate, 29(21), 7613-7632.

Champion, K., Lusch, B., Kutz, J. N. and Brunton, S. L. (2019). Data-driven discovery of coordinates and governing equations. Proceedings of the National Academy of Sciences, 116(45), 22445-22451. doi:10.1073/pnas.1906995116

Craver, C. and Tabery, J. (2019). Mechanisms in Science. In E. N. Zalta (Ed.), The Stanford Encyclopedia of Philosophy (Summer 2019 ed.). Stanford: The Metaphysics Research Lab, Center for the Study of Language and Information, Stanford University

Donohoe, A., Armour, K. C., Roe, G. H., Battisti, D. S. and Hahn, L. (2020). The Partitioning of Meridional Heat Transport from the Last Glacial Maximum to $CO_2$ Quadrupling in Coupled Climate Models. Journal of Climate, 33(10), 4141-4165. doi:10.1175/jcli-d-19-0797.1

Ghil, M. (2015). A mathematical theory of climate sensitivity or, How to deal with both anthropogenic forcing and natural variability? In C.-P. Chang, M. Ghil, M. Latif and J. M. Wallace (Eds.), Climate Change: Multidecadal and Beyond. London, Singapore: World Scientific Publishing Company.

Ghil, M. and Lucarini, V. (2020). The physics of climate variability and climate change. Reviews of Modern Physics, 92(3), 035002.

Jones, R. N. and Ricketts, J. H. (2017). Reconciling the signal and noise of atmospheric warming on decadal timescales. Earth System Dynamics, 8(1), 177-210. doi:https://doi.org/10.5194/esd-8-177-2017

Lucas, C. and Harris, S. (2019). Seasonal McArthur Forest Fire Danger Index (FFDI) data for Australia: 1973-2017. Retrieved from: https://data.mendeley.com/datasets/xf5bv3hcvw/2

Mann, M. E., Steinman, B. A. and Miller, S. K. (2020). Absence of internal multidecadal and interdecadal oscillations in climate model simulations. Nature Communications, 11(1), 1-9.

Mayo, D. G. (2010). Learning from error, severe testing, and the growth of theoretical

knowledge. In D. G. Mayo and A. Spanos (Eds.), Error and Inference: Recent Exchanges on Experimental Reasoning, Reliability, and the Objectivity and Rationality of Science (pp. 28-57). Cambridge UK and New York USA: Cambridge University Press.

Mayo, D. G. (2018). Statistical Inference as Severe Testing. Cambridge: Cambridge University Press.

Mayo, D. G. and Spanos, A. (2011). Error Statistics. In P. S. Bandyopadhyay and M. R. Forster (Eds.), Philosophy of Statistics (Vol. 7, pp. 153-198). Amsterdam: North-Holland.

Willett, K., Dunn, R., Thorne, P., Bell, S., De Podesta, M., Parker, D., . . . Williams Jr, C. (2014). HadISDH land surface multi-variable humidity and temperature record for climate monitoring. Climate of the Past, 10(6).